# Interphase chromosome conformation is specified by distinct folding programmes inherited through mitotic chromosomes or the cytoplasm

Allana Schooley[1], Sergey V. Venev [1,2], Vasilisa Aksenova [3], Jesse W. Lehman[4], Emily Navarrete[5], Athma A. Pai[4], Mary Dasso [3] & Job Dekker [1,2] ✉

Identity-specific chromosome conformation must be re-established at each cell division. To uncover how interphase folding is inherited, we developed an approach that segregates chromosome-intrinsic mechanisms from those propagated through the cytoplasm during G1 nuclear reassembly. Inducible degradation of proteins essential for the establishment of nucleocytoplasmic transport during mitotic exit enabled analysis of folding programmes with distinct modes of inheritance. Here we show that genome compartmentalization is driven entirely by chromosome-intrinsic factors. In addition to conventional compartmental segregation, the chromosome-intrinsic folding programme leads to prominent genome-scale microcompartmentalization of mitotically bookmarked *cis*-regulatory elements. The microcompartment conformation forms transiently during telophase and is subsequently modulated by a second folding programme inherited through the cytoplasm in early G1. This programme includes cohesin-mediated loop extrusion and factors involved in transcription and RNA processing. The combined and interdependent action of chromosome-intrinsic and cytoplasmic inherited folding programmes determines the interphase chromatin conformation as cells exit mitosis.

Interphase chromosome folding is specific to the transcriptional programme of a given cell type[1–4]. Two major processes contribute to this folding: loop extrusion mediated by cohesin and spatial compartmentalization that is probably driven by homotypic affinities of chromatin domains distributed across the genome[5]. Within the three-dimensional (3D) space of the nucleus the genome is segregated into broad euchromatin/heterochromatin compartments and multiple constituent subcompartments that form within (in *cis*) and between

(in *trans*) chromosomes[6]. Heterochromatic loci cluster together, often at the nuclear periphery, and active chromatin domains self-associate most frequently in the nuclear interior. The molecular factors that determine compartmentalization remain poorly understood[7,8]. The distribution of histone modifications along the genome is strongly correlated with subcompartment formation[6,9,10] and it is possible that these modifications display homotypic affinities (that is, histone 3 lysine 9 trimethylation, H3K9me3; ref. 11). In addition, factors that bind

[1]Department of Systems Biology, University of Massachusetts Chan Medical School, Worcester, MA, USA. [2]Howard Hughes Medical Institute, Chevy Chase, MD, USA. [3]Division of Molecular and Cellular Biology, National Institute for Child Health and Human Development, National Institutes of Health, Bethesda, MD, USA. [4]RNA Therapeutics Institute, University of Massachusetts Chan Medical School, Worcester, MA, USA. [5]Institute for Medical Engineering and Science and Department of Physics, Massachusetts Institute of Technology, Cambridge, MA, USA. ✉e-mail: job.dekker@umassmed.edu

at specific histone modifications—that is, BRD proteins for histone 3 lysine 27 acetylation (H3K27ac)[11,12] and HP1 proteins for H3K9me3 (refs. 13,14)—can bridge interactions between marked loci. As gene expression is cell type-specific, changes in chromatin composition and histone modification along the linear genome and, in turn, the spatial positioning of these domains in subcompartments within the nucleus also differ between cell types.

Interphase chromosome folding is actively modulated by cohesin-dependent loop formation. The resulting interaction patterns are also cell type-specific due to the reliance of cohesin activity on the positioning of *cis*-regulatory elements (CREs). Cohesin can be loaded throughout the genome[15] but is more efficiently recruited at open chromatin sites such as enhancers[16–18]. Furthermore, loop extrusion is blocked in a directional manner at CTCF-bound sites[9,19–21] and cohesin complexes are unloaded near the 3′ ends of active genes[16,22]. These dynamics specify a genome-wide 'cohesin traffic pattern' that gives rise to a complex identity-specific chromosome folding pattern observed by the chromosome conformation capture-based technology Hi-C that includes loops (for example, CTCF–CTCF and promoter-enhancer loops), stripes, flares, insulated boundaries and topologically associating domains (TADs)[23].

During mitosis most features of cell type-specific chromosome folding, such as compartments and cohesin-mediated loops, are undetectable by Hi-C. A largely cell type-invariant compressed array of partially self-entangled loops is instead formed by condensin complexes to give rise to rod-shaped mitotic chromatids[24–28]. These loops are not positioned at specific CREs. Many of the chromatin-associated factors that contribute to interphase organization are absent from mitotic chromosomes. Condensins actively remove loop-extruding cohesin complexes during prophase of mitosis[26], and cell-cycle kinases drive the phosphorylation and dissociation of a large proportion of the chromatin-bound factors, including CTCF, transcription factors and, to a large extent, RNA polymerases[29–33]. However, not all regulatory factors dissociate from mitotic chromosomes and a number remain bound at specific *cis*-elements. This phenomenon, referred to as mitotic bookmarking[34], records active loci for reactivation in the next cell cycle. Cell cycle stage-dependent changes in chromosome structure are also observed at the scale of nucleosomes. Although promoters that are active in a given cell type remain largely nucleosome-free, enhancers are less accessible during mitosis and are probably bookmarked to be reopened by remodellers during the next cell cycle[29,35]. Similarly, CTCF sites are thought to be bookmarked by active histone modifications and become occupied by nucleosomes in mitosis, although the extent to which this occurs differs between cell types[29,36,37].

The dramatic changes to chromosome organization that occur during mitosis pose a particular challenge to cycling cells. Every time cells re-enter G1, the cell type-invariant mitotic chromosome structure must be converted to an identity-specific interphase conformation. The relative timing of events during this process has been delineated in several recent studies[38–42]. First, condensins dissociate during telophase and a transient chromosome folding state devoid of extruded loops is formed[39]. Although CTCF is rapidly enriched at post-mitotic chromatin, the cohesin complex is recruited later during cytokinesis[38,39], at which point extrusion rapidly re-establishes loops, TADs and other extrusion-related features. Genome compartmentalization is first detected around telophase but does not reach the full extent of segregation until well into G1 (ref. 39).

Although the kinetics of post-mitotic chromosome reorganization are increasingly known, it remains unclear how the cell type-specific folding programme is transmitted through mitosis. We have established an experimental system that allows us to discriminate the contributions of factors required for cell identity-specific chromosome folding based on whether they are inherited on mitotic chromosomes or transmitted to daughter nuclei through the cytoplasm. We find that the interphase chromatin state is achieved through the combined action of two folding programmes inherited in distinct ways.

## Results

### Experimental approach to insulate the genome from cytoplasmic factors during mitotic exit

During telophase, the nuclear envelope re-forms around newly segregated chromosomes, establishing regulated nucleocytoplasmic transport. We sought to prevent nuclear import from telophase onwards to understand the extent to which type-specific interphase chromosome folding can be re-established independently of G1 cytoplasmic factors. To this end, RanGAP1 and Nup93 were biallelically fused with auxin-inducible degron (AID) and NeonGreen fluorescent protein sequences in DLD-1 cells using CRISPR–Cas9 (Methods, Fig. 1a and Extended Data Fig. 1), enabling disruption of nuclear transport by two mechanisms. Nup93 is a scaffold nucleoporin essential for assembly and stability of nuclear pore complexes[43–45], and RanGAP1 critically regulates the RanGTP gradient required for nucleocytoplasmic shuttling of proteins[46,47].

To prevent nuclear transport during mitotic exit, RanGAP1–AID or NUP93–AID were degraded in arrested prometaphase cells (Supplementary Methods and Fig. 1a,b). The cells were then released through anaphase into G1. Depletion of RanGAP1–AID or Nup93–AID did not impact the kinetics of the initial mitotic exit stages (Fig. 1c,d and Extended Data Fig. 2a–c). The absence of Nup93 after mitotic release resulted in daughter nuclei sealed by a continuous nuclear envelope and an underlying nuclear lamina devoid of mature nuclear pore complexes (Extended Data Fig. 2d,e). Conversely, apparently mature nuclear pore complexes were found at the nuclear envelope of RanGAP1–AID-depleted cells (Fig. 1e and Extended Data Fig. 2e). Despite differences in nuclear morphology, depletion of either RanGAP1–AID or Nup93–AID impeded chromatin decondensation and nuclear volume expansion characteristic of control cells progressing through G1 (Fig. 1f and Extended Data Fig. 2d). Nucleoli and nuclear speckles, two import-dependent organelles with important roles in nuclear organization[48–51], were not found in RanGAP1–AID- or Nup93–AID-depleted nuclei and strong speckle protein-containing granules were found exclusively in the cytoplasm (Extended Data Fig. 2f).

The transport competence of Nup93- and RanGAP1-depleted cells for large macromolecules was quantified by expressing a canonical importin-α nuclear localization signal (NLS) fused to maltose-binding protein (MBP) and mScarlet fluorescent protein (MBP–mScarlet–NLS; Fig. 1g and Extended Data Fig. 2g). Time-lapse fluorescence imaging of control cells demonstrated efficient nuclear import of MBP–mScarlet–NLS following chromosome segregation, with minimal protein observed in the cytoplasm by 5 h. In contrast with the untreated controls, depletion of either Nup93–AID or RanGAP1–AID prevented the nuclear accumulation of MBP–mScarlet–NLS throughout mitotic exit and G1 entry entirely, consistent with a failure to establish nucleocytoplasmic transport. Thus, in this experimental context, the genome is isolated from the cytoplasm once the nuclear envelope has formed in late telophase.

### Post-mitotic chromosomes possess the intrinsic capacity to form a genome-scale microcompartment of candidate CREs

To assess how chromosome-intrinsic factors inform post-mitotic genome folding, we performed Hi-C 3.0 (refs. 52,53; Methods and Supplementary Table 1). Pure G1 populations of cells that had progressed through mitosis in the absence or presence of RanGAP1 or Nup93 were obtained by fluorescence-activated cell sorting (FACS) (Supplementary Fig. 1). Despite an inability to establish active nucleocytoplasmic transport, not only did broad euchromatin/heterochromatin compartments form but they also seemed more pronounced in RanGAP1–AID- and Nup93–AID-depleted cells (Fig. 2a, Extended Data

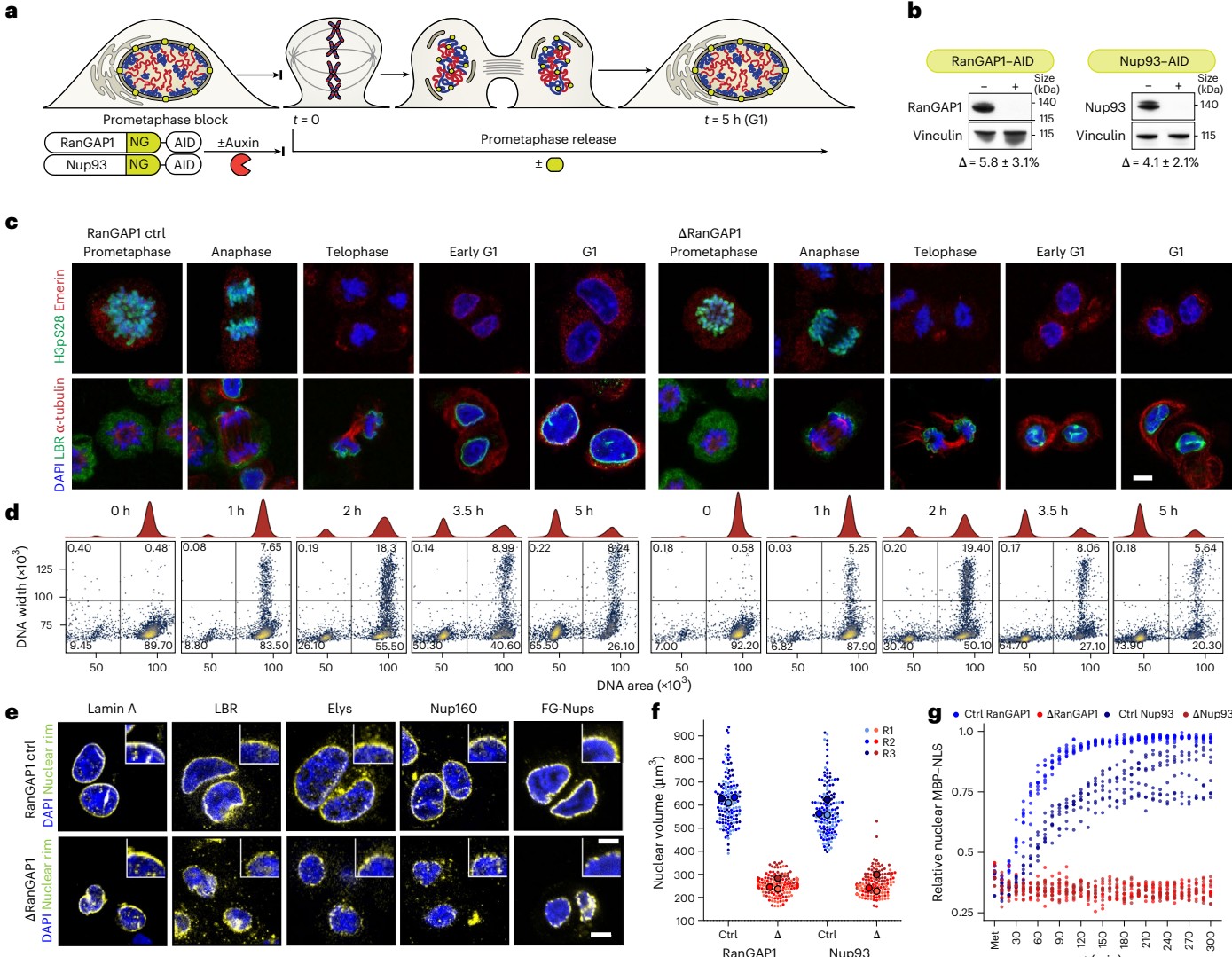

**Fig. 1 | Acute depletion of essential nuclear import machinery during mitotic exit enables the assembly of daughter nuclei isolated from the G1 cytoplasm.**
**a**, Workflow for the depletion of AID-tagged RanGAP1 and Nup93 proteins from the onset of mitotic exit into G1. Auxin-induced degradation is initiated 2 h before nocodazole/mitotic release (time ($t$) = 0). NG, NeonGreen. **b**, Representative western blot images of whole-cell lysates derived from control and auxin-treated DLD-1 cell lines 5 h after mitotic release showing efficient depletion of RanGAP1–AID and Nup93–AID (left). Mean protein levels (±s.d.) normalized to vinculin for depleted lysates relative to untreated controls for three replicates (right). **c**, Representative immunofluorescence images for RanGAP1 control and depleted cells demonstrate congruous kinetics of mitotic release. Loss of histone H3 serine 10 phosphorylation (H3pS10) and recruitment of the nuclear-envelope proteins emerin and lamin B receptor (LBR) are shown for control and RanGAP1–AID-depleted cells. **d**, Representative DNA-content (stained with PI) flow cytometry measurements for RanGAP1 control and depleted cells demonstrate congruous kinetics of mitotic release. The percentage of cells in each quadrant of the flow

cytometry plots is provided. **e**, Representative immunofluorescence images of RanGAP1–AID control and depleted (ΔRanGAP1) cells 5 h after mitotic release demonstrating the presence of nuclear lamina (lamin A), nuclear envelope (LBR) and nuclear pore complex (Elys, Nup160 and FG-rich nucleoporins (FG-Nups))-resident proteins in the absence of RanGAP1–AID. **f**, The nuclear volumes of control and RanGAP1- or Nup93-depleted cells 5 h after mitotic exit indicate a reduced size of import-incompetent nuclei. Individual data points and mean values for each of three independent replicates (R1, R2 and R3) are indicated. **g**, Import competence is not re-established after mitosis in RanGAP1–AID- or Nup93–AID-depleted cells. This is indicated by the reduced relative nuclear-to-total intensity of MBP–mScarlet–NLS after 5 h release from Ro-3306. Ratios are shown for eight cells in three independent fields from onset of mitotic release for one experiment. **c**,**e**, Scale bars, 5 μm (main images) and 1 μm (**e** (insets)). Ctrl, control; DAPI, 4′,6-diamidino-2-phenylindole. Source numerical data and unprocessed blots are provided.

Fig. 3a and Supplementary Fig. 2b,c). In the absence of nuclear import, we observed an additional prominent network of small (<100 kb) highly interacting chromatin domains that interacted ubiquitously with each other in *cis* and *trans*. The chromosome-wide grid-like nature of these interactions implies the formation of a distinct subcompartment in the absence of factors that would normally enter the nucleus after telophase. We refer to these interactions as microcompartments on the basis of their scale and similarity to previously described structures[8,54].

The properties of chromosome-intrinsic microcompartments were explored by identifying the domains of 25–125 kb that mediate strong focal pairwise enrichments in the Hi-C interaction data of RanGAP1–AID-depleted G1 cells. We identified 2,105 strong microcompartment domains (MCDs) along the linear genome encompassing all autosomal chromosomes[9,55] (Supplementary Fig. 3 and Methods). Microcompartment domains consist of gene-dense active regions corresponding to high assay for transposase-accessible chromatin using sequencing (ATAC–seq) coverage and peaks of

H3K4me3 and H3K27ac found in both control and RanGAP1-depleted cells in G1 (Fig. 2b,c and Supplementary Tables 2,3). Accordingly, active promoters and enhancers defined in control cells (candidate CREs, cCREs; Methods) were highly enriched at MCDs (Fig. 2d). Furthermore, the A-compartment profile at MCDs captured by the first Eigenvector (EV1) in control cells was focally enriched in RanGAP1–AID- and Nup93–AID-depleted cells (Fig. 2a,c and Extended Data Fig. 3a,b), indicating a more prominent subcompartment. We confirmed the compartment nature of these regions by genome-wide spectral clustering analysis of RanGAP1-depletion Hi-C data (Methods and Extended Data Fig. 4a,b)[15,56]. More than 60% of the 2,105 MCDs were specifically covered by a single cluster corresponding to highly active euchromatin (Extended Data Fig. 4c).

Based on the comprehensive grid of microcompartments observed in RanGAP1–AID-depleted cells, we leveraged the possibility that all MCDs possess an intrinsic affinity for each other. To quantify the strength of these interactions, we projected all pairwise combinations of MCDs, giving rise to 73,970 and 2,079,615 microcompartment interactions within and between chromosomes, respectively. Aggregate pileups demonstrate an average focal enrichment of interaction frequency of at least fourfold in RanGAP1–AID- and Nup93–AID-depleted cells extending to all genomic distances in *cis* and to *trans* chromosomal contacts (Fig. 2e and Extended Data Fig. 3c). Enriched interactions between the MCDs were also observed in control G1 cells but with substantially reduced frequency (Fig. 2e and Extended Data Fig. 3c), consistent with previous observations of weakly enriched interactions between active promoters and enhancers in wild-type cells[8,54,57]. The quantitatively lower enrichment of MCD contact frequency in Nup93–AID-depleted cells compared with RanGAP1 depletion may reflect clonal variation, as MCDs were assigned in the latter context, but we also noted marked differences in the short-range contact frequency heat maps of Nup93-depleted cells, including a more prominent maintenance of some control loops compared with the RanGAP1 depletion (Extended Data Fig. 3a).

In a complementary analysis we determined the interaction frequencies between cCREs defined in control RanGAP1–AID cells. Pairs of CTCF sites interacted frequently in control cells at genomic distances of <1 megabases (Mb), as expected, but were dramatically reduced in RanGAP1–AID- or Nup93–AID-depleted G1 cells (Fig. 2f and Extended Data Fig. 3d). Conversely, we found that active promoters and enhancers interacted with each other at considerably elevated frequencies regardless of genomic separation when nuclei formed in the absence of RanGAP1 or Nup93 (Fig. 2f and Extended Data Fig. 3d). These interactions also occurred in control cells but at lower frequencies. Together, we found that the propensity to form a distinct microcompartment of active CREs during G1 is an intrinsic capacity of post-mitotic chromosomes that emerges as a pronounced feature in the absence of active nuclear transport.

## A microcompartment of CREs is a transient folding intermediate during normal mitotic exit

Telophase is the last mitotic stage when chromosomes can recruit cytoplasmic factors independently of nuclear import. We therefore sought to determine the folding state of telophase chromosomes using Hi-C. Pro-metaphase-arrested control and RanGAP1–AID-depleted cells were released and fixed after 1.25 or 1.50 h, the point where we observed the greatest proportion of telophase cells in the post-mitotic population (Fig. 1d, upper right quadrant), and stained with propidium iodide (PI). To enrich telophase populations, the cells were sorted based on the intensity and width of the PI signal (Fig. 3a), which briefly peaks in the transition from metaphase to G1 (ref. 58; Fig. 1c). At 1.25 h post release, at least 60% of the sorted cells were in telophase, as evidenced by elongated mid-body microtubules connecting the two masses of highly condensed chromosomes (Extended Data Fig. 5a). Cells sorted in the same way at 1.5 h post release had progressed through telophase and we could identify more than half of the sorted cells having a clear abscission mid-body, indicating they were in cytokinesis. In our experimental system, nuclear volume increased by about 50% from telophase to cytokinesis in control cells and dramatically expanded by almost threefold in early G1 (Fig. 3b). In contrast, RanGAP1–AID-depleted nuclei did not noticeably expand in size beyond telophase.

We performed Hi-C with FACS-sorted populations of prometaphase, telophase, cytokinesis and G1 cells, and observed the progressive acquisition of interphase chromosome compartments and loops in control cells (Fig. 3c and Extended Data Fig. 5b). Remarkably, unperturbed cells were found to form MCD contacts that were visually apparent in contact frequency heat maps by cytokinesis. We compared the 2,105 MCDs identified in G1 to those identified in cytokinesis-sorted RanGAP1-depleted cells and found that more than half of all G1 MCDs were detected at the earlier time point (Extended Data Fig. 5c). In addition, a substantial number of microcompartment anchors were only called in cytokinesis or only in G1 but aggregate analysis demonstrated their interactions did indeed occur in *cis* and *trans* in both cell-cycle stages, albeit with different relative frequencies (Fig. 3d). Pairwise interactions between MCDs identified as cytokinesis-specific, G1-specific or shared cytokinesis-G1 were formed in *cis* and *trans* during telophase and increased in strength towards cytokinesis in both control and RanGAP1–AID-depleted cells (Fig. 3d). In the absence of nuclear import, these active microcompartments continued to strengthen as cells entered G1. By contrast, progression into G1 in control cells coincided with a strong reduction in the frequency of pairwise MCD interactions. Together, these data demonstrate that extensive *cis* and *trans* microcompartmentalization of active cCREs is a normal transient process that peaks during cytokinesis and is then lost or at least severely reduced during G1. Similar short-range microcompartment dynamics have been seen during mitotic exit in mouse cells[59]. This loss was not observed in RanGAP1–AID- or Nup93–AID-depleted cells and MCD interactions continued to increase in frequency as cells progressed through G1.

**Fig. 2 | Depletion of RanGAP1 during mitotic exit reveals a chromosome-intrinsic capacity to form a kilobase scale cCRE compartment. a**, Hi-C interaction frequency maps at 250 kb (chromosome (Chr) 6: 105.0–170.8 Mb–Chr 7: 0–56 Mb; left), 50 kb (Chr 6: 125–145.5 Mb; middle) and 10 kb (Chr 6: 129–130.5 and 136.1–138.2 Mb; right) resolutions showing genome compartmentalization in control (top) and auxin-treated (bottom) RanGAP1–AID cells released from prometaphase for 5 h. First Eigenvector values for *cis* interactions phased by gene density (high gene density > 0). **b**, Hi-C contact matrix at 10 kb resolution (Chr 6: 129–130.5 and 136.1–138.2 Mb) highlighting microcompartments detected in RanGAP1-depletion and constituent MCDs (top). Control gene annotations, and ATAC–seq as well as histone H3K4me3 and H3K27ac CUT&RUN of control and RanGAP1-depleted cells (middle). Control tracks define CREs, which coincide with MCDs. **c**, Heat maps for all 2,105 MCDs centred on contact frequency summits with a 100-kb flank and sorted by length, demonstrating the prevalence of the ATAC–seq, H3K27ac, H3K4me3 and H3K27me3 coverage in control and

RanGAP1-depleted cells, and RNA expression in control cells. Intra-chromosomal EV1 values from 10-kb matrices indicate altered compartmentalization at MCDs following RanGAP1 depletion. **d**, Relative enrichment of control candidate CREs genome-wide, at 10 kb control A-compartment bins (EV1 > 0), and MCDs demonstrating the predominance of active promoters (PLS) and enhancers (proximal and near enhancer-like (pELS), promoter–proximal; dELS, promoter–distal). **e**, Pairwise mean observed/expected contact frequency between all MCDs (top) projected in *cis* (10 kb) and *trans* (25 kb), showing dramatically enhanced interactions at all length scales, and between chromosomes in RanGAP1-depleted G1 cells (bottom). **f**, Aggregate pairwise observed/expected contact frequencies between cCREs assigned in control cells and subjected to hierarchical binning at 10 kb resolution showing enhanced homo- and heterotypic interactions between active promoters and enhancers in RanGAP1-depleted cells at multiple genomic distances in *cis* and in *trans*. Source numerical data are provided.

The MCDs detected during cytokinesis were relatively enriched for marks of active CREs compared with those detected later in G1 (Fig. 3e and Extended Data Fig. 5d). Although cytokinesis-specific MCDs displayed slightly different interaction dynamics during the time course, they were similarly enriched for active CREs. Accordingly, contacts between active promoters and enhancers mirrored the interactions between MCDs during mitotic exit (Extended Data Fig. 5e). The average contact frequency between CREs increased in

telophase, reaching a peak at all length scales during cytokinesis, and was relatively diminished in a distance-dependent way in control cells by early G1.

## Microcompartments enriched for mitotically bookmarked CREs

As mitotic exit coincides with transcriptional reactivation[60,61], we sought to examine the relationship between transcriptional activity and microcompartment formation in control and RanGAP1-depleted

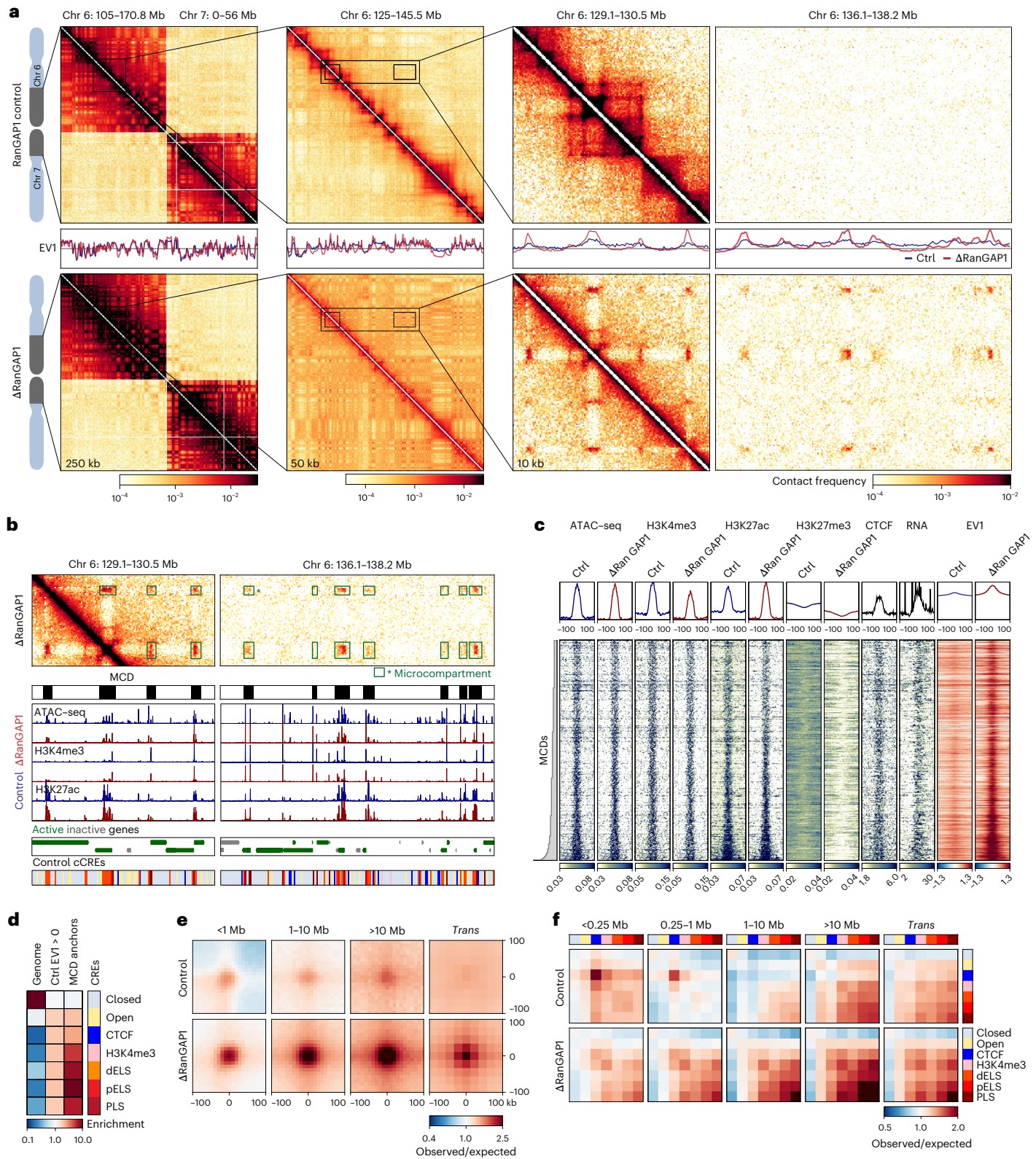

cells. We used Thiol(SH)-linked alkylation for the metabolic sequencing of RNA (SLAM–seq)[62,63] to directly quantify nascent RNA molecules synthesized in 1-h increments from prometaphase to early G1 (Fig. 4a, Extended Data Fig. 6a,b and Methods). We found that the new-to-total RNA ratios (NTRs) increased relative to prometaphase levels from 2 to 5 h in control cells but globally plateaued after the initial minor increase of nascent synthesis, 2 h following prometaphase release, in RanGAP1-depleted cells (Fig. 4b,c and Extended Data Fig. 6b,c).

Three clusters of genes were defined by the time at which post-mitotic nascent RNA synthesis first exceeded an NTR of 10%: prometaphase ($t = 0$), early ($t = 1–2$ h), and late ($t = 3–5$ h) genes. In control cells, nascent RNA synthesis increased over time through waves of gene activation with earlier expressed genes generally reaching higher relative levels of new transcription by G1 (Fig. 4d (left)). In RanGAP1-depleted cells, these genes displayed reduced transcriptional activation during mitotic exit and generally reduced nascent synthesis (Extended Data Fig. 6d). The relatively few genes with substantial levels of nascent synthesis in RanGAP1-depleted cells were predominantly early expressed genes that were transcribed at reduced levels by G1 (Fig. 4d (right)), consistent with a minimal capacity for new transcription during the early stages of mitotic exit that is severely attenuated by G1. Prometaphase and early expressed genes were significantly enriched at MCDs, particularly those detected in cytokinesis (Fig. 4e). However, this association was found for early expressed control genes but not the RanGAP1-depleted set (Fig. 4e), suggesting a greater correspondence of MCDs to the control transcriptional programme inherited from the previous cell cycle. The global reduction in nascent RNA synthesis in RanGAP1-depleted cells indicates that ongoing transcription is not required to maintain or strengthen these interactions in G1. However, the enrichment of early expressed genes at strong MCDs, implies a role for the microcompartmentalization of active elements in driving post-mitotic gene activation in control cells.

The appearance of a microcompartment during telophase that overlaps the specific active CREs of wild-type cells implies an inherited feature. To investigate this possibility, we performed Omni-ATAC–seq[64] (Methods) and cleavage under targets and release using nuclease (CUT&RUN)[65] targeting H3K4me3, H3K27ac or CTCF for cells arrested in prometaphase or released to early G1 (Fig. 5a). Consistent with previous work[29], the ATAC–seq fragment-length distributions indicated more regularly spaced nucleosomes during mitosis compared with G1 cells, which was not impacted by the depletion of RanGAP1 (Extended Data Fig. 7a). We identified 159,072 peaks of ATAC–seq coverage across all conditions and compared their intersection (Fig. 5b). Although the largest fraction of peaks found genome-wide were G1-specific (left), accessible regions at MCDs were predominantly mitotically retained and RanGAP1 independent (right). These peaks correspond to highly accessible H3K4me3- and H3K27ac-dense loci (Fig. 5a,c), consistent with mitotic bookmarking at CREs. More than 90% of the MCDs overlapped at least one bookmarked promoter- or enhancer-annotated

ATAC–seq peak and the valency of these transcriptional elements was increased at the stronger domains that were detected in cytokinesis (Fig. 5d and Extended Data Fig. 7b). A small subset of G1-specific peaks that became accessible in the import-deficient G1 cells were also enriched at MCDs. However, more than half of the MCDs did not overlap any promoter or enhancer specific to these G1-specific peaks, suggesting that the G1-acquired open sites are not essential for MCD interactions (Extended Data Fig. 7c). Together, these data suggest that the microcompartments formed during telophase are driven by mitotically inherited regions of robust chromatin accessibility.

Histone H3K27 acetylation is implicated in both post-mitotic gene activation[40,41,61,66] and 3D genome organization[8,57]. We observed that bookmarked peaks of enriched H3K27ac not only corresponded to MCDs but could also recapitulate microcompartment dynamics during mitotic exit, forming enriched contacts in *cis* and *trans* that increased from telophase to cytokinesis and were attenuated as cells entered G1 in the presence of RanGAP1 (Extended Data Fig. 7d,e). These observations are consistent with a capacity for chromatin readers, such as Bromo and Extra-Terminal (BET) domain proteins, to bind acetylated histones during mitosis[12,66,67] and form phase-separated condensates[11,68–70]. To determine whether they mediate microcompartment formation during mitotic exit, BET domain proteins were targeted using the competitive inhibitor JQ1 or dBET6 (a proteolysis targeting chimaera, PROTAC), which degrades BRD2 and BRD4 in prometaphase-arrested cells (Fig. 5e). Surprisingly, inhibition or loss of BET domain proteins did not attenuate pairwise interaction frequencies between MCDs or bookmarked H3K27ac peaks in control or RanGAP1-depleted cells during mitotic exit (Fig. 5f and Extended Data Fig. 7f), indicating that another factor or H3K27ac itself must drive the transient clustering of MCDs.

## Nuclear transport-dependent pruning of long-range indiscriminate microcompartment interactions

The condensin-mediated loop array that compacts mitotic chromosomes is replaced by the extrusive activity of cohesin around telophase[38,39,42], giving rise to CTCF loops, TADs and insulating domain boundaries. Average Hi-C contact probability plotted as a function of genomic separation, $P(s)$, provides an estimation of extruded loop size at the distance of minimal decay[25,71–74]. In control cells, condensin loops of 200–300 kb disappeared at telophase and were undetectable by cytokinesis. Following G1 entry, loops of approximately 100 kb were observed, consistent with cohesin-mediated extrusion (Fig. 6a). In the absence of RanGAP1, the condensin loops were lost with similar kinetics to controls but 100-kb loops were not established and the $P(s)$ curve indicates the absence of any extruded loops from cytokinesis onwards. Accordingly, loops at convergent CTCF motifs first appeared in telophase and strengthened by at least fivefold in early G1 control cells, but these interactions remained rare throughout mitotic exit in the absence of RanGAP1 (Fig. 6b–d). Depletion of Nup93–AID

**Fig. 3 | A transient microcompartment of chromosome-intrinsic affinities is first formed during telophase. a**, Enrichment of telophase and cytokinesis cells by FACS for Hi-C analysis (left). Fixed cells were collected 1.25–1.50 h following prometaphase release and sorted based on DNA (PI) content and width (top right). The percentage of cells in each quadrant of the flow cytometry plots is provided. Immunofluorescence images showing enrichment of cells in telophase or cytokinesis based on chromatin (DAPI) and microtubule (α-tubulin) morphology (bottom right). Scale bar, 5 μm **b**, The nuclear volume of control and RanGAP1-depleted cells enriched in telophase, cytokinesis or G1 indicates a lack of nuclear growth after telophase in import-incompetent nuclei. Data for 25 DAPI-stained nuclei per condition are provided along with the mean of each of the two replicates (R1 and R2). **c**, Hi-C interaction frequency maps (Chr 14: 53.5–55.5 versus 67.8–69.0 Mb) at 10 kb resolution showing genome organization in control and auxin-treated RanGAP1–AID cells in prometaphase, telophase, cytokinesis or early G1. EV1 values for *cis* interactions phased by gene density

(high gene density > 0). MCDs detected in RanGAP1-depleted cytokinesis- or G1-sorted cell populations are shown. **d**, Pairwise mean observed/expected contact frequency between MCD anchors in *cis* (10 kb) and *trans* (25 kb) showing enhanced interactions in telophase that peak at cytokinesis in control cells. 2,105 G1 and 1,791 cytokinesis MCD anchors are categorized as cytokinesis-specific, shared G1-cytokinesis or G1-specific in RanGAP1-depleted cells. **e**, Heat maps for subsetted MCD anchors centred on contact frequency summits with a 100-kb flank and sorted by MCD length demonstrate the prevalence of H3K4me3 and H3K27ac coverage in mitotic, control G1 and RanGAP1-depleted G1 (ΔG1) cells at MCD anchors (bottom). The *cis* 10 kb EV1 values for control and ΔG1 cells, control CTCF coverage and convergent CTCF loop anchor assignment of control G1 cells are shown. Average profiles of the corresponding features, with orange representing cytokinesis-specific, green representing shared G1-cytokinesis and blue representing G1-specific MCDs (top). Cyto, cytokinesis; pro-Meta, prometaphase; Telo, telophase. Source numerical data are provided.

similarly prevented the global establishment of strong interphase loops (Extended Data Fig. 8a,b), suggesting that nuclear import is a general requirement for cohesin-mediated loop extrusion during G1 entry. Immunofluorescence confirms exclusion of the cohesin subunit Rad21 from post-mitotic nuclei assembled in the absence of RanGAP1 or Nup93, whereas CTCF retained access to chromatin in the nucleus (Fig. 6c and Extended Data Fig. 8c). Accumulation of CTCF on the chromatin provides support for previous findings suggesting the rapid recruitment of CTCF following mitotic exit precedes the cohesin-mediated extrusion of interphase loops[38].

Loop extrusion counteracts epigenetically defined chromatin compartmentalization[72,74–76] and we observed a decrease in MCD

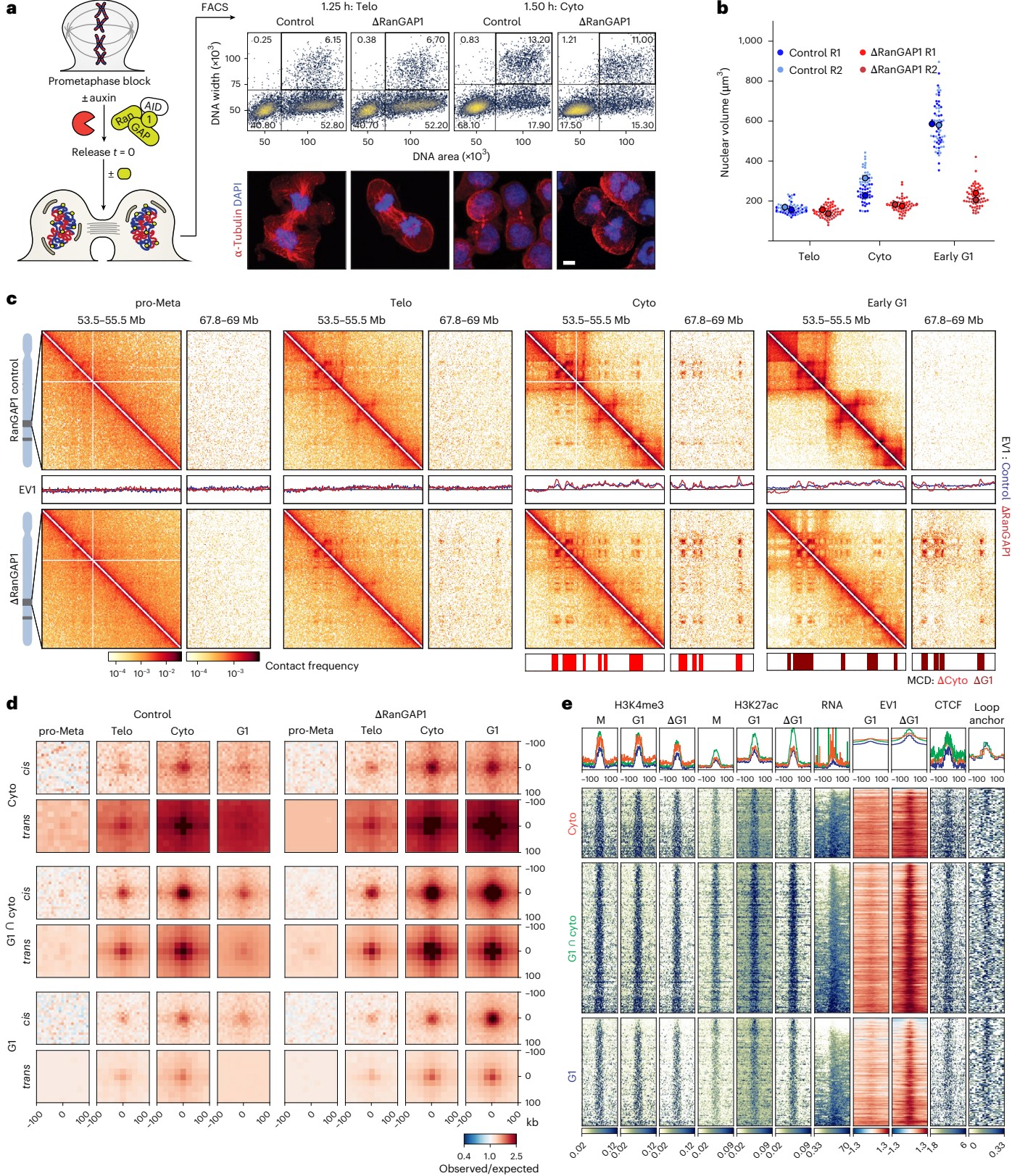

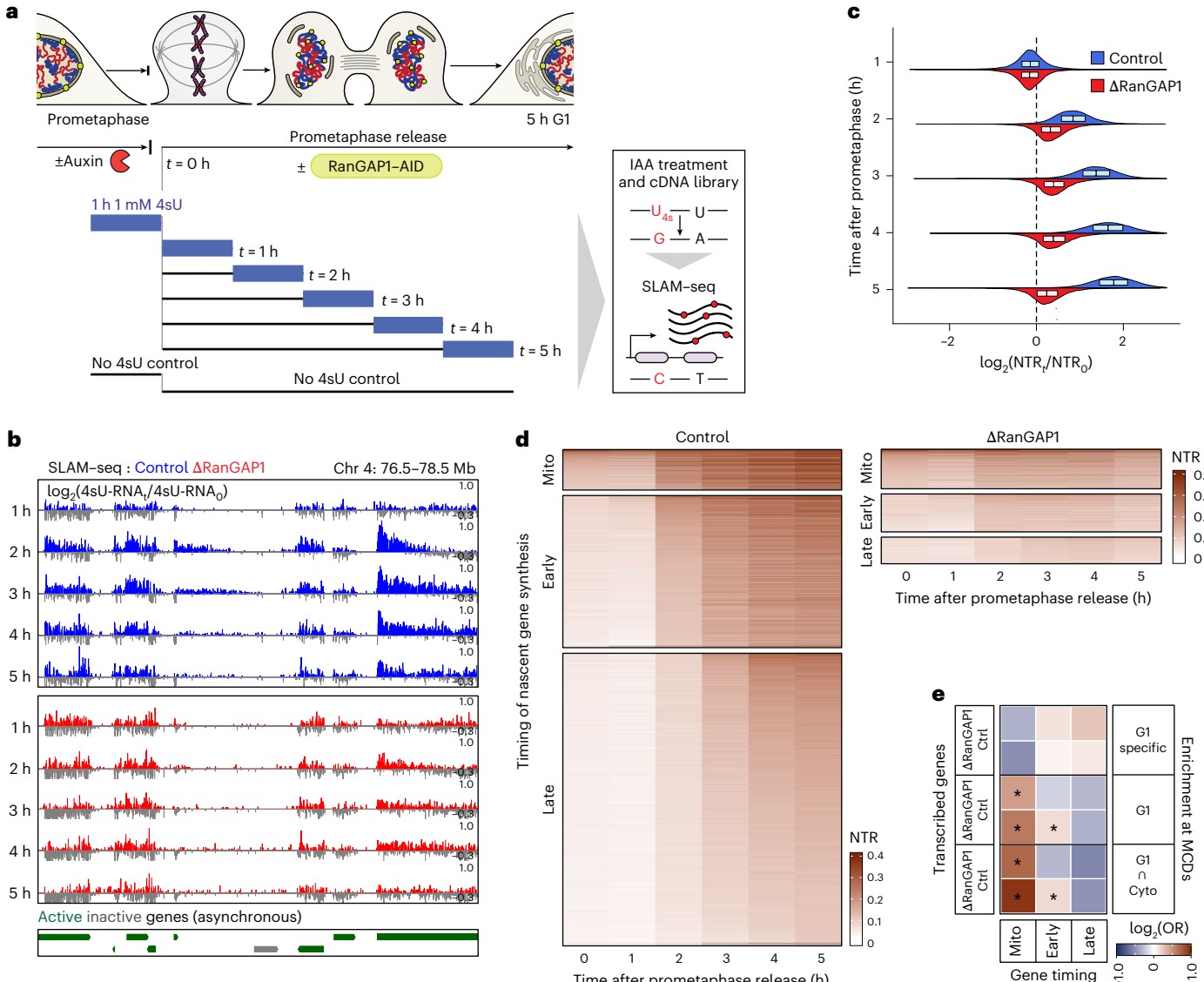

**Fig. 4 | The early post-mitotic transcriptional programme corresponds to MCDs but not microcompartment formation. a**, Workflow for pulse labelling of nascent transcripts during prometaphase or mitotic release in the presence or absence of RanGAP1. Nascent transcripts were labelled by the addition of 4-thio-uridine (4sU) for 1 h before RNA extraction at prometaphase ($t$ = 0) and 1, 2, 3, 4 or 5 h after release. The addition of auxin leads to thiol-specific alkylation and enables site-specific substitution of a guanine. SLAM–seq identifies T-to-C conversions by high-throughput sequencing and enables the quantification of nascent and previously transcribed RNA pools. **b**, Coverage tracks of 4SU-converted reads in control and RanGAP1-depleted cells collected hourly following mitotic release normalized to the prometaphase ($t$ = 0)-arrested signal demonstrating attenuated post-mitotic transcription in RanGAP1-depleted cells. **c**, Global distribution of the mean of the NTR ratio relative to time 0 (prometaphase-arrested) of control and RanGAP1-depleted samples. Substantial transcriptional activity starting 2 h after mitotic release that continues to

increase into G1 characterizes genes in control samples but is attenuated in the absence of RanGAP1. Boxplots indicate the median and interquartile range (box bounds) of the NTR fold changes for three independent SLAM–seq libraries per time point. **d**, Heat maps of the mean NTRs for genes classified by the temporal stage at which NTR ≥ 10% ($n$ = 9,486 control (left) and 2,872 RanGAP1-depleted (right) genes): prometaphase, 0 h (mito); early, 1–2 h and late, 3–5 h. Transcriptional clusters identified in control samples contrast the attenuated transcriptional activity for clusters identified in RanGAP1-depleted samples. **e**, Odds ratios (ORs) showing the strength of association between genes at MCDs and post-mitotic gene reactivation clusters. Genes expressed by the early samples are enriched in control but not RanGAP1-depleted conditions at strong G1-cytokinesis-intersected MCDs (bottom). *$P$ < 0.1; one-sided Fisher's exact test, followed by Benjamini–Hochberg correction. Source numerical data are provided.

interaction strength in control cells as the strength of extruded loops increased in G1 (Fig. 6d,e). The reduced average MCD–MCD interaction frequency was driven by long-range interactions >750 kb, beyond the range of most extruded loops (Fig. 6f). To address the relationship between extrusion and the loss of telophase microcompartments as cells enter G1, MCDs were categorized by the presence of a CTCF loop anchor. At 10 kb resolution, extrusion anchors were found at more than 75% of the MCDs (Fig. 6g). The MCD interactions that overlapped

at least one CTCF loop anchor were only maintained over shorter distances (<750 kb) from cytokinesis to G1, whereas extrusion-free MCD contacts lacked a distance-dependent loss in contact frequency. These findings imply a role for loop extrusion in modifying the global network of pairwise MCD interactions as cells exit mitosis through a process we refer to as 'pruning'. Microcompartments that appeared in cytokinesis and coincided with interphase CTCF loops were not only retained but strengthened in control G1 cells (Fig. 6h–j), suggesting that

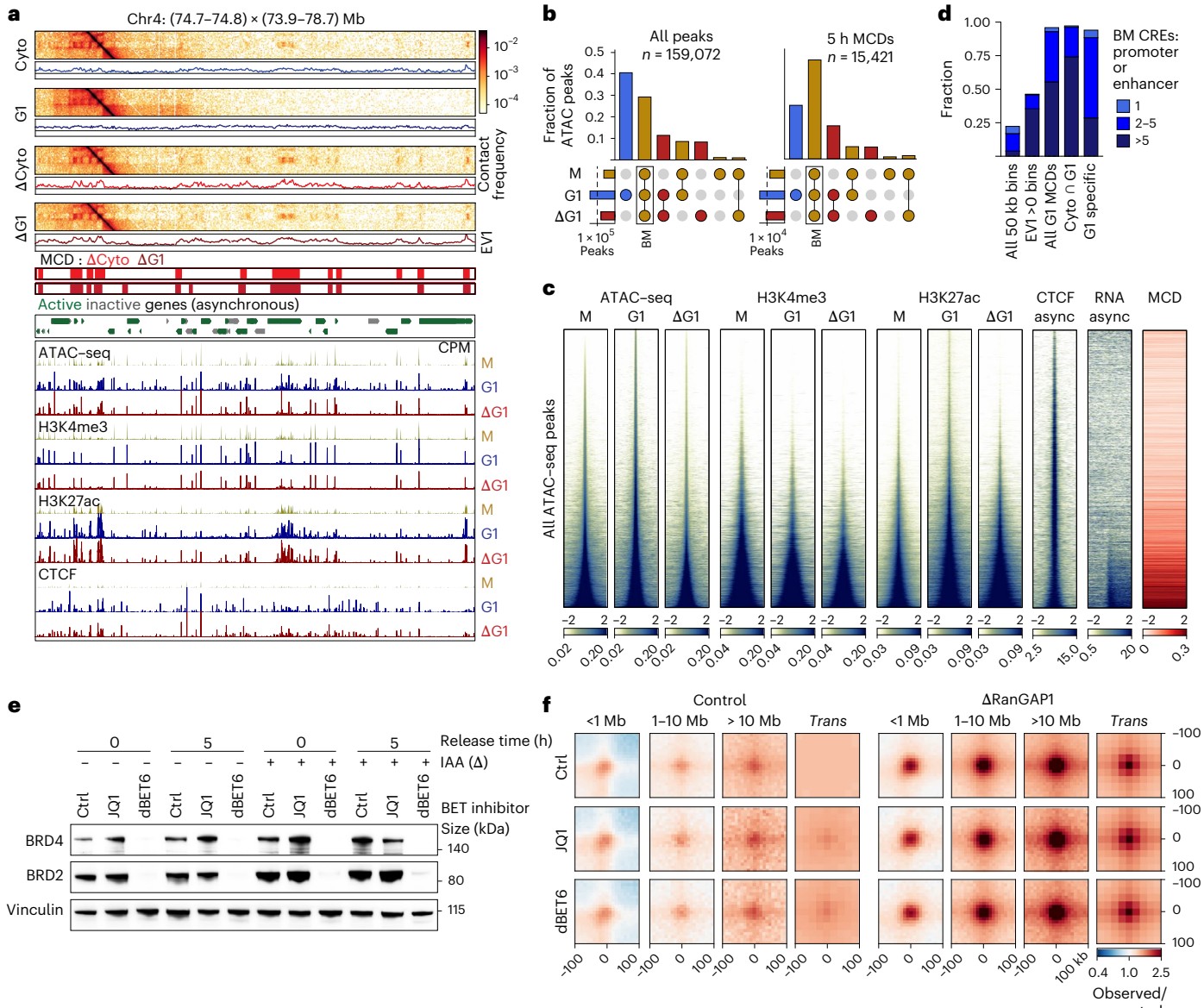

**Fig. 5 | Microcompartment domains are bookmarked during mitosis.**
**a**, Representative Hi-C contact matrices at 10 kb resolution (Chr 4: 74.7–
74.8 × 73.9–78.7 Mb) highlighting microcompartment interactions for MCDs
detected in cytokinesis and G1 RanGAP1-depleted cells. Coverage tracks from
ATAC−seq as well as CTCF, H3K4me3 and H3K27ac CUT&RUN generated in
prometaphase-arrested, G1 control and RanGAP1-depleted G1 cells indicate
the prevalence of mitotic booking at active genomic elements. Gene
annotations are shown for bulk RNA sequencing. CPM, counts per million.
**b**, Intersections of all (left) or MCD-overlapped (right) ATAC−seq peaks
detected in prometaphase, G1 control and RanGAP1-depleted G1 cells plotted
as a fraction of the total. The number of called peaks are indicated. BM,
bookmarked peaks. **c**, Heat maps centred on the union set of ATAC−seq peaks
detected in prometaphase-arrested or G1 cells released for 5 h with or without
RanGAP1. Stacks sorted by prometaphase ATAC−seq signal demonstrate
co-occurrence of G1 signals at mitotic peaks and the prevalence of H3K27ac,
H3K4me3, RNA sequencing and MCD anchors. Async, asynchronized.

**d**, Prevalence of bookmarked CREs assigned on control prometaphase and
G1 ATAC−seq, H3K4me3 and H3K27ac coverage. Fractions of 50 kb bins
genome-wide or having EV1 > 0 as well as all G1 shared G1-cytokinesis (cyto)
or G1-specific MCDs overlapping the indicated valency of promoter and
enhancer elements are shown. **e**, Representative western blot images of
whole-cell lysates derived from control and RanGAP1-depleted cells arrested
in prometaphase or 5 h after mitotic release, followed by treatment with either
JQ1 or dBET6 indicate efficient degradation of both BRD2 and BRD4 in the
dBET6-treated cells but not JQ1-treated cells. Vinculin was used as a loading
control. **f**, Pairwise mean observed/expected contact frequency between MCD
anchors in control and RanGAP1-depleted cells demonstrating unchanged cell
cycle-dependent interaction strength following treatment with either JQ1 or
dBET6, used to inhibit or degrade BET proteins, respectively. G1, G1 control;
ΔG1, RanGAP1-depleted G1; M, prometaphase-arrested. Source numerical data
and unprocessed blots are provided.

although the initial MCD–MCD interactions are cohesin-independent,
cohesin-mediated loop extrusion specifically reinforces these con-
tacts. The fact that cohesin loops are typically less than 1 Mb can explain
why only relatively short-range microcompartments are maintained in
this manner, whereas longer-range contacts are pruned in G1 (Fig. 6f).
As these MCDs become tethered to a small set of nearby MCDs to give

rise to interphase extrusion loops, interactions with other more distal
MCDs become far less probable.

We further classified microcompartment interactions based
on the domain identity of the constituent MCDs. We found the most
reduced MCD–MCD interactions in control G1 cells overlap a loop
anchor and span distinct looping domains (inter-TAD). Conversely,

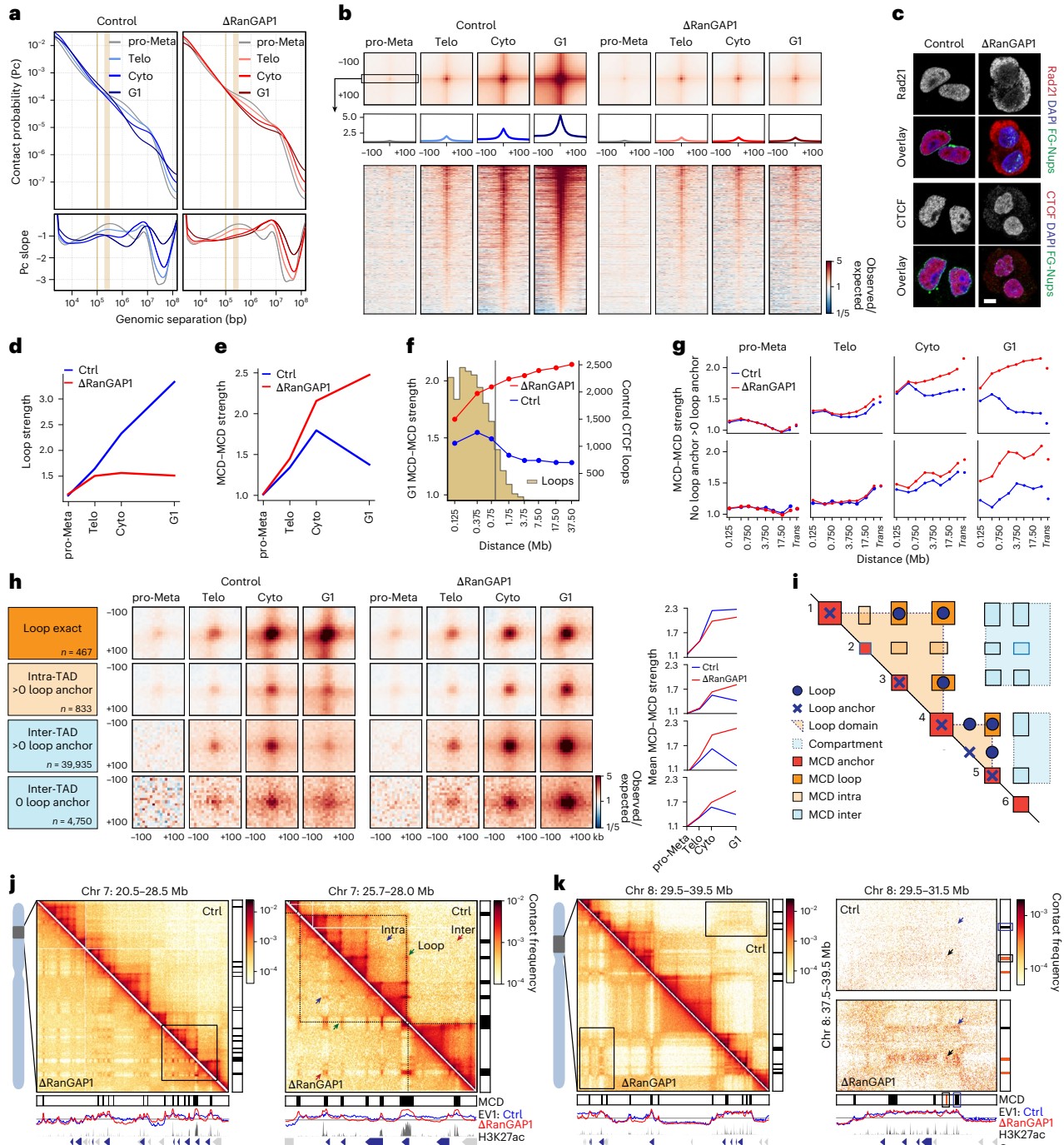

**Fig. 6 | Microcompartment interactions are pruned in nuclear transport-competent cells by early G1. a**, *P*(*s*) plots for Hi-C data from control and RanGAP1-depleted FACS-sorted cells at discrete phases. Brown shaded lines indicate the average size of cohesin- or condensin-extruded loops, 100kb or 200–300 kb loops, respectively. **b**, Mean observed/expected Hi-C contact frequency at 18,613 convergent CTCF loops demonstrating RanGAP1-dependent progression during mitotic exit. Interactions averaged over three 10 kb bins across the 200 kb CTCF motif-centred window and stack-ups sorted by G1 loop strength are shown. **c**, Representative immunofluorescence images of RanGAP1–AID control and depleted cells in G1 demonstrating nucleocytoplasmic localization of Rad21 and CTCF. Scale bar, 5 μm. **d,e**, Mean convergent CTCF loop strength (**d**) and mean strength of *cis* MCD–MCD contacts (**e**) of RanGAP1–AID control and depleted cells during mitotic exit. **f**, Mean strength of *cis* MCD–MCD contacts in G1 demonstrating the distance-dependent reduction in interaction strength beyond the range of control convergent CTCF loops in control cells. The 90th percentile of loop length is indicated by a grey line **g**, Mean strength

over time of MCD–MCD contacts classified by the presence (> 0) or absence of at least one CTCF loop anchor. **h**, Mean observed/expected MCD–MCD contact frequency at 10 kb during mitotic exit categorized by the presence of a convergent CTCF loop or loop anchor (>0) and looping domain status. **i**, Schematic of the microcompartment fates in G1 cells. **j**, Representative Hi-C interaction frequency maps at 25 kb (Chr 7: 20.5–28.5 Mb) resolution and a 10 kb zoom-in (Chr 7: 25.7–28.0 Mb) showing genome organization between looping domains in control (top) and RanGAP1–AID-depleted G1 cells (bottom). **k**, Representative Hi-C interaction frequency maps at 25 kb (Chr 8: 29.5–39.5 Mb) and 10 kb (Chr 8: 29.5–31.4 Mb versus 36.5–39.8 Mb) resolutions showing genome organization at loop-anchored and extrusion-free MCDs in control and RanGAP1–AID-depleted G1 cells. **j,k**, MCD positions, control cell EV1 values, H3K27ac CUT&RUN signal and gene annotations are shown. Colored arrows indicate fates of different microcompartments in control and RanGAP1–AID-depleted G1 cells. Ctrl, control; Cyto, cytokinesis; pro-Meta, prometaphase; Telo, telophase. Source numerical data are provided.

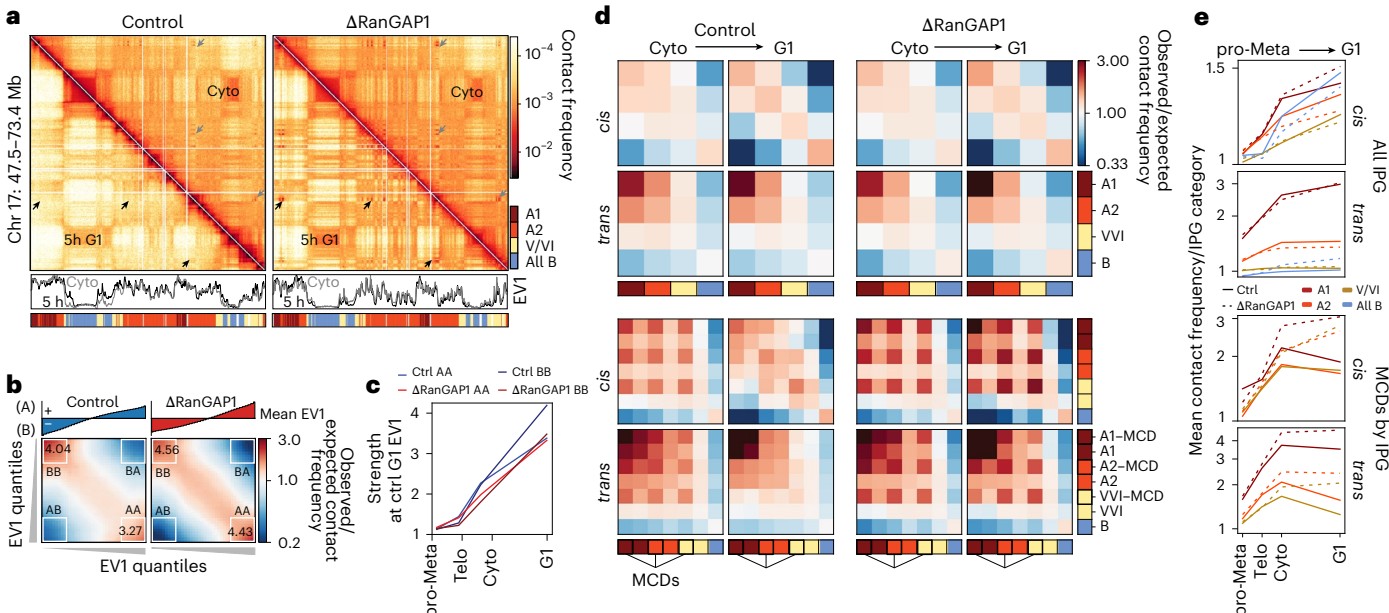

**Fig. 7 | The post-mitotic microcompartment is resorbed by multiple distinct G1 subcompartments. a**, Representative Hi-C interaction frequency maps at 50 kb resolution (Chr 17: 47.5–73.4 Mb) showing changes in genome organization from cytokinesis (upper triangle) to G1 ($t$ = 5 h; lower triangle). EV1 values and IPGs are shown. Arrows indicate microcompartments observed in control cytokinesis-sorted cells (grey), which are resorbed into subcompartments by early G1 (black). **b**, Saddle plots representing the segregation of active (A) and inactive (B) chromatin compartments in *cis* for control and RanGAP1-depleted cells 5 h after mitotic release. The EV1 from each condition was used to rank 25 kb genomic bins into equal quantiles and the average interaction frequency between these ranked bins was normalized to the expected interactions to build the heat map. Average preferential A–A and B–B interactions for the strongest 20% A and B loci are indicated (white squares). **c**, Average preferential A–A and B–B interaction strength in control

and RanGAP1–AID-depleted cells demonstrating increased A and B segregation over the course of mitotic exit. Ranked bins designating A and B compartments were derived from the EV1 of the 5 h control G1 cells at 25 kb resolution for comparison. **d**, Pairwise aggregate intra (*cis*) and interchromosomal (*trans*) observed/expected contact frequency between DLD-1 IPGs showing enhanced homotypic interactions in control and RanGAP1–AID-depleted cells from cytokinesis to early G1. Aggregate observed/expected interactions between MCDs as a subset of each IPG demonstrate the cell-cycle dynamics of a distinct microcompartment in the presence or absence of RanGAP1 (bottom). **e**, Quantification of the aggregate saddle plots in **d**. The mean observed/expected homotypic interaction frequency is shown for all IPGs (top) and IPGs stratified according to the presence or absence of an MCD. Lines are coloured as per the subcompartments in **d**. Source numerical data are provided.

intra-TAD microcompartments and inter-TAD contacts that did not overlap any CTCF loop anchors were reduced to a much lesser extent as control cells exited mitosis (Fig. 6h–j). Notably, the differences in MCD–MCD interactions with respect to loop extrusion anchors observed 5 h after mitotic release in RanGAP1–AID cells could be recapitulated in Nup93–AID cells, confirming generality in DLD-1 (Extended Data Fig. 8a–d). Together, we find that the extensive indiscriminate cCRE microcompartment formed during mitotic exit is pruned following the establishment of active nucleocytoplasmic transport. We propose that cohesin-mediated loop extrusion, and the resulting TADs and CTCF–CTCF loops, impose constraints on which pairwise MCD–MCD interactions remain and which microcompartments melt into larger subcompartments (Fig. 6k).

**Interphase nuclei retain the capacity to form a cCRE microcompartment after establishment of the pruned G1 state**

To evaluate whether the pruned state can be reversed, we performed Hi-C in cells depleted of RanGAP1 specifically after mitosis by the addition of auxin 3.5 h after prometaphase release (Extended Data Fig. 9a,b). At 3.5 h, loops had formed and MCD interactions were mostly pruned in control cells, whereas RanGAP1-depleted cells lacked extrusion loops and possessed extensive microcompartment interactions (Extended Data Fig. 9c). RanGAP1 loss after early G1 leads to a modest gain of long-range contacts between MCDs that are infrequent in control cells and indicates the capacity to re-form MCD contacts after pruning has occurred. This gain in MCD interactions coincided with reduced nuclear localization of cohesin (Extended Data Fig. 9d),

suggesting that continuous active nuclear transport is required to maintain nuclear cohesin levels and the fully pruned microcompartment. The relatively minor impact of G1 RanGAP1 depletion could be explained by retained levels of nuclear cohesin. Degradation of RanGAP1 during G1 most closely resembled the early G1 (3.5 h) control condition (Supplementary Fig. 2c,d). This state is characterized by an increase in the size of cohesin-extruded loops (Extended Data Fig. 9e, vertical arrows) and the presence of both MCD–MCD interactions and convergent CTCF loops (Extended Data Fig. 9f,g). Nonetheless, loss of nuclear transport after the occurrence of G1 pruning facilitated an additional increase in average MCD interactions at all genomic distances particularly across looping domains (Extended Data Fig. 9h), indicating a requirement for continuous active nuclear transport to maintain pruning of non-specific MCD interactions and a retained capacity to form these long-range contacts in interphase.

**The cCRE microcompartment is resorbed by multiple distinct active G1 subcompartments**

Using EV1 from Eigenvector decomposition of the Hi-C data to define A and B compartments revealed the expected acquisition of homotypic compartmental segregation after mitotic exit in both control and RanGAP1-depleted cells, which increased in strength through telophase, cytokinesis and G1 (Fig. 7a–c). The absence of active nuclear transport as cells progressed to G1 quantitatively impacted active and inactive chromatin compartmentalization (Supplementary Fig. 2c), leading to finer-scale active domains captured by EV1 (Fig. 7a and Extended Data Fig. 10a,b) and enhanced A/B segregation (Fig. 7b and Extended Data

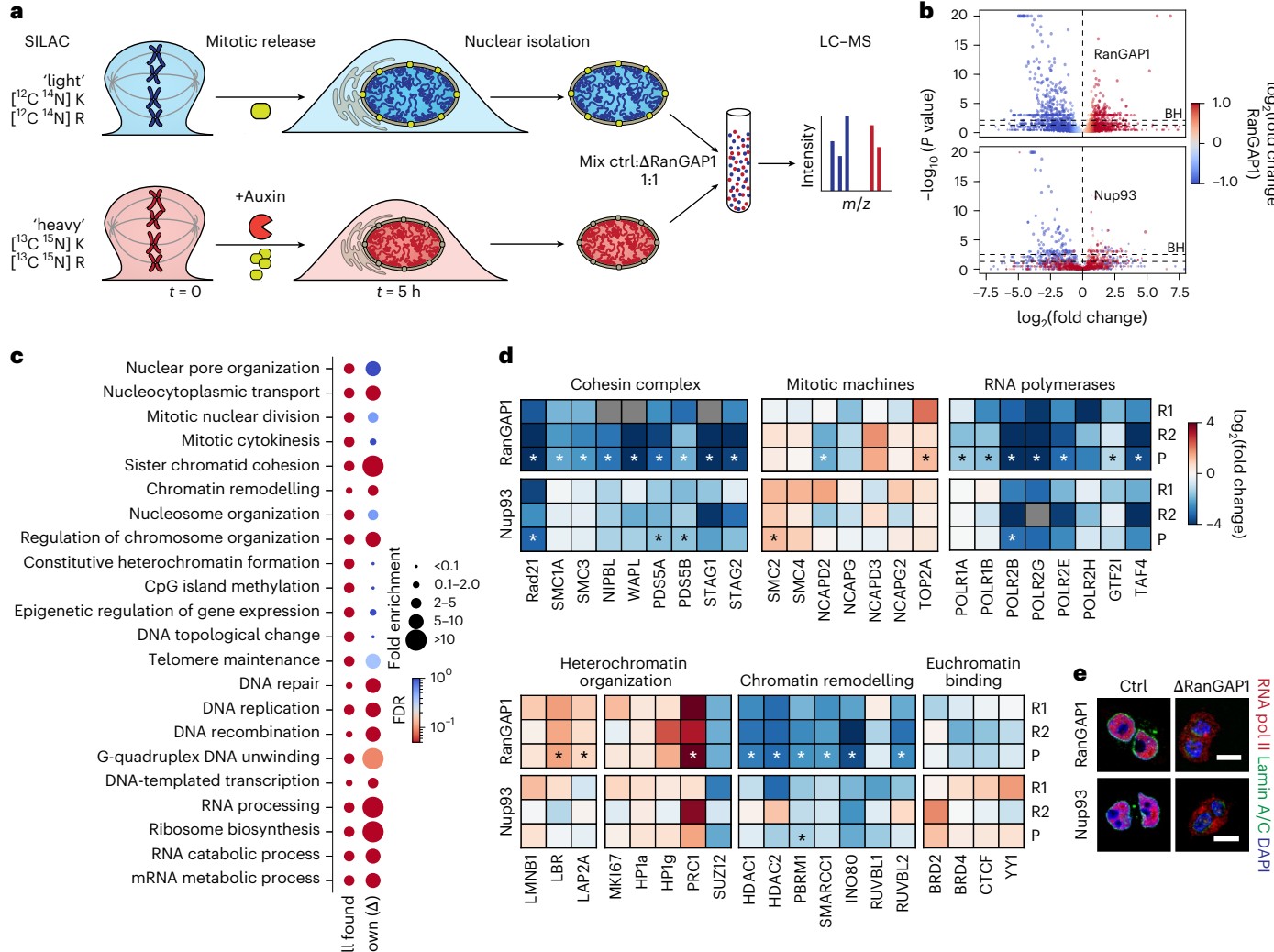

**Fig. 8 | The nuclear proteome of transport-deficient G1 cells. a,** Experimental workflow for SILAC-based LC–MS quantification of nuclear proteome alterations found in cells entering G1 ($t$ = 5 h) in the absence of either RanGAP1 or Nup93. Auxin-induced degradation was initiated 2 h before nocodazole/mitotic release ($t$ = 0). **b,** Volcano plots for the consensus list of proteins identified in early G1 nuclear isolates from RanGAP1– (top) and Nup93–AID (bottom) cells. For each protein, the enrichment in auxin-treated versus control (log$_2$-tranformed) is plotted against the multiple-testing Benjamini–Hochberg (BH)-adjusted $P$ values from two-sided non-parametric permutation tests ($P$ = 0.05 and the $P$ = BH threshold are indicated). Two pooled replicates for heavy and light reversed experiments are shown and each protein is coloured according to the auxin-treated/control ratio in the RanGAP1–AID. **c,** Gene ontology over-representation analysis for selected biological processes among all proteins identified in RanGAP1–AID and Nup93–AID control G1 nuclei (all found), or proteins reduced

by at least twofold in the nuclei of both RanGAP–AID- and Nup93–AID-depleted cells (down (δ)). Dots are coloured according to the false discovery rate (FDR) and dot sizes indicate fold enrichment. **d,** Changes in the abundance of select proteins involved in genome folding and function in RanGAP1–AID- and Nup93–AID-depleted G1 nuclei. The log$_2$-transformed fold enrichment in auxin-treated versus control nuclei was calculated separately for each replicate and for the pooled (reversed) LC–MS spectra. Any significant difference in the pooled samples, based on Benjamini–Hochberg (BH)-adjusted $P$ values, are indicated by an asterix. **e,** Representative immunofluorescence images (at least two independent experiments) of RanGAP1–AID or Nup93–AID control and depleted cells fixed 5 h after mitotic release demonstrating the nucleocytoplasmic localization of RNA pol II (red). Lamin A/C indicates the nuclear periphery. Scale bars, 10 μm.

Fig. 10c) without global changes to A/B identity (Fig. 7c and Extended Data Fig. 10d). These observations are consistent with previous findings in Nipbl-depleted cells where cohesin-dependent loop extrusion was effectively absent[72]. The binary classification of active and inactive chromatin from Hi-C data is an oversimplification of genome compartmentalization[77] that can be expanded to include cell type-specific subcompartments of different types by various methods[9,15,78,79]. To further investigate intrinsic compartmentalization, we employed interaction profile groups (IPGs) that were previously defined in wild-type DLD-1 cells[80] yielding active (A1 and A2), inactive (pooled to 'all B') and transcriptionally intermediate (V/VI) chromatin subcompartments. Preferential homotypic interactions between these subcompartment

loci increased as cells progressed from cytokinesis to G1 in control and RanGAP1-depleted cells (Fig. 7a,d,e (top)), indicating that compartmentalization at the scale of IPGs is entirely driven by determinants and factors associated with telophase chromosomes.

Microcompartment domains occurred at similar frequencies at A1 (6.6%), A2 (6.7%) and V/VI (5.9%) IPGs, and were largely absent from the B compartment (0.3%). We found that MCDs interacted most frequently with other MCDs, regardless of their subcompartment status, during cytokinesis (Fig. 7d,e (bottom)). This preference for MCD–MCD interactions was lost in control G1 cells when MCD interaction frequencies reflected the canonical subcompartment interaction affinities. Conversely, the MCD–MCD interactions continued to strengthen

during G1 in the absence of RanGAP1, maintaining a strong preference for microcompartmental interactions over contacts with other loci of the same subcompartment type. These data are consistent with the modified spectral clustering performed on RanGAP1-depletion Hi-C data, which demonstrated a distinct interaction profile for MCDs that did not correspond to a single specific IPG defined in wild-type cells (Extended Data Fig. 4). These results demonstrate that the cCRE microcompartment is quantitatively and temporally distinct from subcompartments defined in control G1 cells. In control cells, MCDs initially interacted regardless of subcompartment status but these interactions dissolved (were 'resorbed') and replaced by interactions with loci, including with loci that were not MCDs, that were generally assigned the same corresponding subcompartment status during mitotic exit. This resorption did not occur in nuclear import-deficient cells for which the cCRE microcompartment continued to strengthen (Fig. 7a, e (arrows) and Extended Data Fig. 10e).

### Identification of factors that require nuclear import for chromatin enrichment

For a better understanding of the chromatin state that gives rise to the cCRE microcompartment and to identify cytoplasmic factors that normally modify this pattern, we employed stable isotope labelling by amino acids in cell culture (SILAC)[81], followed by liquid chromatography with mass spectrometry (LC–MS) of nuclei isolated from cells exiting mitosis (Fig. 8a). Focusing on common changes to the nuclear proteome between RanGAP1 and Nup93 depletion, we found that universal changes to the protein composition were mostly factors that failed to localize to the nucleus as cells entered G1 (Fig. 8b and Supplementary Tables 4,5). Classification of these proteins according to annotated functions[82,83] identified biological processes that were retained or lost in import-deficient nuclei (Fig. 8c). As expected, chromatin-associated processes were enriched in the proteomes of isolated control nuclei. Consistent with our observations of normal mitotic exit kinetics, the absence of active nuclear transport did not consistently impact the presence of factors related to mitotic cell division, including proteins essential to mitotic chromosome condensation (Fig. 8d). We confirmed that the cohesin complex was completely absent in transport-incompetent nuclei. Other strongly depleted processes included DNA replication and repair as well as RNA processing and ribosome biogenesis, the latter being consistent with the absence of nuclear speckles and nucleoli in depleted nuclei (Extended Data Fig. 2e). Proteins involved in DNA-templated transcription were found to require nuclear import for access to the genome in early G1, which was confirmed by a dramatic reduction in components of the basal transcription machinery (Fig. 8d) and cytoplasmic localization of RNA pol II by immunofluorescence (Fig. 8e) in both RanGAP1- and Nup93-depleted cells. Finally, although we found that chromatin organization factors were relatively less consistent between depletions, many chromatin remodelling enzymes were substantially absent from the nucleus in both RanGAP1 and Nup93 depletions, whereas structural components of heterochromatin and euchromatin may be less dependent on transport for nuclear localization in newly divided cells. We conclude that loop extrusion, transcription and RNA processing probably do not occur, or occur to a dramatically reduced degree, in nuclei formed during mitotic exit in the absence of active nuclear transport. Given that canonical chromatin compartments and the cCRE microcompartment form under these conditions, our data suggest that these active processes are mechanistically dispensable for compartmentalization during mitotic exit.

## Discussion

We propose the existence of two folding programmes that specify interphase chromosome conformation as cells exit mitosis and enter G1. The first programme is driven by factors that associate with chromosomes no later than telophase to drive genome-wide compartmentalization based on the epigenetic signatures of the previous cell cycle. This programme includes and may even be driven by mitotically bookmarked *cis* elements and *trans* factors that form a prominent indiscriminate cCRE microcompartment starting in telophase. Although inherited in the same way, the cCRE microcompartment is distinct from conventional subcompartments, both in composition and cell cycle-dependent dynamics. A second folding programme starts after nuclear-envelope formation and requires active nucleocytoplasmic transport, implying that the relevant factors are inherited through mitosis in the cytoplasm. This programme includes cohesin, which drives the formation of interphase loops and TADs. It is intriguing that the two main known mechanisms of chromosome folding, compartmentalization and loop extrusion are inherited through mitosis in two distinct and physically segregated ways.

The extensive cCRE microcompartment observed during mitotic exit is a transient folding state that is extensively modulated as cells enter and progress through G1. Microcompartments constitute a discrete compartment type derived from different active subcompartments that display preferential homotypic interactions during cytokinesis and persist into G1 in the absence of nuclear active transport. The cCRE microcompartment, and canonical A and B compartments and subcompartments are all formed without import of factors from the cytoplasm, indicating that affinity-driven compartmentalization is generally mediated by factors stably associated with mitotic chromosomes or rapidly recruited before telophase. Several lines of evidence support the intrinsic inheritance of chromosome compartments. We found that the cCREs at strong MCDs are bookmarked during mitosis. Furthermore, histone modifications associated with active bookmarked loci (for example, H3K4Me3 and H3K72Ac) and more generally corresponding to subcompartments are at least partly retained during mitosis. It is possible that histone modifications are sufficient to mediate affinity-driven compartmentalization, although in vitro studies have suggested that bridging factors, such as BET proteins, mediate interactions between loci marked with acetylated histones[11]. Previous work has shown that Brd2 and Brd4 contribute to genome compartmentalization during interphase[84] and at condensin-depleted mitotic chromosomes[85], respectively. However, we found that inhibition or degradation of BRD2 and BRD4 did not impact microcompartment interactions during mitotic exit, which implies an alternative mechanism for microcompartment affinity.

Consistent with previous reports[60,61], we observed that a burst of transcription occurs shortly after mitotic exit. The earliest genes expressed following mitotic exit in control cells are preferentially enriched at MCDs compared with genes of the late transcriptional programme, suggesting that microcompartment formation in telophase–cytokinesis could contribute to early post-mitotic gene reactivation. We noted that by G1, when microcompartment interactions are most frequent in RanGAP1-depleted cells, nascent RNA synthesis was strongly attenuated and the transcriptional machinery was excluded from the nucleus. Together, these findings suggest that microcompartment contacts formed during mitotic exit could promote early post-mitotic gene reactivation in control cells but that microcompartment formation itself is independent of transcription. Similarly, compartmentalization has been linked to genome localization at speckles[86] and nucleoli[87] but the absence of these structures in nuclear transport-deficient cells did not adversely impact any level of global compartmentalization we examined.

We found that a second folding programme is inherited through the cytoplasm at the end of mitosis and thus relies on nuclear transport-dependent factors. Our ATAC–seq data demonstrate that the cell type-specific G1 chromatin landscape relies on chromosome-intrinsic and cytoplasmic factors. Although many cCREs overlapping MCDs remain accessible, bookmarked sites may also be inaccessible during mitosis and require active remodelling during G1 (ref. 29). Accordingly, our proteomics analysis identified a number

of import-dependent chromatin remodelling complexes that may contribute to re-establishing the G1-specific architecture. Cohesin can be recruited anywhere along the genome but loop extrusion patterns are sensitive to the presence and location of active *cis* elements[16,22] and transcriptional machinery[88]. The second folding programme inherited through the cytoplasm may therefore open distal regulatory sites and define a cell type-specific loop-extrusion traffic pattern.

The nuclear import requirement for cohesin accumulation at post-mitotic chromosomes, recently observed by microscopy[42], physically and temporally segregates two modes of chromatin extrusion at the end of mitosis, which is consistent with the existence of an extrusion-free chromosome folding intermediate during telophase[39]. Although the relevance of this separation remains to be seen, it enabled functional segregation of the two main chromatin folding mechanisms in our system.

The chromosome-intrinsic capacity for previously active and mitotically bookmarked promoters and enhancers to interact gives rise to a comprehensive array of non-specific contacts during telophase–cytokinesis. Initial microcompartment formation is promiscuous in *cis* and *trans*, and does not require loop extrusion. The formation of extrusion-independent promoter-enhancer contacts has been observed previously in different cell contexts and cell-cycle stages[54,59,89–93]. It is possible that this microcompartment serves to rapidly re-establish gene-regulatory contacts during mitotic exit. As cytoplasmic factors enter the nucleus, long-range (>1 Mb) MCD–MCD interactions are substantially reduced or pruned. We propose that G1 pruning of indiscriminate MCD contacts is driven by cohesin-mediated loop extrusion. Most MCDs overlap CTCF sites that anchor extruded loops in G1. These MCDs retain short-range interactions with other MCDs, possibly reinforced by loop extrusion. Pruning during mitotic exit ensures that MCD–MCD interactions start to obey rules of TAD formation such that interactions within extrusion domains are more likely to be retained than interactions between domains.

Because the effect of pruning can be partially reversed by Ran-GAP1 depletion in G1, we conclude that interphase chromatin folding is the result of continuous interplay between intrinsic affinity-based compartmentalization and cytoplasmic inherited processes, such as loop extrusion. We propose that the chromosome-intrinsic folding programme reveals a universal propensity for active gene-regulatory elements to interact through affinity-driven interactions. A second folding programme superimposes a more deterministic logic through regulated cohesin-mediated loop extrusion that ensures the specific pairing of cCREs to ensure cell type-specific gene expression.

## Online content

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

## Methods

### Experimental methods

**Genome modifications.** The CRISPR–Cas9 system was used to endogenously target the *RanGAP1*, *RCC1* (refs. 94,95), *NUP93* (ref. 45) and *AAVS1* (ref. 96) genes. With the exception of nuclear import assays, all experiments described here employed human colorectal adenocarcinoma DLD-1 cells (American Type Culture Collection, CCL-221) expressing either RanGAP1 or Nup93 homozygously tagged with NeonGreen and an AID, infra-red protein (IFP)-tagged RCC1 and Tir1, as described previously. We refer to these cell lines as RanGAP1–AID and Nup93–AID. For cell lines used in nuclear import assays, the sequences of MBP and mScarlet were amplified by PCR from pMAL (NEB) and pmScarlet_αTubulin_C1 (Addgene, catalogue number 85045), respectively. The NLS sequence was synthesized and all fragments were inserted by Gibson reaction (NEB, E2611S) into the multiple cloning site of AAVS1_Puro_PGK1 vector (Addgene, catalogue number 68375) through replacement of 3×FlagTwinStep-Tag. MBP–mScarlet–NLS was inserted into the AAVS1 locus in RanGAP1–AID and NUP93–AID DLD-1 cell lines.

**Cell culture and cell-cycle synchronization.** Modified DLD-1 cells lines (RanGAP1–AID, Nup93–AID, RanGAP1–AID/MBP–NLS and Nup93/MBP–NLS) were cultured at 37 °C with 5% $CO_2$ in DMEM, high glucose, GlutaMAX supplement, pyruvate medium (Gibco, 10569010) with 10% fetal bovine serum (Gibco, 16000069) and 1% penicillin–streptomycin (Gibco, 15140122). The cells were synchronized in prometaphase of mitosis using a standard double thymidine-nocodazole block protocol. All incubations and washes were carried out in solutions pre-warmed to 37 °C. The cells were seeded at low density in DMEM medium containing 4 mM thymidine (Sigma, T1895) and incubated for 17 h, released (after two PBS washes) for 7 h in fresh medium to replicate the genome and enter mitosis, and incubated with 4 mM thymidine for an additional 17 h to obtain synchronous populations blocked in S-phase. After the second thymidine block, the cells were released (after two PBS washes) in fresh medium for 5 h, followed by an additional 5 h incubation in the presence of 50 ng ml$^{-1}$ nocodazole (Sigma, M1404) to accumulate in prometaphase. Degradation of RanGAP1–AID and Nup93–AID was achieved in prometaphase-arrested cells by the addition of 1 mM of 3-indoleacetic acid (IAA, auxin; Sigma-Aldrich, 15148). Following incubation for an additional 2 h, the control and depleted cells were either collected for subsequent analyses (described later) or released into fresh medium to progress through mitosis for 1.25 h (telophase), 1.5 h (cytokinesis) or 5 h (early G1).

To deplete RanGAP1 specifically in G1, the medium was exchanged 3.5 h after nocodazole release and adherent cells were incubated with auxin for an additional 6.5 h into late G1 (10 h). Matched 10 h control and depleted conditions were also subjected to removal of non-adherent cells at 3.5 h to ensure similar G1 synchrony. To inhibit BET protein activity during mitotic exit, 500 nM JQ1 (Selleckchem, S7110) or dBET6 (PROTAC; Selleckchem, S8762) was added to prometaphase-arrested cells in the presence or absence of auxin. After another 2 h of incubation, the control and RanGAP1-depleted cells were released for 5 h into early G1 in the presence or absence of BET inhibitors.

**Import assay of MBP–mScarlet–NLS in AID-tagged cell lines.** DLD-1 RanGAP1–AID and NUP93–AID cell lines, expressing MBP–mScarlet–NLS, were cultured on four-well glass-bottomed chambers (Ibidi) and synchronized with CDK1 inhibitor (Selleckchem, Ro-3306) for 16 h. After 15 h in the CDK1 inhibitor, the cells were treated with fresh medium or medium containing 1 mM auxin for 1 h in the presence of the CDK1 inhibitor. The cells were washed three times with fresh medium and released in regular medium or medium containing 1 mM auxin. The cells were imaged on an Eclipse Ti2 inverted microscope (Nikon) equipped with a spinning disk confocal system (UltraVIEW Vox Rapid Confocal Imager; PerkinElmer) and controlled with the Volocity software (PerkinElmer) utilizing a Nikon PlanFluor ×40/1.3 oil-immersion

objective lens. The cells were imaged in FluoroBrite DMEM (Thermo Fisher Scientific) medium. The microscope was equipped with a temperature-, $CO_2$- and humidity-control chamber that maintained a 5% $CO_2$ atmosphere and 37 °C. RanGAP1-NG, NUP93-NG, MBP–mScarlet–NLS and RCC1iRFP670 fluorescent protein signals were excited with 488-nm (20–24% power, 200–250 ms exposure), 568-nm (5% power, 50–60 ms) and 640-nm (100% power, 150–200 ms exposure) laser lines, and binning set to 2×, respectively. A series of six z-slices, 2 μm optical sections, were acquired every 2 min and monitored for 6 h for three independent fields. Images were captured and analysed using the Volocity (PerkinElmer) and Image J (National Institutes of Health) software. Images represent a single z-stack.

**Nuclear import image analysis.** The intensity of the MBP–mScarlet–NLS in three independent regions of interest (5 × 5 pixels) was measured every 10 min in the nucleus and the cytoplasm. To calculate relative nuclear intensity, we used the following equation:

$$RelInt.Nu = AvgInt.Nu − AvgInt.B/(AvgInt.Nu − AvgInt.B)$$

$$+(AvgInt.Cyt − AvgInt.B),$$

where Avg Int. Nu is the average intensity of MBP–mScarlet–NLS in the nucleus, Avg Int. Cyt is the average intensity of MBP–mScarlet–NLS in the cytoplasm and Avg Int. B is the average background intensity of MBP–mScarlet–NLS in the cell-free area. The relative nuclear-to-cytoplasmic intensity of MBL–mScarlet–NLS was plotted. Data are expressed as individual values analysed every 10 min for eight control or auxin-treated cells of two independent fields of one experiment. Assessment of MBL–mScarlet–NLS import was performed in two independent experiments.

**Western blotting.** Adherent cells were dissociated with accutase and equal numbers of cells were lysed with 2×Laemmli buffer (60 mM Tris pH 6.8, 10% glycerol, 2% SDS, 100 mM dithiothreitol) by boiling for 10 min. Proteins were separated on 4–12% NuPage bis-Tris gels using 1×MES running buffer (NuPAGE 20X buffer NP0002) for 45 min at 175 V in an Invitrogen XCell Sure-Lock mini gel and blotting system. The gels were transferred to 0.2-μm nitrocellulose membrane in Pierce 10X Western Blot Transfer Buffer, Methanol-free (Thermo Fisher Scientific, 35040) at 30 V for 1 h. For immunoblotting, the membranes were blocked with 5% milk in TBS-T (1×TBS + 0.1% Tween-20) at room temperature for at least 1 h. Primary antibodies were diluted in block buffer and incubated overnight at 4 °C and horseradish peroxidase (HRP)-linked secondary antibodies for 1 h at room temperature. The blots were developed and imaged using SuperSignal West Dura Extended Duration Substrate (Thermo Fisher Scientific, 34076) and a Bio-Rad ChemiDoc system. The following primary antibodies were used: mouse anti-RanGAP1 (OTI1B4; Novus Biologicals, NBP2-02623), mouse anti-Nup93 (F-2; Santa Cruz Biotechnology, sc-374400), rabbit anti-BRD4 (E2A7X; Cell Signaling Technologies, 13440), rabbit anti-BRD2 (EPR7642; Abcam, ab139690) and rabbit anti-vinculin (EP18185; Abcam, ab129002). The following secondary antibodies were used: goat anti-mouse IgG–HRP (Cell Signaling Technologies, 7076) and goat anti-rabbit IgG–HRP (Cell Signaling Technologies, 7074).

**Immunofluorescence.** Immunofluorescence was performed using standard methods. For early G1 immunofluorescence, cells were released from prometaphase onto glass coverslips and fixed with 4% paraformaldehyde in PBS for 30 min at room temperature. For mitotic release experiments, the cells were first fixed and then concentrated onto glass coverslips using an Epredia Cytospin at 150g. The fixed samples were incubated with PBS containing 2% Triton X-100 for 1 h before primary and secondary antibody staining in PBS containing 0.1% Triton X-100 for 3 and 1 h, respectively. After staining with 10 μg ml$^{-1}$ DAPI in PBS for 10 min at room temperature, the coverslips

were mounted in Vectashield antifade medium (Vector Labs, H-1000-10) for confocal imaging. The following primary antibodies were used: mouse anti-α-tubulin (Sigma T6199), rabbit anti-histone H3pS28 (Abcam, ab5169), rabbit anti-lamin B-receptor (Abcam, ab32535), rabbit anti-lamin A (Abcam, ab26300), mouse anti-Elys (BioMatrix Research, BMR00513), rabbit anti-Nup160 (Abcam, ab73293), mouse anti-Mab414 (Abcam, ab24609), rabbit anti-SON (Thermo Fisher Scientific, PA5-65107), mouse anti-NPM1 (Thermo Fisher Scientific, 60096-1), rabbit anti-Rad21 (Abcam, ab154769), rabbit anti-CTCF (Cell Signaling Technologies, 2899) and rabbit anti-RNA pol II pS2 (Abcam, ab5095). The following secondary antibodies were used: goat anti-mouse IgG H&L Alexa Fluor 488 (Abcam, ab150113), goat anti-mouse IgG H&L Alexa Fluor 568 (Abcam, ab175473), goat anti-rabbit IgG H&L Alexa Fluor 488 (Abcam, ab15007) and goat anti-rabbit IgG H&L Alexa Fluor 568 (Abcam, ab175471).

**Confocal microscopy.** Images were acquired using a Nikon A1 point-scanning confocal microscope with GaAsP detectors (488 and 561 nm lasers) or a high sensitivity MultiAlkali PMT (405 nm laser) and an Apo TIRF, 1.49 numerical aperture, ×60 oil-immersion objective (Nikon). For chromatin volume estimation, fixed DAPI-stained cells were imaged in 40–60 consecutive 0.2-μm z-slices. The 3D volume of DAPI-containing signal was used as a proxy for chromatin volume and measured using the '3D Objects Counter' Plugin[97] of the Fiji software[98] on pre-filtered (Gaussian, σ = 2) image stacks.

**Flow cytometry.** Cells were collected at various points of mitotic exit. Adherent cells were dissociated with accutase (Thermo Fisher Scientific, A11105-01) and pooled with non-adherent collected cells to assess the entire population. To assess the cell-cycle profile (DNA content), cell pellets were resuspended in 200 μl PBS and fixed with 800 μl of cold 100% ethanol. The cells were stored at −20 °C for at least 24 h. Approximately $1 × 10^6$ fixed cells were stained with 50 μg ml$^{-1}$ PI (Thermo, P1304MP), diluted in 1 ml PBS containing 50 μg ml$^{-1}$ RNase A (Roche, 10109169001) and 0.1% saponin, for 1 h at room temperature. After staining, the cells were spun, and the pellets were resuspended in 1 ml PBS and passed through a 35-μm filter (Falcon, 352235). Flow cytometry was performed on a MACSQUANT set-up. Analysis was performed using the FlowJo software (v10) and plots reflect populations gated for debris but not doublets.

**Hi-C fixation and FACS analysis.** Cells were collected at various points of mitotic exit and fixed for Hi-C 3.0 analysis[52] with a few modifications to facilitate cell sorting. Adherent cells collected 5 and 10 h after prometaphase release were dissociated with accutase. Prometaphase (t = 0) and early (t = 1.25–1.50 h) released cells were directly collected by shake-off. Cell suspensions were pelleted and treated with accutase for an additional 5 min at room temperature to prevent aggregation and washed with HBSS (Thermo Fisher Scientific, 14025134). Fixation proceeded first with 1% formaldehyde (Fisher Scientific, BP531-25) in HBSS for 10 min, which was quenched with 0.125 M glycine for 5 min at room temperature and 15 min on ice. Next, the cells were fixed with 3 mM disuccinimidyl glutarate in PBS for 40 min at room temperature with rotation, followed by a second quenching with 0.125 M glycine. The fixed cells were washed twice with PBS + 0.1% BSA and snap-frozen in liquid nitrogen before staining for FACS analysis.

To sort cells by DNA content, approximately $10 × 10^6$ fixed cells were stained with 50 μg ml$^{-1}$ PI, diluted in 5 ml PBS containing 50 μg ml$^{-1}$ RNase A and 0.1% saponin, for 1 h at room temperature. The cells were then spun and washed with PBS before resuspension in 2 ml PBS + 0.1% BSA and passage through a 35-μm filter. Propidium iodide-stained cell suspensions were sorted on a BD FACS Melody system using the 561 nm laser for forward scatter, side scatter and PI. All populations were gated based on forward scatter/side scatter to eliminate cell debris and cells sorted for either prometaphase (4n) or G1 (2n) DNA content were also

subject to doublet discrimination. To enrich for telophase or cytokinesis, cells fixed 1.25 and 1.50 h after mitotic release, respectively, were sorted based on doubled PI signal (DNA content) area and width. All sorted cells were collected in PBS containing 1% BSA and washed twice in PBS before being snap-frozen.

**Hi-C.** Chromosome conformation capture was performed as previously described[52] with some modifications. Synchronized cells were crosslinked with 1% formaldehyde and 3 mM disuccinimidyl glutarate, and enriched in specific cell-cycle stages by FACS, as described above. Sorted cells ($1–5 × 10^6$) were collected and snap-frozen for storage at −80 °C before lysis. After lysing the cells and digesting the chromatin with 400 U each of DpnII and DdeI (NEB, R0543M and NEB, R0175) overnight, the DNA ends were labelled with biotinylated dATP (LifeTech, 19524016) using 50 U Klenow DNA polymerase (NEB, M0210). Blunt-end ligation was performed with 50 U T4 ligase (Life Technologies, 15224090) at 16 °C for 4 h, followed by overnight reverse crosslinking with 400 μg ml$^{-1}$ proteinase K (Thermo Fisher Scientific, 25530031) at 65 °C. The DNA was purified using phenol–chloroform extraction and ethanol precipitation, and concentrated on a 30 kDa Amicon Ultra column (EMD Millipore, UFC5030BK). Biotin was removed from the unligated ends in 50 μl reactions using 50 U T4 DNA polymerase (NEB, M0203) per 5 mg of DNA. Following DNA sonication (Covaris S220 system) and solid-phase reversible immobilization bead-size fractionation to generate DNA fragments 100–300 bp in length. The DNA ends were repaired using 7.5 U T4 DNA polymerase, 25 U T4 polynucleotide kinase (NEB, M0201) and 2.5 U Klenow DNA polymerase (NEB, M0210). Libraries were enriched for ligation products by biotin pulldown with MyOne streptavidin C1 beads (Invitrogen, 65001). To prepare for sequencing, A-tailing was performed using 15 U Klenow DNA polymerase (3′–5′ exo-; NEB, M0212) and either TruSeq DNA LT kit indexed adaptors (Illumina, 20015964) or NEBNext multiplex oligos (NEB, E7780S) were employed. Libraries were amplified in PCR reactions for 5–7 cycles using a TruSeq DNA LT kit (Illumina, 15041757) and subjected to solid-phase reversible immobilization bead-size selection before sequencing on either an Illumina HiSeq 4000 or a NovaSeq 6000 system using the paired-end 50 bp or 100 bp modules. Two biological replicates were performed for each condition, with the exception of early mitotic exit populations in telophase or cytokinesis. Hi-C datasets have been deposited at National Center for Biotechnology Information (NCBI)'s Gene Expression Omnibus (GEO; GSE277875; SuperSeries, GSE278023).

**ATAC–seq.** Chromatin accessibility was investigated using Omni-TAC–seq[99,100] with some modifications. Prometaphase-arrested cells were collected by mitotic shake-off and adherent cells were dissociated using accutase 5 h after prometaphase release. Viable cells (50,000) were permeabilized in 50 μl cold Resuspension Buffer (10 mM Tris–HCl pH 7.4, 10 mM NaCl and 3 mM MgCl$_2$) containing 0.1% NP-40 (MP Biomedicals, 0219859680), 0.1% Tween-20 and 0.01% digitonin (Promega, G9441) for 3 min on ice. The cells were isolated by centrifugation at 500g and 4 °C for 10 min, and the buffer was exchanged by washing once with the detergent-free Resuspension Buffer. Tagmentation was performed in permeabilized cells resuspended in 50 μl Tagmentation Buffer (Diagenode, C01019043) containing 100 nM adaptor-loaded Tn5 transposase (Diagenode, C01070012-30), 0.001% digitonin and 0.1% Tween-20 at 37 °C for 30 min with intermittent mixing. The reaction was stopped by the addition of Binding Buffer (Qiagen MinElute PCR Kit, 28004) and DNA was purified using a Qiagen MinElute PCR kit according to the manufacturer's protocol. Purified DNA was eluted in 21 μl Elution Buffer and stored at −20 °C for library preparation. To prepare ATAC–seq libraries for sequencing, custom barcoded primers based on a previous design[101] were used in an initial five-cycle pre-amplification using NEBNext high fidelity PCR master mix (NEB, M0541). The number of PCR cycles for PCR amplification was determined using quantitative PCR. Following PCR amplification, libraries were purified using

solid-phase reversible immobilization beads, with a sample-to-bead ratio of 1.0:1.5. ATAC−seq libraries were sequenced on an Illumina NextSeq 2000 machine using the 50 bp paired-end reagents. Two biological replicates were performed for each condition. ATAC−seq datasets have been deposited at the NCBI GEO accession GSE277731 (SuperSeries GSE278023).

**CUT&RUN.** The CUT&RUN protocol was modified from Skene and Henikoff[65] and applied to control or RanGAP1-depleted cell populations arrested in prometaphase or released for 5 h into G1. Approximately $5 \times 10^6$ cells per condition were harvested and lysed (0.1% Triton X-100, 20 mM HEPES-KOH pH 7.9, 10 mM KCl, 0.5 mM spermidine, 20% glycerol and Roche EDTA-free protease inhibitor; Sigma-Aldrich, 11873580001) for 10 min on ice. Due to the inclusion of prometaphase samples, the cells were centrifuged for 3 min at 600$g$ and 4 °C after every wash or buffer exchange. The lysed cells were incubated for 5 min on ice with wash buffer (20 mM HEPES pH 7.5, 150 mM NaCl, 0.5 mM spermidine, 0.1% BSA and Roche protease inhibitor) containing 2 mM EDTA and washed once more with wash buffer before resuspension in 1 ml wash buffer for antibody binding. Primary antibodies to histone H3K4me3 (Abcam, ab8580; 1:100), H3K27ac (D5E4; Cell Signaling Technologies, 8173; 1:100), H3K27me3 (Cell Signaling Technologies; 1:100) or CTCF (Cell Signaling Technologies, 2899; 1:50) were bound overnight on a rotator at 4 °C. A control IgG (guinea pig anti-rabbit; ABIN101961; 1:100) sample was included for each replicate in control prometaphase and G1 cell populations. The samples were washed three times (2 min each) in wash buffer with rotation at 4 °C, followed by pAG-MNase binding (1/20 dilution; CUTANA pAG-MNase for ChIC/CUT&RUN workflows; EpiCypher, 15-1016) for 1 h at 4 °C with rotation. Following three washes (5 min each at 4 °C) with rotation, the samples were resuspended in 150 µl wash buffer and MNase was activated on wet ice by the addition of $CaCl_2$ (final concentration of 2 mM) for 30 min. The reaction was quenched with 150 µl of 2×STOP buffer (200 mM NaCl, 20 mM EDTA, 4 mM EGTA, 50 µg ml$^{-1}$ RNase A and 40 µg ml$^{-1}$ glycogen) containing 1:1,000 of 10 ng ml$^{-1}$ heterologous DNA (CUTANA *E. coli* spike-in DNA; EpiCypher, 18-1401) and incubated for 20 min at 37 °C. After centrifugation at 16,000$g$ for 5 min, the supernatant was incubated at 65 °C for 10 min with 0.1% SDS, 0.2 mg ml$^{-1}$ proteinase K and 50 µg ml$^{-1}$ RNase A. Extraction of DNA using 1:1 phenol:chloroform extraction and ethanol precipitation was followed by AMpure XP bead (Beckman Coulter, A63881) size selection to remove fragments larger than about 700 bp. Libraries were prepared for sequencing using the NEBNext Ultra II end repair/dA-tailing module (NEB, E7546) and NEBNext multiplex oligos for Illumina (NEB, E7600 and E7780). The libraries were amplified in PCR reactions for 14–18 cycles and subjected to AMpure XP bead-size selection before sequencing on an Illumina NextSeq 2000 system using the paired-end 50 bp module. Two biological replicates were performed for each condition. The CUT&RUN datasets have been deposited at the NCBI GEO (GSE308844; SuperSeries, GSE278023).

**Sequencing of nascent and total RNA.** DLD-1 cells were arrested at prometaphase, as described earlier, and released to progress through mitosis over 5 h, with samples collected every hour. The cells were labelled with 1 mM 4sU in 1-h non-overlapping intervals comprising the final hour of culture for each collection time point. Unlabelled controls were included at prometaphase arrest and at the end of the 5-h time course, and each experiment was performed in triplicate. At the end of each time point, the cells were collected and snap-frozen in liquid nitrogen, and the cell pellets were lysed in TRIzol reagent before total RNA extraction using a Zymo RNA Clean & Concentrate kit. Before RNA extraction, six ERCC spike-ins were added to the lysed pellet at a concentration relative to the number of cells harvested: three ERCCs labelled with 10% 4sU and three unlabelled ERCCs, as described previously[102]. SLAM−seq libraries were prepared as previously described[62]. Briefly, ribosomal RNA was depleted as previously described[103], the

RNA was treated with 10 mM iodoacetamide to alkylate 4sU bases and libraries were prepared using a NEBNext Ultra II directional RNA library prep kit according to the manufacturer's recommendations. Library quality was confirmed using Agilent TAPE station reagents and libraries were sequenced on an Illumina NextSeq 2000 platform (2 × 100 paired-end reads) to an average depth of $67 \times 10^6$ reads per library. The SLAM−seq datasets have been deposited at the NCBI GEO (GSE309609; SuperSeries, GSE278023).

**Stable isotope labelling by amino acids in cell culture.** We performed SILAC described previously[81,104]. Before cell-cycle synchronization, DLD-1 RanGAP1−AID and Nup93−AID cells were labelled by culturing them for five days in heavy (L-arginine-$^{13}C_6$, $^{15}N_4$ hydrochloride and L-lysine:2HCl $^{13}C_6$ $^{15}N_2$; Sigma-Aldrich, 608033 and Cambridge Isotope, CNLM-291-H, respectively) or light (L-arginine hydrochloride and L-lysine; Sigma-Aldrich, A6969 and L5501, respectively) DMEM for SILAC medium (Thermo Fisher Scientific, A33822) with dialysed fetal bovine serum (Sigma-Aldrich, F0392). SILAC was maintained throughout the cell synchronization experiment and nuclei were isolated for LC−MS. For each cell line, one replicate experiment was performed using light control and heavy IAA-treated samples, and labelling was then reversed for the second biological replicate.

**Nuclear isolation and liquid chromatography–mass spectrometry.** Nuclei were isolated by manual disruption based on a previous protocol[105] for mass spectrometry analysis. Early G1 labelled cells were collected 5 h after prometaphase release and washed with PBS. The cells ($10 \times 10^6$) were carefully resuspended with a broad pipette tip in 1 ml hypotonic buffer (10 mM HEPES pH 7.9, 1.5 mM $MgCl_2$ and 10 mM KCl) containing 0.1 mM phenylmethyl sulfonyl fluoride, 0.5 mM dithiothreitol (Thermo Fisher Scientific, BP172-25) and 1×protease inhibitor cocktail (Thermo Fisher Scientific, 78440), and incubated on ice for 25 min. The cells were then ruptured by douncing 40× in a pre-chilled homogenizer (tight pestle (B)) and nuclei were isolated by centrifugation for 5 min at 1,500$g$ and 4 °C, and resuspended in a 10 mM Tris buffer pH 7.4 containing 2 mM $MgCl_2$. The samples were adjusted to 1×Laemmli buffer (60 mM Tris pH 6.8, 10% glycerol, 2% SDS and 100 mM dithiothreitol) and heated to >85 °C for 10 min before gel electrophoresis. Heavy and light samples were mixed at a 1:1 ratio of total protein based on quantification of stained gels (GelCode Blue Safe Protein Stain; Thermo Fisher Scientific, 24594). The final protein gels were stained with GelCode blue safe protein stain and bands were excised for LC−MS sample processing.

Overnight trypsin digestion was performed at 37 °C. After protein digestion and drying in a SpeedVac, the samples were reconstituted in 25 µl MS solvent (5% acetonitrile and 0.1% formic acid) and 3.8 µl was injected into a Fusion Lumos Orbitrap mass spectrometer in OTOT mode with a 90 min gradient. Peptides were searched against the SwissProt human database in Maxquant and proteins were subject to a two-peptide cutoff. Two replicates were performed for each condition, reversing the SILAC labelling, and >3,000 proteins were detected in each sample at 1% false discovery rate (summarized in Supplementary Tables 4 and 5). Fold changes and Benjamini−Hochberg-corrected $P$ values comparing control and depleted cell conditions were determined using the Q+ analysis by Scaffold. Full processed datasets are in Supplementary Tables 4 and 5, and raw data have been deposited to the ProteomeXchange Consortium via the PRIDE[106] partner repository with the dataset identifier PXD056346.

### Genomic analyses
**Published tracks and annotations.** Publicly available CTCF chromatin immunoprecipitation with sequencing (ChIP−seq) coverage from untreated DLD-1 cells and the corresponding bigWig (GSM4238559) was used to define cCREs and throughout the paper for visualization unless otherwise indicated. Variant 1 of the MA0139 CTCF motif

annotation from the Jaspar database[107] was used to define convergent extrusion loops. Publicly available RNA-sequencing data for the DLD-1-RanGAP1 cell line (GSE132363: GSM3860900, GSM3860901 and GSM3860902) were used to define active genes for visualization and ATAC–seq analysis. Raw data were processed using the nf-core/rnaseq pipeline (v.3.15.0; https://github.com/nf-core/rnaseq )[108]. Genes with fragments per kilobase of transcript per million mapped reads (FPKM) > 0 in all three replicates and FPKM > 1 in the pooled dataset were defined as active. This information is summarized in Supplementary Table 3.

*Cis*-regulatory elements. We used a combination of publicly available CTCF ChIP–seq coverage for DLD-1 together with the control G1 histone H3K4me3 and H3K27ac CUT&RUN and ATAC–seq data generated in this study to define CREs according to the procedure defined by ENCODE[109]. Briefly, we annotated the list of cell line-independent DNase hypersensitive sites from ENCODE v2 with *z*-score-transformed ATAC–seq, H3K4me3 and H3K27ac signal, and used the distance to the nearest transcription start site (TSS) to assign DHSs to seven CRE groups: PLS (22,555 promoter-like: open, H3K4me3 enriched, within 200 bp of the nearest TSS), pELS (28,863 combined proximal and near enhancer-like: open, H3K27ac enriched and within 2 kb of the nearest TSS), dELS (43,807 distal enhancer-like: open, H3K27ac enriched and at least 2 kb away from the nearest TSS), H3K4me3 (1,417 open H3K4me3: open, H3K4me3 enriched and at least 200 bp away from the nearest TSS), CTCF (9,911 open, CTCF enriched and not enriched in any other marks) and finally, open (43,316 open, not enriched in any marks). Where indicated, binned genomic loci were assigned one cCRE based on the following hierarchy: PLS > pELS > dELS > CTCF > just open > not open. The fold enrichment of binned cCREs in specific genomic regions was determined using the ChromHMM 'OverlapEnrichment' function[110], which normalizes the fraction of bases in a particular state (that is, PLS) found at a subset of genomic loci (that is, MCDs) to the fraction of those loci region genome-wide.

**Aggregation stackup analysis.** Stackups demonstrate the behaviour of a given signal (for example, H3K4me3 CUT&RUN) at a set of genomic loci, typically centred at those loci and presented as a heat map, where each row represents the signal around the individual genomic locus. We took advantage of the Python application programming interface (API)[111] built around UCSC BBI library[112] to extract a signal stored in a bigWig or bigBed files given a set of same-sized genomic intervals using the 'stackup' function. We ensured our intervals had the identical size by centering on the loci of interest and providing fixed-length upstream and downstream flanks. For every stackup, we also generated a summary signal profile by averaging the stackup across all rows for every column. We used 100 kb for upstream and downstream flank sizes, and aggregated the signal into 100 bins for every row of the stackup, logarithmic colour scales were used throughout the stack-ups with the exception of EV1 profiles and bigBed-derived signals—coverages of loop anchors, MCD anchors and so on.

**Hi-C data pre-processing.** Hi-C libraries were processed using the distiller-nf pipeline[113] (v.0.3.4): paired-end reads were mapped to hg38 human reference genome using bwa mem[114] in a single-sided fashion (-SP); read alignments were parsed and classified into pairwise interactions, or pairs, by parse from the pairtools package[115] (v.1.0.2) and the additional '–walk-policy all' option was used to rescue multiway interactions (walks); after the removal of duplicates, uniquely mapped and rescued pairs were further filtered according to their alignment quality (MAPQ > 30) and subsequently aggregated into binned contact matrices in the cooler format[116] at a resolution of 1, 2, 5, 10, 25, 50, 100, 250, 500 and 1,000 kb; contact matrices were normalized using the iterative correction normalization[77] with the default parameters (for example, the first two diagonals were excluded from balancing

at each resolution to avoid short-range ligation artifacts); bins with extreme genomic coverage, as detected by the MADmax (maximum allowed median absolute deviation) filter[72], were masked and excluded from the analysis. Summary mapping statistics, as produced by the pairtools module from MultiQC[117], are in Supplementary Table 1 and Supplementary Fig. 1a. We used insulation tracks and *cis* Eigvectors to assess the reproducibility of the replicates, which were pooled together as depicted in Supplementary Fig. 1a.

**Extraction of Hi-C features.** *Common setting for feature extraction*. To extract Hi-C features relevant for our analyses we used the cooltools package[55] (v.0.7.0); specifically, we leveraged the cooltools Python API for scripting our analyses in the form of Jupyter Notebooks[118]. We used arms of human autosomal chromosomes (Chr 1–22) as a genome partitioning for the analyses (parameter 'view_df') and default parameters in the API function calls unless specified otherwise. In the following sections, we provide a brief description of specific functions that were used to extract each feature.

*Scaling analysis*. Frequency of interactions as a function of genomic separation (scaling plots, *P*(*s*)) was calculated using balanced Hi-C data binned at 1 kb using the 'expected_cis' function with smoothing in log-space enabled, data for chromosome arms were aggregated to generate an average genome-wide scaling. The 'gradient' function of the numpy[119] package was used to calculate the rate at which the interaction frequency changed with distance, that is, scaling plot derivatives in log-log space as demonstrated in the 'contacts_vs_distance' notebook from the 'open2c_examples' repository (https://github.com/open2c/open2c_examples).

*Normalization by 'expected'*. Most downstream analyses require Hi-C matrices to be 'flattened', that is, normalized to the decay of interaction frequency with genomic distance. We use the expected_cisfunction to calculate such an 'expected'. In case of *trans* or interchromosomal data, matrices were normalized to average levels of interchromosomal interactions, calculated using the 'expected_trans' function. Results of these functions were passed to the downstream analyses when applicable.

*Insulation*. Diamond insulation score[120] was calculated using the 'insulation' function at 10 kb resolution and 100 kb averaging window size (size of the insulation diamond).

*Eigenvectors and compartments*. Eigenvector analysis[77] was performed separately for each chromosome arm using the 'eigs_cis' function at 10, 25, 50 and 250 kb, where gene density was used to 'phase' Eigenvectors, that is, Eigenvector tracks were 'flipped' if they anticorrelated the gene density track. The EV1 Eigenvectors (ones with the highest Eigenvalues, which typically correspond to Hi-C compartments) of select samples were saved as bigWig files to use in stackup analysis and visualization.

*Pairwise-class averaging via saddle plots*. To evaluate how different classes of genomic loci interact with each other in 3D, that is, saddle plot analysis[77], we used the 'saddle' function. First, assignment of the classes to genomic loci (for example, compartment status, CRE status and so on) was done on a bin level and passed as a 'track' parameter to the function. Second, the average level of interactions was calculated for each possible combination of classes from the flattened contact map (observed/expected). Finally, a class-pairwise average interaction matrix was constructed.

Assigning different classes to genomic bins was done in a specific manner. For published DLD-1 IPGs[80], we used the chromatin state assignments directly at 50 kb resolution after merging B2/3 and B4 heterochromatin classes into 'B'. Candidate CREs were hierarchically assigned to 10 and 25 kb genomic bins in the following order: PLS, pELS, dELS, H3K4me3-open, CTCF and finally, open sites (for example,

if a given 10 kb bin contained a pELS element, then the entire bin was assigned pELS status, regardless of other elements present in that bin). 'Continuous' EV1 tracks were digitized into 38 quantiles after excluding 2.5% of the extreme EV1 values from each end of the spectrum.

We estimated the strength of the A compartment as an enrichment of AA interactions over AB: AA / ((AB + BA) / 2), where AA is an average of observed–expected interactions between the EV1 quantiles with the 20% strongest A-compartment identity, and AB(=BA) is an average of the observed–expected interactions between the quantiles with 20% strongest A and B identities. Similarly for the B compartment, strength was estimated as: BB / ((AB + BA) / 2).

*Average pileup and quantification analysis.* The 'pileup' function was used to explore the local interaction pattern of a set of two-dimensional genomic features (defined by a pair of genomic locations, for example, in our case an all-by-all grid of MCDs, with their various subsets and a set of called loops). Briefly, local contact maps (normalized to the expected) centred on a given feature and with a fixed flank of 100 kb, a 'snippet', were extracted as a stack and then averaged all together or in groups of features that meet a certain criterion, for example, in our case subgroups of the microcompartment grid by genomic distance, subgroups of the grid overlapping other features like extrusion dots or domains and so on. *Cis*-chromosomal pileups were performed on 10 kb contact maps, whereas *trans*-chromosomal pileups were performed on 25 kb maps, unless specified otherwise.

Local interaction patterns, average pileups, were also used to quantify average strength of a given genomic feature. Both for loops and the grid of microcompartments, strength was defined as a ratio between the signal in the centre of the pileup and the signal in the periphery. Specifically, we used a 50 × 50 kb window in the centre together with four 60 × 60 kb corners (as a periphery signal) for the grid of MCDs in *cis*, a 25 × 25 kb centre (single pixel in the middle) and four 50 × 50 kb corners (periphery) for the grid of MCDs in *trans*, and a 30 × 30 kb centre and four 70 × 70 kb corners (periphery) for loop strength calculations.

*Loop calling and definition of extrusion domains.* To detect a reference set of extrusion loops for our analyses, we pooled control Hi-C data for 5 and 10 h at 10 kb resolution (given their similarity, as demonstrated in Supplementary Fig. 1c,d) to achieve higher sequencing depth and then used the 'dots' function to call significantly enriched interactions. We used 'cluster_filtering=False' and otherwise the default parameters to apply more stringent singleton filtering afterwards. Detected interactions were further filtered to ensure they were compatible with the convergent CTCF–CTCF interaction.

The resulting list of 18,615 interactions/loops were also used to define extrusion domains, intuitively, it is the outermost loop from a subgroup of nested loops that defines a domain. Specifically, loops were clustered by their anchor 1 to group those situated on the same extrusion line, then most upstream anchor 1 of the cluster was used together with the most downstream anchor 2 of the cluster to define such fully inclusive intervals. Nested and significantly overlapping (when the overlap between intervals was >70% of either of the intervals) intervals from the resulting list were merged, yielding the final list of 3,401 domains.

**Enrichment-based detection of microcompartment domains.** We defined MCDs as the genomic loci/anchors that give rise to the strongly interacting off-diagonal rectangular domains that are clearly visible on the 5 h RanGAP1-depletion contact map and set out to detect them using screening procedure akin to the detection of dots: detect enriched pixels that stand out relative to the local background and after grouping them by proximity, detect those groups that continue interacting with others across distances, that is, those that continue "checkering". Such checkering anchors are the target feature that we aimed to detect in the first place, that is the MCDs.

Next, we described specific steps involved in MCD detection. Variable size and shape of observed MCD–MCD interaction domains dictated the choice of convolution kernels that "describe" the local vicinity for a given group of pixels (Supplementary Fig. 2b): the vertical (V) kernel is meant to facilitate detection of horizontally elongated domains, whereas the horizontal (H)–vertically elongated domains, where a square-shaped group of pixels M (middle)–corresponds to the part of the domain being tested. Convolutional kernels M, V and H were swept across distance decay-corrected contact map (observed/expected) at 10 kb resolution, up to 30 MB for computational efficiency, and enriched pixels were selected with a simple thresholding approach: pixels for which the M was at least twice as bright as either the V or H. The density-based clustering approach 'OPTICS' from the sklearn package[121] (v.1.4.1; Supplementary Fig. 2c) was used to filter out singletons and small groups of enriched pixels, and preserve larger more robust groups of enriched pixels ('min_samples = 5' and 'max_eps = 33 kb' parameters were used). Pixels that remained after the clustering step were used to calculate the coverage of enriched pixels (or anchor valency; Supplementary Fig. 2d), and finally, we applied the 1D peak detection function 'find_peaks' of the scipy package[122] to the coverage track to detect prominent peaks (Supplementary Fig. 2e), which were treated as the final MCD anchors. Each anchor was characterized by its footprint interval and a summit–we used footprints for most of the downstream analysis, except for the average pileups and stack-ups, where we used the summits as an MCD genomic coordinate instead of the centre of the footprint. for example. The exact details of implementation are available on *GitHub* (https://github.com/dekkerlab/inherited-folding-programs.git).

We detected a total of 2,105 MCDs in the 5 h RanGAP1 depletion sample and 1,791 MCDs in the RanGAP1-depleted sample at cytokinesis, 68% of which were contained in the former sample. We used three subsets of MCDs for downstream analysis: 565 cytokinesis MCDs that did not overlap the 5 h ones ('Cyto-specifc'), 1,218 cytokinesis MCDs that were also present in G1 ('Cyto at G1' or 'Cyto + G1') and 876 G1 MCDs that were not present in the cytokinesis sample ('G1-specific').

We also identified 4,623 permissive MCDs by pooling the 5 h and 10 h G1 RanGAP1-depleted Hi-C samples—we refer to this set as Mega or permissive MCDs. This set generally demonstrated the same trends, albeit with the addition of weaker MCDs and potentially some erroneous MCD calls, and was only used for the spectral clustering characterization (details in the next section).

**Spectral clustering of RanGAP1-depletion Hi-C data.** Motivated by the recent progress in the application of spectral clustering[15,56] to the analysis of Hi-C data, we set out to apply this methodology specifically to the RanGAP1-depletion data. Such an analysis complements our enrichment-based MCD screening procedure, enabling us to test whether MCDs indeed demonstrate similar interaction patterns genome-wide.

To date, we performed spectral clustering of combined 5 h and 10 h G1 RanGAP1-depletion Hi-C data, following a previously described procedure[15], and here we provide a brief description of our implementation and highlight key differences. We start by combining RanGAP1 depletion G1 5 h and 10 h Hi-C samples at 10 kb resolution to reduce data sparsity. The combined contact frequency map is flattened by normalizing it to the expected: average contact frequency decay with distance for intra- and interchromosome arm regions and pairwise interchromosomal interaction levels for *trans* data[55]. Additional 'bad bins' corresponding to translocated loci were manually identified (Supplementary Table 6) and masked for further analysis to ensure leading Eigenvectors of the contact map explain compartmentalization patterns instead of the discordance between reference assembly and the state of the genome in DLD-1 cell line[15]. The pre-processed contact map was further balanced using an iterative correction procedure[77] and finally eigendecomposed.

This procedure differs from the published approach[15] by performing Eigendecomposition on a full genome-wide contact map that includes both *cis* and *trans* interactions. The primary motivation for this is to achieve Eigendecomposition at a high resolution (10 kb) relevant for MCD detection. Dense matrix Eigendecomposition at such resolution is challenging, as it would require approximately 1 TB of local memory. To overcome this, we used modified version of the publicly available prototype of sparse Hi-C contact map Eigendecomposition (https://github.com/open2c/open2c_vignettes/blob/main/sparse_eigendecomp.ipynb). This prototype leverages ARPACK[123] implementation of sparse matrix Eigendecomposition, available in SciPy[122] as the scipy.sparse.linalg.eigsh function. The sparse contact map was fed into the function as a scipy.sparse.linalg.LinearOperator vector function that defines the matrix-vector product between our sparse contact map and said vector.

Performing Eigendecomposition on combined *cis* and *trans* data presents a challenge for interpretability due to the variation in chromosome sizes, stark contrast in sparsity between *cis*- and *trans*-contact frequencies and potential differences in *cis* and *trans* compartmentalization patterns. However, in this particular case of RanGAP1-depleted Hi-C data, we did not observe obvious effects of chromosome size variation until Eigenvector 10 (sorted by importance, that is, absolute Eigenvalue) and confirmed that the leading non-trivial Eigenvector is highly correlated to a concatenated list of chromosome-wide EV1s calculated using the cooltools.eigs_cis procedure[55] at 10 kb (Extended Data Fig. 4a). Thus, we continued our downstream analysis with the 2–9 most significant Eigenvectors (Extended Data Fig. 4b), where E1 was excluded for clustering purposes as it is a trivial 'flat' Eigenvector corresponding to the sum of rows/columns of the contact map. Eigenvectors E2–E9 were unit-normalized, weighted by the square root of the corresponding Eigenvalue and subjected to *K*-means clustering as described previously[15]. The number of clusters *k* = *8* was chosen as it provided optimal overlap between the permissive set of MCDs and a given MCD-associated cluster, that is, >60% of the cluster is covered with MCDs and >60% of MCDs are covered with intervals from that cluster (holds true by counting both overlapping nucleotides and number of intervals). Imposing cluster-informed grouping and sorting onto epigenetic tracks (Extended Data Fig. 4c) reveals strong enrichment for active marks H3K27ac/H3K4me3 in the MCD-associated cluster C2. There is one apparent deviation from this trend we have described so far and it is the cluster most closely associated with the A1 IPG identified in wild-type DLD-1 cells[80]: this cluster harbours a subset (approximately 20%) of 'stubborn' MCDs that 'refuse' to cluster with the 'mainstream' MCD-associated cluster. The fate of this subset of MCDs from A1 IPG (most GC-rich, speckle-associated and early replicating[15]) requires further investigation and is out of the scope of this paper. Overall, we were able to capture the main trends and characteristics of MCDs as defined using our enrichment-based method; however, further development into clustering approaches might enable a more precise and comprehensive MCD detection.

**ATAC–seq analysis.** Batch pre-processing of raw ATAC–seq data from control and auxin-treated RanGAP1–AID cells synchronized in prometaphase or released for 5 h to early G1 was performed using the nf-core/atacseq pipeline (v.2.1.0; https://github.com/nf-core/atacseq )[108]. Adaptor-trimmed paired-end reads were mapped to the hg38 reference genome using bwa mem and filtered according to the standard pipeline for mapping quality, mitochondrial reads, PCR duplicates and read length (<2,000 bp). Similarity between replicates was confirmed using DESeq2 (ref. [124]) and filtered alignments were merged and de-duplicated across replicates. The summary mapping statistics of ATAC–seq samples generated in this study are in Supplementary Table 2. Fragment-length distributions were determined from binary alignment map (BAM) files using deepTools[125]. Read ends were derived from the filtered alignments (BAM files) and modified to account for tn5

by shifting ± reads by +4/−5 bp for use in all downstream applications. ATAC–seq coverage tracks were generated from shifted read ends using BEDtools[126] and scaled to $1 \times 10^6$ mapped reads.

We used MACS3[127] to find ATAC–seq peaks of accessibility in the pooled and single replicate datasets using default parameters with a shift/extend of −75/+150. Peaks called in pooled datasets were only considered true when they overlapped a peak in each of the two constituent replicates by at least 50%. To compare these pooled peaks between conditions, the union set was merged using bioframe[128].

**CUT&RUN analysis.** Batch pre-processing of raw CUT&RUN data from control and auxin-treated RanGAP1–AID cells synchronized in prometaphase or released for 5 h to early G1 was performed using the nf-core/atacseq pipeline (v.2.1.0; https://github.com/nf-core/atacseq)[108]. Adaptor-trimmed paired-end reads were mapped to the hg38 reference genome using bwa mem and filtered according to the standard pipeline for mapping quality, mitochondrial reads, PCR duplicates and read length (<2,000 bp). Similarity between replicates was confirmed using DESeq2 (ref. [124]). The summary mapping statistics of CUT&RUN samples generated in this study are in See Supplementary Table 2. Read ends were derived from the filtered alignments (BAM files) and extended ±25 bp for most visualizations. Coverage tracks were generated from read ends using BEDtools[126] and scaled to $1 \times 10^6$ mapped reads. Peaks of enriched H3K27ac signal were called from paired-end bedgraph files using SEACR[129] with normalization to cell cycle-matched IgG controls. Bookmarked peaks are considered the subset of control G1 peaks that overlap a peak called in the control prometaphase sample.

**SLAM–seq analysis.** SLAM–seq reads were aligned to Ensembl GRCh38v95 with Hisat-3N[130], specifying T>C conversions. T>C conversions in sequenced reads were obtained by filtering Hisat-3N-aligned BAM files for reads with a Yf:i tag (corresponding to the number of T>C converted nucleotides) greater than one and thereafter annotated as nascent. Conversion efficiency was confirmed in ERCC spike-ins by obtaining the proportion of reads with T>C substitutions in 4sU-labelled versus unlabelled ERCC species. Global conversion efficiency was estimated by calculating the percentage of total RNA reads harbouring T>C substitutions in labelled and unlabelled samples. GRAND-SLAM was used to estimate the RNA NTR of each gene[63] and genes with at least 100 reads were retained for further analyses. Fisher's exact tests were used to test for significant associations between genes expressed in different stages of mitotic exit and genes found in MCDs (alternative = greater); the resulting *P* values were corrected using the Benjamini–Hochberg method.

**Statistics and reproducibility.** No statistical method was used to pre-determine sample size in any experiments. Two replicates for each condition were performed for most experiments (Hi-C, ATAC–seq, CUT&RUN and SILAC LC–MS). For SLAM–seq, three independent experiments were performed. Insulation tracks and *cis* Eigvectors were used to assess the reproducibility of the Hi-C replicates (Supplementary Fig. 2c,d), which were pooled for most of the presented plots and analyses (Supplementary Fig. 2a). ATAC–seq similarity between replicates was confirmed using DESeq2 (ref. [124]) and filtered alignments were merged and de-duplicated. All conclusions were verified in individual replicates. Unless otherwise indicated, all immunofluorescence (Figs. 1c,e, 6c, 8e and Extended Data Figs. 2a,c,e–g, 8c, 9d) and western blot images (Figs. 1b, 5e and Extended Data Figs. 1a,b, 9b) are representative of at least two independent replicates, and flow cytometry panels are representative of at least three independent experiments (Fig. 1d and Extended Data Fig. 2b). Nuclear import quantification (Fig. 1g) and images (Extended Data Fig. 2g) are representative of two independent experiments. No data were excluded from these analyses, and randomization or blinding of experiments was not applicable.

## Reporting summary

Further information on research design is available in the Nature Portfolio Reporting Summary linked to this article.

## Data availability

Any requests regarding cell lines and plasmids should be directed to Mary Dasso's laboratory. The datasets generated in this publication have been deposited in the NCBI GEO as a SuperSeries accessible through the accession number GSE278023, consisting of GSE277875 (Hi-C), GSE277731 (ATAC–seq), GSE308844 (CUT&RUN) and GSE309609 (SLAM–seq). The following published datasets were used in this study (Supplementary Table 3): GSE132363, GSE178593 and GSE214012. The mass spectrometry proteomics data have been deposited to the ProteomeXchange Consortium via the PRIDE partner repository with the dataset identifier PXD056346. Source data are provided. All other data supporting the findings of this study are available from the corresponding author on reasonable request.

## Code availability

Open2C scripts and notebooks used in this study are publicly available in *GitHub*: https://github.com/open2c and https://github.com/dekkerlab/inherited-folding-programs. No other customized codes were developed for this study.

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

## Acknowledgements

We thank the members of the Dekker laboratory, the Open2C community (especially N. Abdennur), for discussion on experiments and data analysis, and C. Navarro for discussion and help revising and editing the paper. We thank the UMass Electron Microscopy Core (K. Reddig and G. Hendricks, supported award numbers S10OD025113-01 and S10OD021580 from the National Center For Research Resources),

the UMass Proteomics Core, the UMass FACS core and the UMass Deep Sequencing Core (E. Kittler, M. Zapp and D. Wilmot). We thank R. Kaufhold for help with constructing the MBP–mScarlet–NLS construct. This work was supported by grants from the National Human Genome Research Institute (grant numbers HG003143 and HG011536 to J.D. and HG012967 to A.A.P.) and the National Institute for General Medical Sciences (R35GM133762 to A.A.P.). J.D. is an investigator of the Howard Hughes Medical Institute. V.A. and M.D. were supported by the Intramural Research Program of the Eunice Kennedy Shriver National Institute of Child Health and Human Development at the National Institutes of Health, USA (grant number ZIAHD008954). This research was in part supported by the Intramural Research Program of the National Institutes of Health (NIH). The contributions of the NIH authors are considered works of the United States Government. The findings and conclusions presented in this paper are those of the authors and do not necessarily reflect the views of the NIH or the US Department of Health and Human Services. J.W.L. was supported by the National Institute of Allergy and Infectious Diseases (grant number F31AI189160). E.N. was supported by the Graduate Research Fellowship Program from the National Science Foundation.

## Author contributions

A.S. and J.D. conceived and designed the study. All cell lines were designed and engineered in the M.D. laboratory. V.A. re-designed NUP93– and RanGAP1–AID cell lines, and performed nuclear import assays and analysis. A.S. performed Hi-C, epigenomic profiling assays, proteomics and all experiments except for the nuclear import assays. S.V.V. analysed Hi-C data, developed analytical methods and supported all other analyses. A.S. analysed Hi-C, ATAC–seq and other relevant datasets. J.W.L. and A.S. performed nascent RNA labelling and SLAM–seq experiments, which was analysed by J.W.L. and A.A.P. E.N. performed preliminary analysis of RanGAP1-depletion Hi-C. A.S. and J.D. wrote the paper with input from S.V.V., V.A., J.W.L., E.N., M.C.D. and A.A.P.

## Competing interests

J.D. is a member of the scientific advisory board of Arima Genomics, San Diego, CA, USA and Omega Therapeutic, Cambridge, MA, USA. J.D. is inventor on patent application US 12,146,186 B2, held by the University of Massachusetts Chan Medical School, Harvard College, the Whitehead Institute for Biomedical Research, and the Massachusetts Institute of Technology, which covers Hi-C technology. The other authors declare no competing interests.

## Additional information

**Extended data** is available for this paper at https://doi.org/10.1038/s41556-025-01828-1.

**Correspondence and requests for materials** should be addressed to Job Dekker.

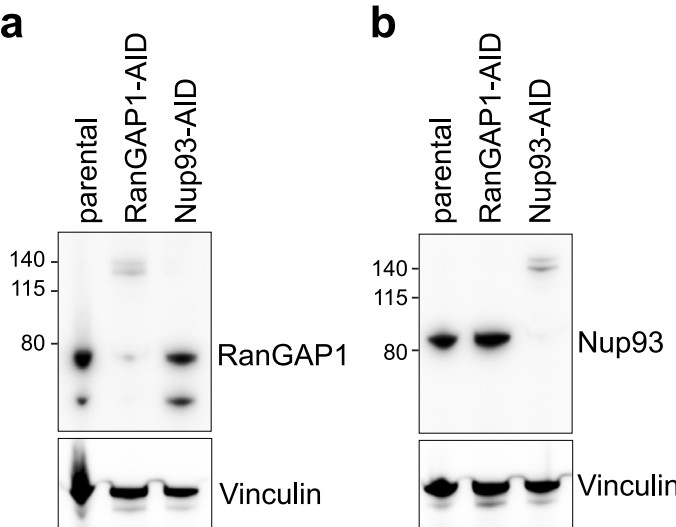

**Extended Data Fig. 1 | Endogenous protein levels of wt and AID-tagged RanGAP1 and Nup93. a,b**, Representative western blot images of whole-cell lysates derived from parental, RanGAP1–AID, and Nup93–AID DLD-1 cell lines demonstrating relative size and abundance of wt and AID-tagged RanGAP1 (**a**) or AID-tagged Nup93 (**b**). Vinculin was probed as a loading control. Unprocessed blots are available.

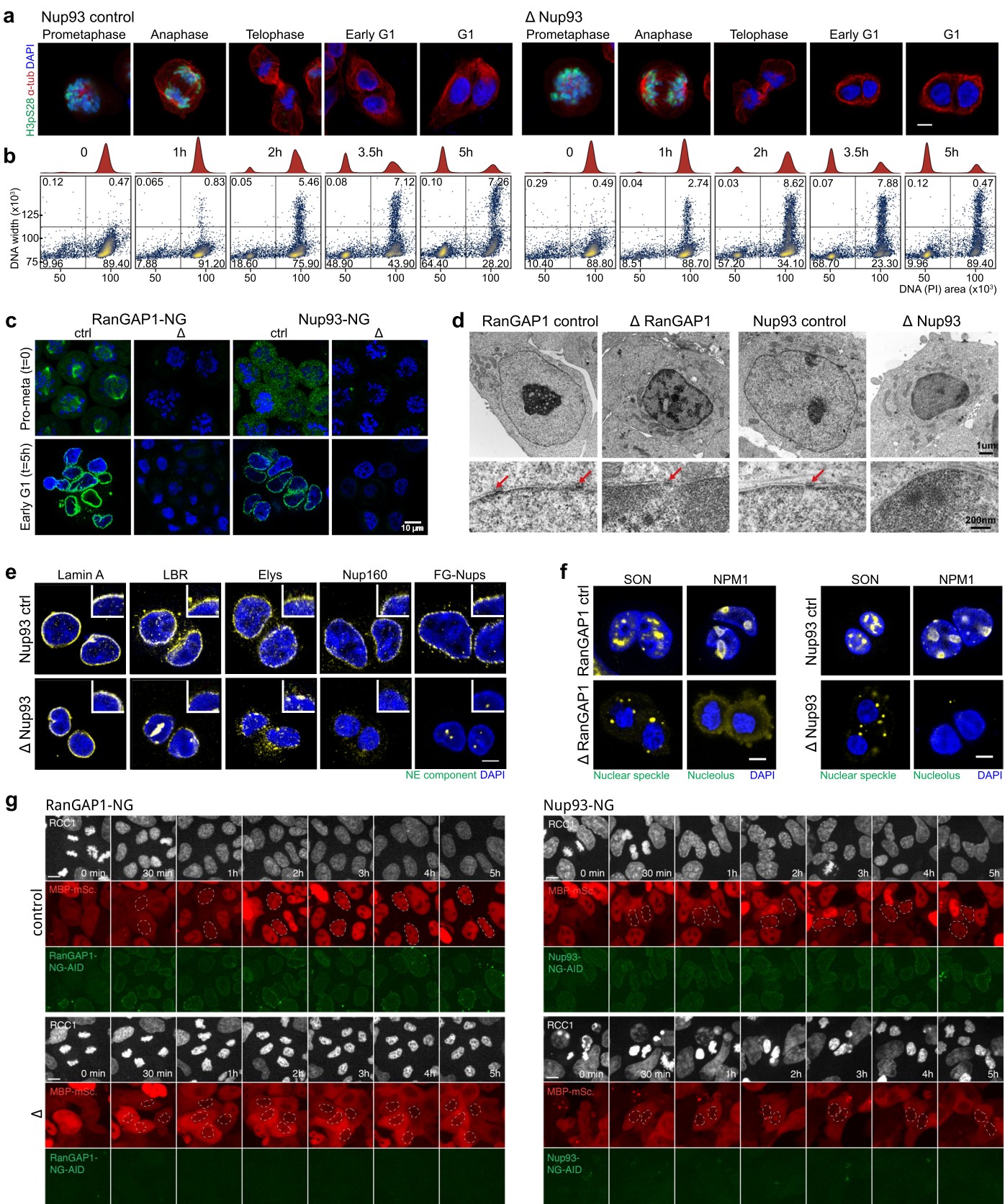

**Extended Data Fig. 2 | See next page for caption.**

**Extended Data Fig. 2 | Acute depletion of RanGAP1–AID or Nup93–AID during mitotic exit enables the assembly of daughter nuclei isolated from the G1 cytoplasm. a**, Representative immunofluorescence images for Nup93 control and depleted cells. Loss of Histone H3 serine 10 phosphorylation (H3pS10P) and DNA content at indicated time points indicates similar mitotic exit kinetics in the presence or absence of Nup93. Scale bar, 5 um. **b**, DNA-content flow cytometry measurements for Nup93 control and depleted cells indicating similar mitotic exit kinetics in the presence or absence of Nup93. **c**, Representative fluorescent images of endogenous NeonGreen-tagged RanGAP1–AID or Nup93–AID demonstrate efficient degradation following 2 h auxin treatment during mitotic arrest and 5 h release to G. Scale bar, 10 um. **d**, Representative transmission electron micrographs of RanGAP1–AID- or Nup93–AID-depleted cells fixed in G1 reveal relatively small nuclei with hyper-condensed chromatin. Arrows indicate nuclear pores embedded in the double lipid bilayers of the nuclear envelope, which are not found in Nup93-depleted nuclei. **e**, Representative immunofluorescence images of Nup93 control and depleted cells 5 h after mitotic release demonstrating presence of nuclear lamina (lamin A) and nuclear envelope (LBR) proteins as well as the DNA-binding nuclear pore complex protein, Elys, in the absence of Nup93. Structural (Nup160) and late associating (FG-Nups) nucleoporins are not found in Nup93-depleted nuclei. Scale bar, 5 um, or 1 um (inset). **f.** Nuclear speckle (SON) and nucleolar (NPM1) resident proteins are mis-localized to the cytoplasm in RanGAP1–AID and Nup93–AID-depleted nuclei. Representative immunofluorescence images of cells fixed 5 h after mitotic exit are shown. Scale bar, 5 um. **g**, Representative time-lapse fluorescent images of RanGAP1-NG-AID or Nup93-NG-AID control and depleted cells released from Ro-3306 for 5 h. Endogenously tagged RCC1-RFP demarcating chromatin and the MBP–mScarlet-NLS import substrate are shown.

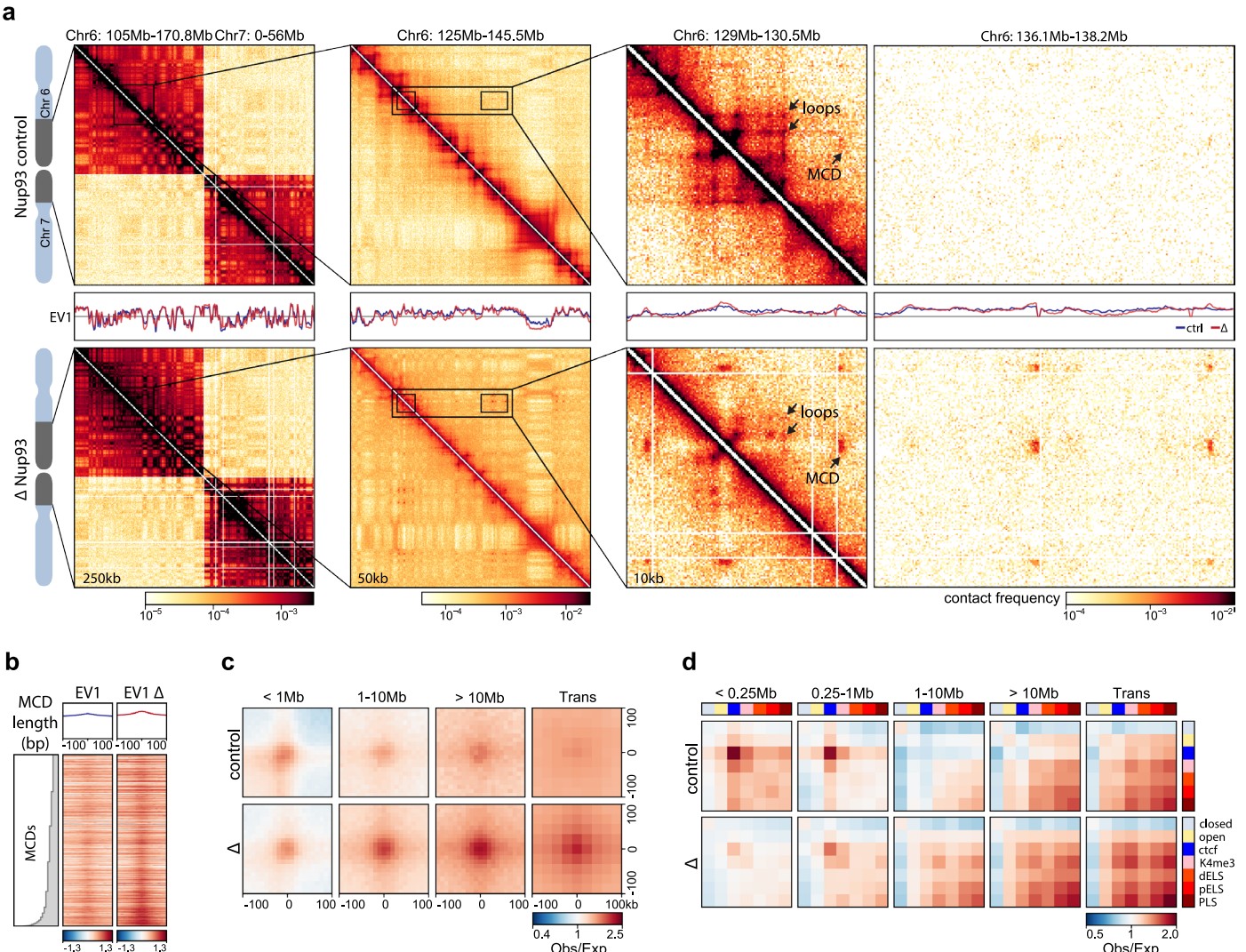

**Extended Data Fig. 3 | A chromosome-intrinsic capacity to form a kilobase scale cCRE compartment emerges upon depletion of Nup93 during mitotic exit. a**, Representative Hi-C interaction frequency maps at 250 kb (Chr 6: 105–170.8 Mb–Chr 7: 0–56 Mb), 50 kb (Chr 6: 125–145.5 Mb), and 10 kb (Chr 6: 129–130.5 v. 136.1–138.2 Mb) resolution showing genome compartmentalization in control and auxin-treated DLD-1 Nup93–AID cells released from prometaphase arrest for 5 h. First eigenvector (EV1) values for *cis* interactions are phased by gene density. Examples of canonical loops and MCDs present in the Nup93 depletion are indicated by arrows. **b**, Heat maps for all RanGAP1 MCDs centred on contact frequency summits with 100 kb flanking regions and sorted by anchor length demonstrate differences in intra-chromosomal EV1 values derived from 10 kb matrices in control and Nup93-depleted cells. **c**, Pairwise mean observed/expected contact frequency at 10 kb between all RanGAP1 MCDs projected in *cis* and *trans* showing enhanced interactions at all length scales and between chromosomes in Nup93-depleted G1 cells. **d**, Pairwise aggregate observed/expected contact frequency between cCREs assigned in control cells and subjected to hierarchical binning at 10 kb resolution showing enhanced homo- and heterotypic interactions between active promoters (PLS) and enhancers (promoter-proximal:pELS, promoter-distal:dELS) in Nup93-depleted cells compared to controls at multiple genomic distances in *cis* and in *trans*.

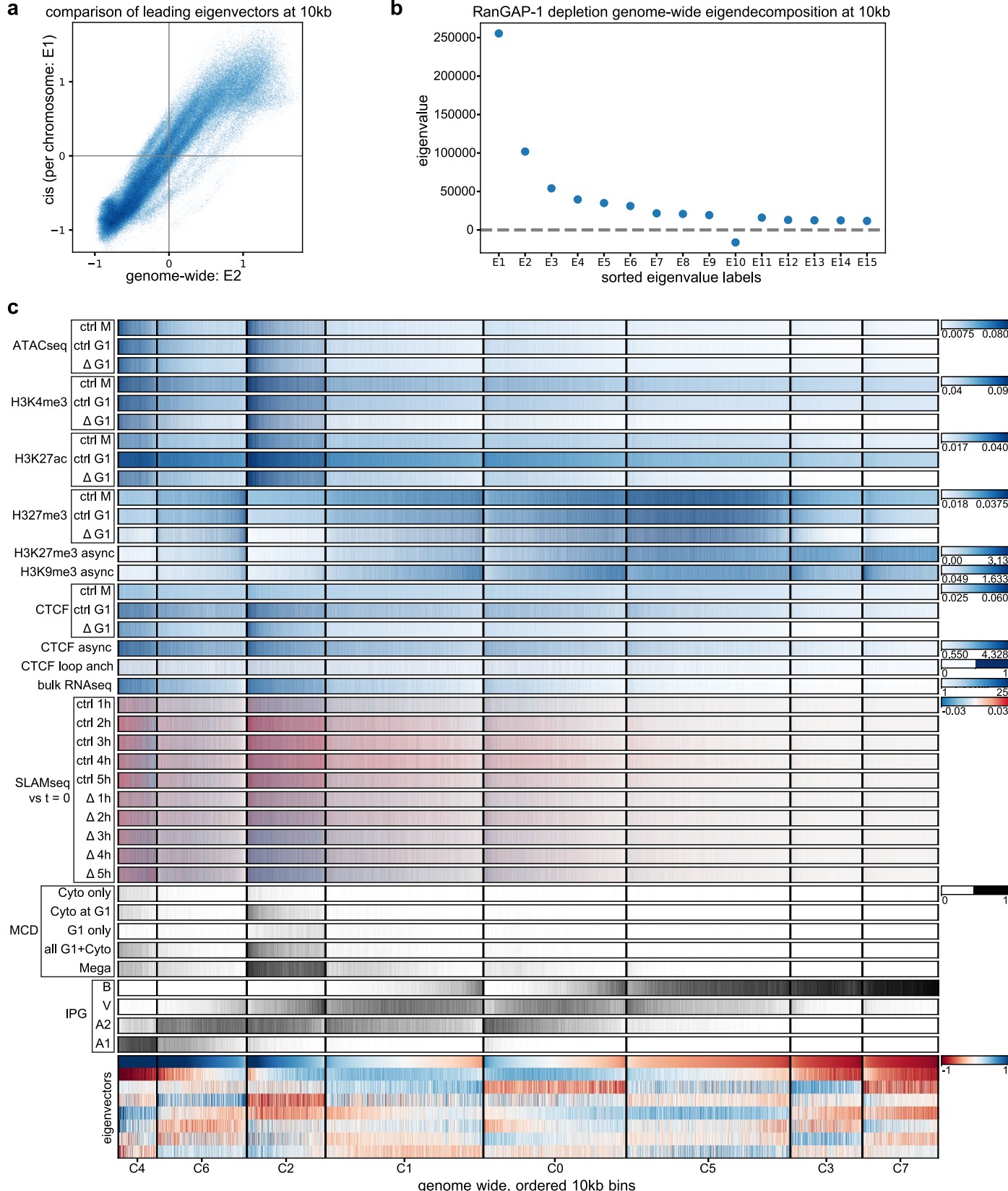

**Extended Data Fig. 4 | See next page for caption.**

**Extended Data Fig. 4 | Spectral clustering of RanGAP1-depletion Hi-C data identifies distinct interaction profiles at MCDs. a**, Scatter plot between genome-wide Eigenvector E2 (derived from spectral clustering analysis) on x-axis, and chromosome-level cis-eigenvectors E1 concatenated together for all autosomal chromosomes on y-axis. **b**, Eigenvalues derived from spectral clustering analysis of RanGAP1-depleted Hi-C data sorted by their absolute values. E2–E9 were used in the clustering analysis. **c**, Overview of spectral clustering of Hi-C data from 5 h G1 RanGAP1-depleted cells binned at 10 kb showing enrichment of MCDs in a specific chromatin profile. Heat maps of mean signal intensity for various functional genomics features binned the same way from top to bottom: ATAC–seq, H3K4me3, H3K27ac, H3K27me3 and CTCF

CUT&RUN, extrusion dot anchors ("loop anchors" detected in control G1 Hi-C data), bulk asynchronous cell RNA-seq coverage, SLAM–seq derived nascent RNA coverage over time (vs time 0, -/+ RanGAP1), MCD density (Cyto-specific, shared between Cyto and G1, G1-specific, all Cyto and G1 combined, permissive set of MCDs detected in a combined 5 h and 10 h G1 RanGAP1 depletion sample ("Mega")), and DLD-1 IPGs (B-combined, V-combined, A2 and A1). Weighted eigenvectors E2–E9 are depicted below the epigenetic marks. Different clusters are demarcated by black vertical lines on each track and the EVs heat map. Clusters are ordered by their average E2, and sorted by E2 within each cluster. Loci corresponding to masked "bad bins", chrX, chrY and chrM are omitted for clarity.

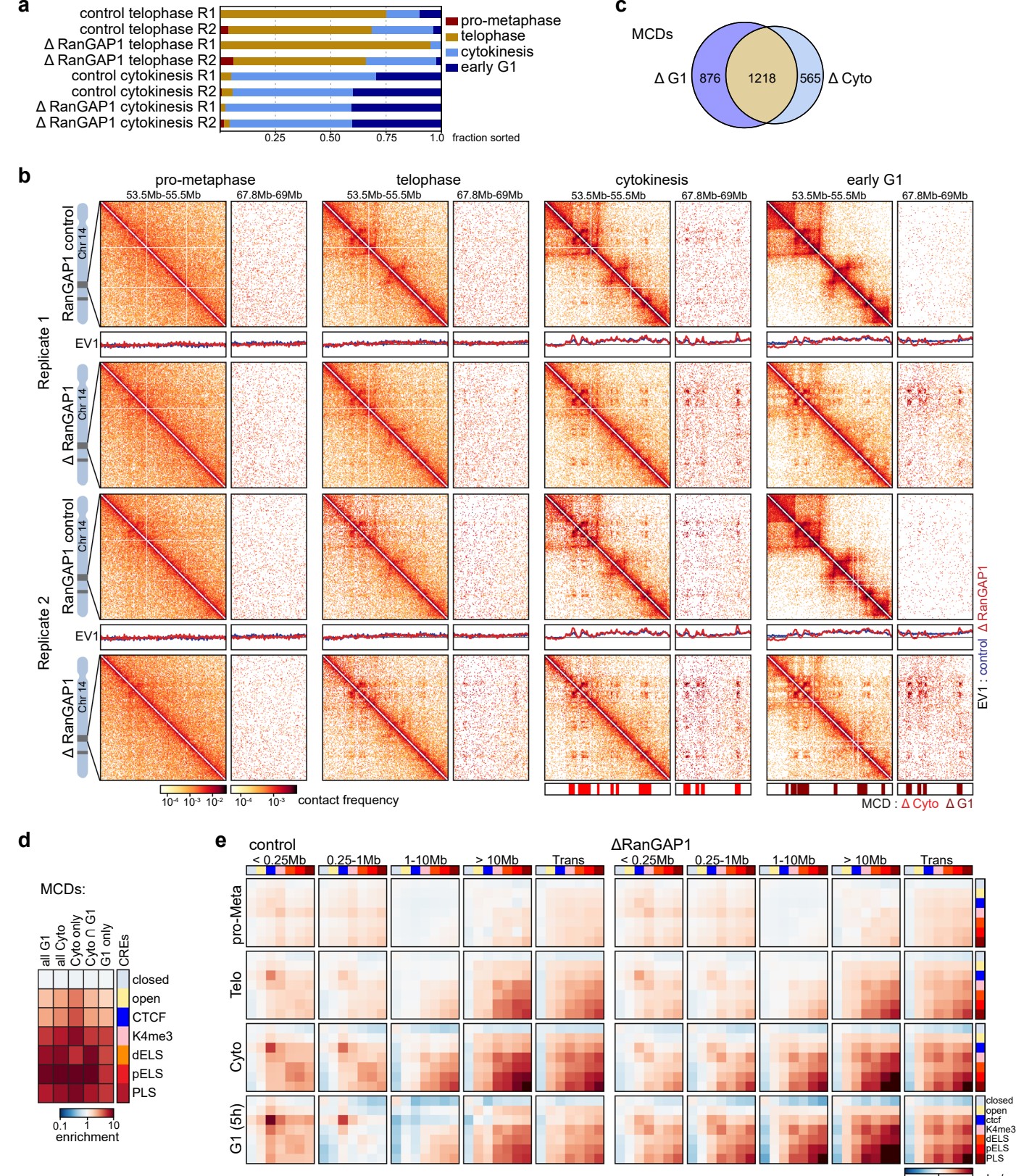

**Extended Data Fig. 5 | See next page for caption.**

**Extended Data Fig. 5 | *Cis*-regulatory elements form a transient microcompartment during mitotic exit. a**, Cell-cycle staging, based on DAPI and tubulin immunofluorescent morphology, for control and RanGAP1-depleted cell populations isolated by FACS 1.25 or 1.5 h after prometaphase release. Manual scoring of at least 25 cells in each of two replicates per condition demonstrates the successful enrichment of telophase or cytokinesis cells. **b**, Overlap of microcompartment domains identified in either cytokinesis or G1 RanGAP1-depleted cells specifies a majority of shared (1,218) MCDs as well as cytokinesis- (565) or G1-specific (876) subsets. **c**, Representative Hi-C interaction frequency maps at 10 kb (Chr 14: 53.5–55.5 v. 67.8–69 Mb) resolution showing genome organiszation in individual replicates of control and auxin-treated DLD-1 RanGAP1–AID cells enriched in prometaphase, telophase, cytokinesis, or early G1 (5 h after release). Matched first eigenvector (EV1) values for *cis* interactions are phased by gene density (A > 0). MCDs detected in RanGAP1-depleted cytokinesis- or G1-sorted cell populations are shown. **d**, Relative fold enrichment of control cCREs at cytokinesis-specific, cytokinesis-G1 shared, or G1-specific MCDs demonstrating the predominance of active promoters and enhancers particularly at the shared domains. **e**, Pairwise aggregate observed/expected contact frequency between cCREs assigned in control cells and subjected to hierarchical binning at 10 kb resolution showing transient enhanced homo- and heterotypic interactions between active promoters (PLS) and enhancers (promoter-proximal:pELS, promoter-distal:dELS) at multiple genomic distances in *cis* and in *trans* during mitotic exit. Ctrl, control; cyto, cytokinesis; pro-Meta, prometaphase; telo, telophase. Source numerical data are provided.

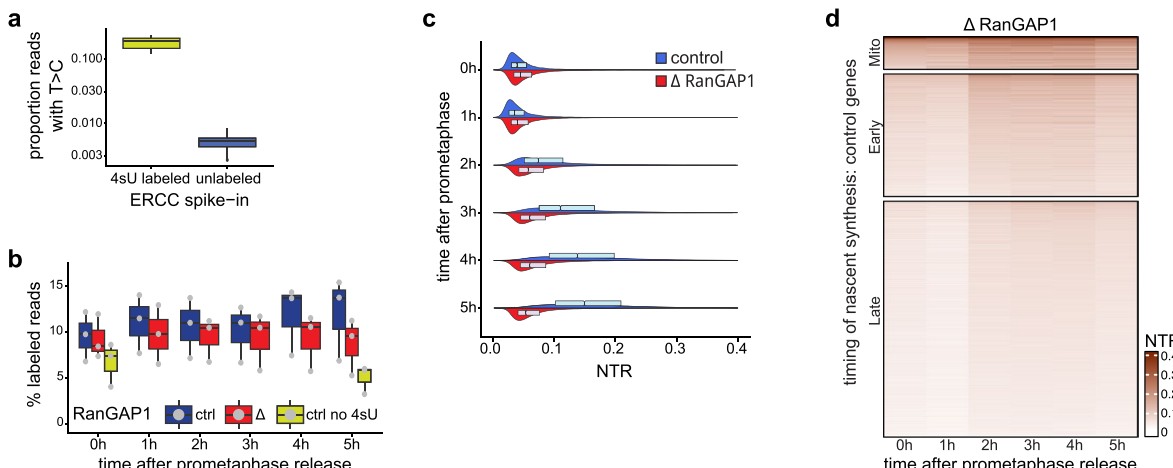

**Extended Data Fig. 6 | Nascent transcription reactivation during mitotic exit in the presence or absence of nuclear import establishment. a**, Proportion of uniquely aligned SLAM–seq reads mapping to ERCC spike-ins that contain greater than one T>C substitution. Reads from three ERCC species in vitro transcribed with 10% 4sU have significantly more T>C substitutions (left) than reads mapping to 3 ERCC species with no 4sU-molecules (right) (Welch's two-sample t-test; p < 2.2 × 10⁻¹⁶). Boxplots indicate median and interquartile range of NTR fold changes for three independent SLAM–seq libraries per time point. **b**, Percentage of uniquely- aligned reads per SLAM–seq sample that contain greater than one T>C substitutions. 4sU-labelled samples have significantly more substituted reads than unlabelled controls (Welch's two-sample t-test; p = 0.0017). Data shown across time points, with three independent SLAM–

seq libraries per time point and condition. Boxplots indicate median and interquartile range of NTR fold changes for three independent SLAM–seq libraries per time point. **c**, Global distributions of mean new-to-total RNA ratios (NTRs) from SLAM–seq libraries estimated using GRAND-SLAM. Control samples show increased transcriptional activity following mitotic exit relative to RanGAP1-depleted samples. Boxplots indicate median and interquartile range of NTR fold changes for three independent SLAM–seq libraries per time point. **d**, Heat map of gene timing classification derived on control sample NTRs, but shown for NTRs observed for those genes in RanGAP1-depleted samples. NTRs from depletion conditions reveal a temporally disrupted gene expression program. Source numerical data are provided.

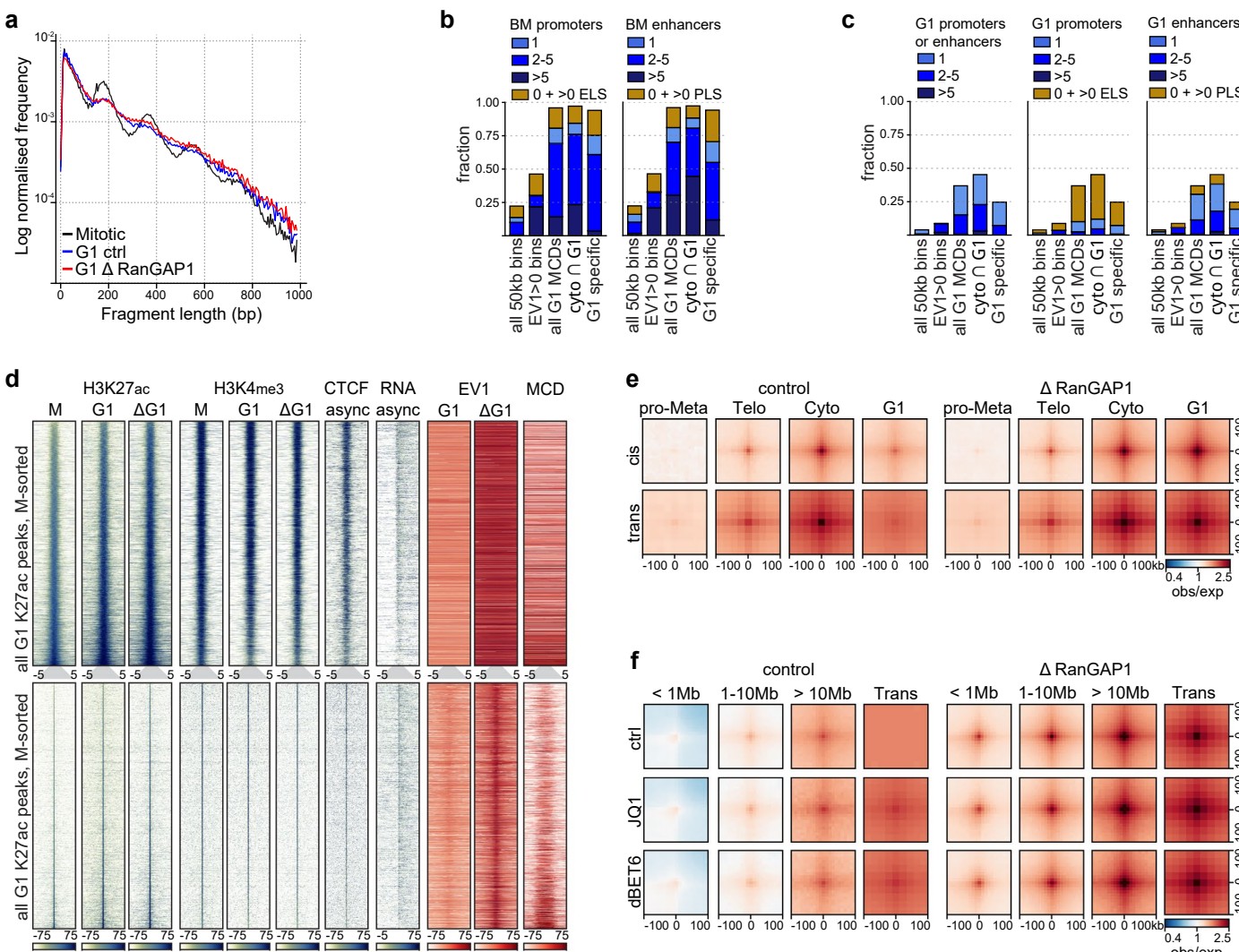

**Extended Data Fig. 7 | Chromatin accessibility at microcompartment domains during mitosis and G1. a**, Relative fragment-length distributions of ATAC–seq reads indicating regular nucleosome positioning in mitotically arrested cells and more dynamic G1 architecture in control and RanGAP1-depleted cells released for 5 h. **b**, Prevalence of bookmarked CREs based on control prometaphase and G1 ATAC–seq, H3K4me3, and H3K27ac coverage. The fraction of 50 kb bins genome-wide or having EV1 > 0, as well as all G1, shared G1-Cyto, or G1-specific MCDs, which overlap the indicated valency of either promoter or enhancer elements individually are shown. **c**, Prevalence of G1-specific CREs based on control and RanGAP1-depleted G1 ATAC–seq, H3K4me3, and H3K27ac coverage. The fraction of 50 kb bins genome-wide or having EV1 > 0, as well as all G1, shared G1-Cyto, or G1-specific MCDs, which overlap the indicated valency of promoter and enhancer elements (left) or either promoter (PLS) or enhancer (ELS) elements individually are shown. **d**, Heat maps centred on 5,202 peaks of enriched H3K27ac signal called in the G1 control with 5 kb (top) or 75 kb

flanking regions and sorted by strength of the prometaphase H3K27ac signal demonstrating prevalence MCDs and of elevated intra-chromosomal EV1 values from 10 kb matrices in control and RanGAP1-depleted cells. H3K27ac and H3K4me3 coverage in prometaphase and G1 control or RanGAP1-depleted cells, as well as CTCF and RNAseq from asynchronous cells are shown.
**e**, Pairwise mean observed/expected contact frequency between 5,202 mitotically bookmarked peaks of enriched H3K27ac signal projected in *cis* (10 kb) and *trans* (25 kb) demonstrating enhanced interactions in telophase that peak around cytokinesis of mitotic exit in control cells and continue to strengthen in the absence of RanGAP1. **f**, Pairwise mean observed/expected contact frequency between bookmarked H3K27ac peaks in control and RanGAP1-depleted cells demonstrating unchanged cell cycle-dependent interaction strength upon treatment with JQ1 or dBET6, to inhibit or degrade BET proteins, respectively. Ctrl, control; cyto, cytokinesis; pro-Meta/M, prometaphase; telo, telophase. Source numerical data are provided.

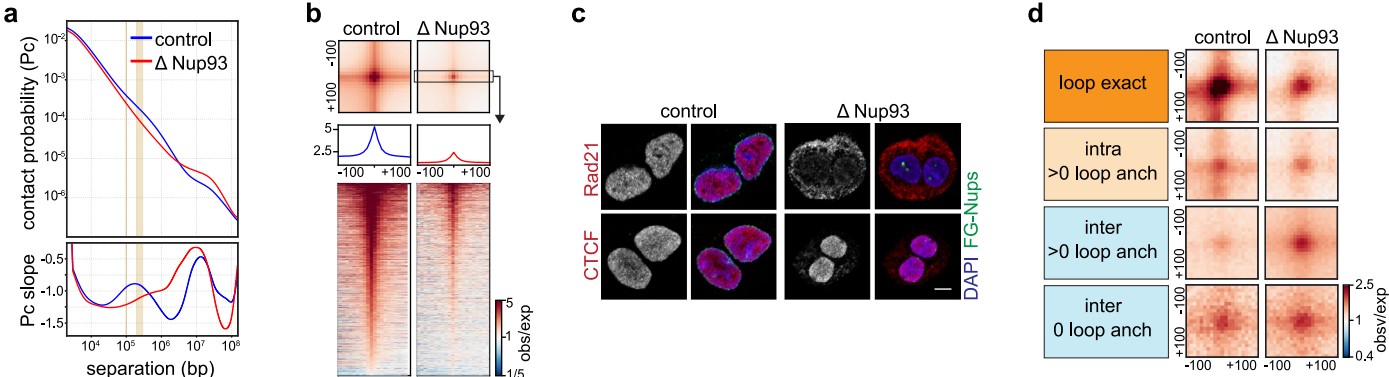

**Extended Data Fig. 8 | Microcompartment pruning generally requires active nuclear transport in G1. a**, P(s) and derivative P(s) plots for Hi-C data from FACS-sorted early G1 (t = 5 h) control and Nup93-depleted cells. **b**, Pairwise mean observed/expected Hi-C contact frequency between convergent CTCF loops identified in pooled interphase RanGAP1–AID control Hi-C data demonstrating Nup93-dependent looping interactions in early G1 (t = 5 h). Average signal for three central 10 kb bins across the 200 kb CTCF motif-centred window and stack-ups sorted by G1 loop strength are shown. **c**, Representative immunofluorescence images (at least two independent experiments) of Nup93–AID control and depleted cells fixed 5 h after mitotic release demonstrating the nucleocytoplasmic localization of the cohesin complex subunit, Rad21, and boundary transcription factor, CTCF. Scale bar, 5 um. **d**, Pairwise mean observed/expected contact frequency between MCDs in early G1 (t = 5 h) in control and RanGAP1-depleted cells. MCD–MCD contacts are categorized by the presence of a convergent CTCF loop or loop anchor (>0) and the looping domain status of the constituent MCDs, as indicated.

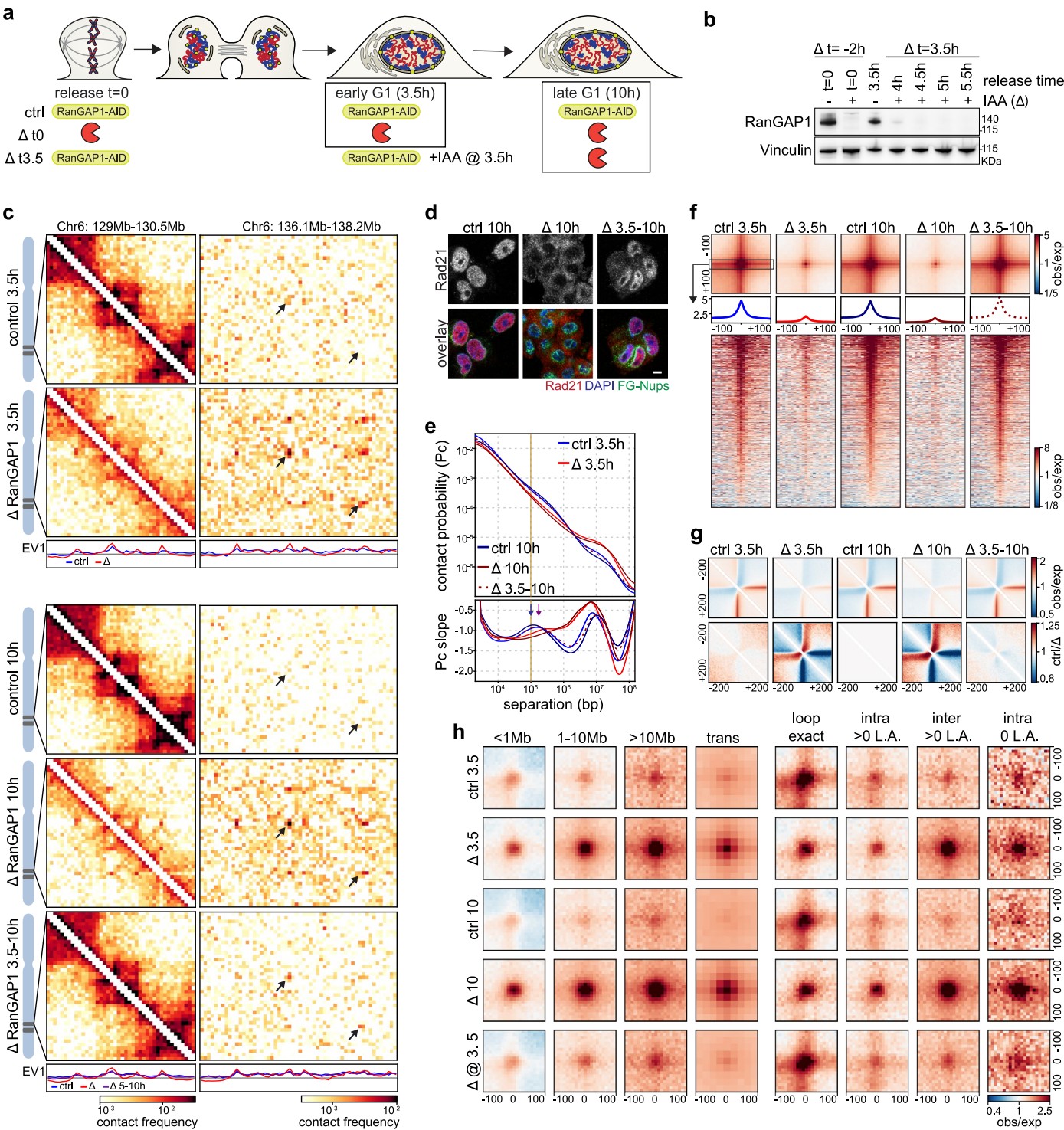

**Extended Data Fig. 9 | Nuclear transport modulates microcompartment pruning in G1 cells. a**, Workflow for RanGAP1–AID depletion in G1. Control and RanGAP1-depleted cells were fixed at t = 3.5 and 10 h and compared to G1 depletion from 3.5–10 h. **b**, Representative western blot images of whole-cell lysates collected every 30 min following G1 depletion at t = 3.5 h. **c**, Hi-C interaction frequency maps at 10 kb resolution (Chr 6: 129.1–130.5 & 136.1–138.2 Mb) for RanGAP1–AID control and depleted cells released from prometaphase for 3.5 or 10 h or G1 depletion from 3.5–10 h. EV1 values phased by gene density (A > 0). Arrows indicate MCD–MCD interactions detected in G1 RanGAP1 depletion. **d**, Representative immunofluorescence images of RanGAP1–AID control and mitotic (t = 0) or early G1-depleted (t = 3.5 h) cells fixed 10 h after mitotic release demonstrating nucleocytoplasmic localization of Rad21. Scale bar, 5 um. **e**, P(s) plots for Hi-C data from RanGAP1–AID control

and mitotic (t = 0) or early G1-depleted (t = 3.5 h) cells 10 h after mitotic release. Vertical line indicates average control loop size, which is increased (arrows) in RanGAP1–AID-depleted cells. **f**, Pairwise mean observed/expected Hi-C contact frequency in late G1 (t = 10 h) at convergent CTCF loops, demonstrating retained loops in early G1 (t = 3.5 h) RanGAP1–AID-depleted cells. Average signal for three 10 kb bins across the 200 kb CTCF motif-centred window and stack-ups sorted by G1 loop strength are shown. **g**, Observed/expected Hi-C interaction pileups at forward-oriented loop anchor CTCF motifs plotted in a 400 kb window at 10 kb resolution. Ratios for depleted vs control cells are shown **h**, Mean observed/ expected MCD–MCD contact frequency 10 h after prometaphase release control and RanGAP1–AID-depleted (at t = 0 or 3.5 h) cells. Contacts are categorized by distance or the presence of a convergent CTCF loop or loop anchor ("≥0") and looping domain status. Unprocessed blots are provided.

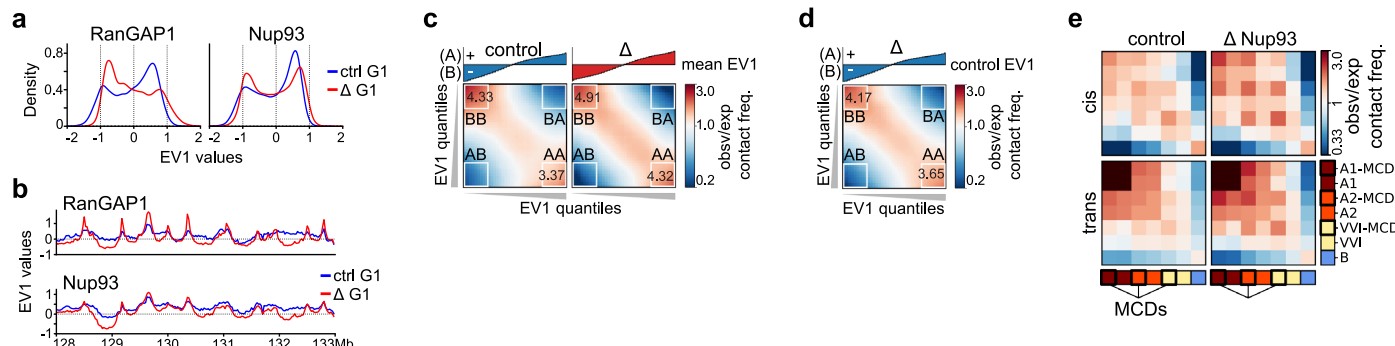

**Extended Data Fig. 10 | A distinct microcompartment is formed in the absence of RanGAP1 or Nup93. a**, Distributions of EV1 values from Eigenvector decomposition of 25 kb binned Hi-C data from control and auxin-treated RanGAP1–AID and Nup93–AID cells in G1. **b**, Representative examples of 25 kb EV1 tracks from the Hi-C data of control and RanGAP1–AID- or Nup93–AID-depleted cells in early G1. **c**, Saddle plots representing the segregation of active (A) and inactive (B) chromatin compartments in *cis* for control and Nup93–AID-depleted cells 5 h after mitotic release (G1). The first Eigenvector from each condition was used to rank 25 kb genomic bins and quantification of the average preferential A–A and B–B interactions for the top 20% strongest A and B loci are indicated. **d**, Saddle plots representing the segregation of active (A) and inactive (B) chromatin compartments defined in control cells at 25 kb resolution and plotted for Nup93–AID-depleted cells 5 h after mitotic release (G1). Quantification of the average preferential A–A and B–B interactions for the top 20% strongest A and B loci are indicated. **e**, Pairwise aggregate observed/expected contact frequency between IPGs derived from DLD-1 cells and further categorized by the presence or absence of MCDs showing enhanced homotypic interactions between MCDs in Nup93–AID-depleted cells in early G1.

# Reporting Summary

## Statistics

For all statistical analyses, confirm that the following items are present in the figure legend, table legend, main text, or Methods section.

| n/a | Confirmed | |
|---|---|---|
| ☐ | ☒ | The exact sample size (*n*) for each experimental group/condition, given as a discrete number and unit of measurement |
| ☐ | ☒ | A statement on whether measurements were taken from distinct samples or whether the same sample was measured repeatedly |
| ☐ | ☒ | The statistical test(s) used AND whether they are one- or two-sided *Only common tests should be described solely by name; describe more complex techniques in the Methods section.* |
| ☒ | ☐ | A description of all covariates tested |
| ☒ | ☐ | A description of any assumptions or corrections, such as tests of normality and adjustment for multiple comparisons |
| ☐ | ☒ | A full description of the statistical parameters including central tendency (e.g. means) or other basic estimates (e.g. regression coefficient) AND variation (e.g. standard deviation) or associated estimates of uncertainty (e.g. confidence intervals) |
| ☐ | ☒ | For null hypothesis testing, the test statistic (e.g. *F*, *t*, *r*) with confidence intervals, effect sizes, degrees of freedom and *P* value noted *Give P values as exact values whenever suitable.* |
| ☒ | ☐ | For Bayesian analysis, information on the choice of priors and Markov chain Monte Carlo settings |
| ☒ | ☐ | For hierarchical and complex designs, identification of the appropriate level for tests and full reporting of outcomes |
| ☒ | ☐ | Estimates of effect sizes (e.g. Cohen's *d*, Pearson's *r*), indicating how they were calculated |

*Our web collection on statistics for biologists contains articles on many of the points above.*

## Software and code

Policy information about availability of computer code

| Data collection | NovaSeq Control Software 1.8.0 (Hi-C) and Illumina BaseSpace/DRAGEN secondary analysis v4.0 (ATACseq, Cut&Run, SLAMseq) |
|---|---|
| Data analysis | Code used in this study can be found on Github:<br>https://github.com/dekkerlab/inherited-folding-programs<br><br>We also used the following scripts/software:<br><br>https://github.com/open2c/distiller-nf v0.3.4<br>https://github.com/open2c/pairtools v1.0.2<br>https://github.com/open2c/cooler v0.8.11<br>https://github.com/open2c/cooltools v0.7.0<br>https://github.com/open2c/bioframe v0.3.1<br>https://github.com/open2c/coolpuppy v0.9.5<br>pybbi v0.3.2<br>clodius v0.3.5<br>bedtools v2.29.2<br>matplotlib v3.5.2<br>scaffold v5.3.3<br>MaxQuant v2.6.5.0<br>nf-core/ataqseq v2.1.0<br>nf-core/RNAseq v3.15.0 |

```
https://github.com/loosolab/TOBIAS
macs3 v3.0.2
scikit-learn v0.23.0
scipy v1.5.2
R v4.1.0
DESeq2 v3.15
Image Lab 6.0.1 builder 34
FlowJo v10
HISAT-3N

https://github.com/open2c/open2c_vignettes/blob/main/sparse_eigendecomp.ipynb
https://github.com/nf-core/atacseq
```

For manuscripts utilizing custom algorithms or software that are central to the research but not yet described in published literature, software must be made available to editors and reviewers. We strongly encourage code deposition in a community repository (e.g. GitHub). See the Nature Portfolio guidelines for submitting code & software for further information.

# Data

Policy information about availability of data

All manuscripts must include a data availability statement. This statement should provide the following information, where applicable:
- Accession codes, unique identifiers, or web links for publicly available datasets
- A description of any restrictions on data availability
- For clinical datasets or third party data, please ensure that the statement adheres to our policy

The datasets generated in this publication have been deposited in NCBI's Gene Expression Omnibus as a SuperSeries accessible through GEO accession number GSE278023, consisting of GSE277875 (Hi-C), GSE277731 (ATAC-seq), GSE308844 (Cut&Run), and GSE309609 (SLAMseq). The following published datasets were used in this study (Supplementary Table 3): GSE132363, GSE178593, and GSE214012. The mass spectrometry proteomics data have been deposited to the ProteomeXchange Consortium via the PRIDE (Perez-Riverol et al. 2022) partner repository with the dataset identifier PXD056346. Source data are provided with this paper. All other data supporting the findings of this study are available from the corresponding author on reasonable request.

GEO accession GSE278023: reviewer token is qtcpqeowhrovfon.
ProteomeXchange #PXD056346: Username: reviewer_pxd056346@ebi.ac.uk Password: aGGcVWQJR8gb

# Research involving human participants, their data, or biological material

Policy information about studies with human participants or human data. See also policy information about sex, gender (identity/presentation), and sexual orientation and race, ethnicity and racism.

| Reporting on sex and gender | *Use the terms sex (biological attribute) and gender (shaped by social and cultural circumstances) carefully in order to avoid confusing both terms. Indicate if findings apply to only one sex or gender; describe whether sex and gender were considered in study design; whether sex and/or gender was determined based on self-reporting or assigned and methods used.* *Provide in the source data disaggregated sex and gender data, where this information has been collected, and if consent has been obtained for sharing of individual-level data; provide overall numbers in this Reporting Summary. Please state if this information has not been collected.* *Report sex- and gender-based analyses where performed, justify reasons for lack of sex- and gender-based analysis.* |
|---|---|
| Reporting on race, ethnicity, or other socially relevant groupings | *Please specify the socially constructed or socially relevant categorization variable(s) used in your manuscript and explain why they were used. Please note that such variables should not be used as proxies for other socially constructed/relevant variables (for example, race or ethnicity should not be used as a proxy for socioeconomic status).* *Provide clear definitions of the relevant terms used, how they were provided (by the participants/respondents, the researchers, or third parties), and the method(s) used to classify people into the different categories (e.g. self-report, census or administrative data, social media data, etc.)* *Please provide details about how you controlled for confounding variables in your analyses.* |
| Population characteristics | *Describe the covariate-relevant population characteristics of the human research participants (e.g. age, genotypic information, past and current diagnosis and treatment categories). If you filled out the behavioural & social sciences study design questions and have nothing to add here, write "See above."* |
| Recruitment | *Describe how participants were recruited. Outline any potential self-selection bias or other biases that may be present and how these are likely to impact results.* |
| Ethics oversight | *Identify the organization(s) that approved the study protocol.* |

Note that full information on the approval of the study protocol must also be provided in the manuscript.

# Field-specific reporting

Please select the one below that is the best fit for your research. If you are not sure, read the appropriate sections before making your selection.

☒ Life sciences ☐ Behavioural & social sciences ☐ Ecological, evolutionary & environmental sciences

For a reference copy of the document with all sections, see nature.com/documents/nr-reporting-summary-flat.pdf

# Life sciences study design

All studies must disclose on these points even when the disclosure is negative.

| | |
|---|---|
| Sample size | No statistical methods were used to predetermine sample size. The use of two replicates is common practice in molecular biology and genomics, given the cost and statistics are needed to be well balanced (PMID:25317452). |
| Data exclusions | No data were excluded from these analyses since all the data generated in this study passed quality control. |
| Replication | All experiments have at least two independent biological replicates. All findings described in the manuscript were confirmed in all individual replicates. |
| Randomization | Randomization of this study was not necessary as we did not allocate datasets into experimental groups. |
| Blinding | All analyses did not require blinding because results were directly linked with the data and this is neither a clinical study with large cohorts nor a genetics study with large numbers of samples. |

# Reporting for specific materials, systems and methods

We require information from authors about some types of materials, experimental systems and methods used in many studies. Here, indicate whether each material, system or method listed is relevant to your study. If you are not sure if a list item applies to your research, read the appropriate section before selecting a response.

Materials & experimental systems

| n/a | Involved in the study |
|---|---|
| ☐ | ☒ Antibodies |
| ☐ | ☒ Eukaryotic cell lines |
| ☒ | ☐ Palaeontology and archaeology |
| ☒ | ☐ Animals and other organisms |
| ☒ | ☐ Clinical data |
| ☒ | ☐ Dual use research of concern |
| ☒ | ☐ Plants |

Methods

| n/a | Involved in the study |
|---|---|
| ☒ | ☐ ChIP-seq |
| ☐ | ☒ Flow cytometry |
| ☒ | ☐ MRI-based neuroimaging |

## Antibodies

| | |
|---|---|
| Antibodies used | commercial antibodies used in this study are listed in the relevant supplementary methods.<br><br>Western blotting:<br>Primary antibodies: 1:500 mouse anti-RanGAP1 (OTI1B4, Novus Biologicals, NBP2-02623), 1:500 mouse anti-Nup93 (F-2, Santa Cruz Biotechnology, sc-374400), 1:1000 rabbit anti-vinculin (EP18185, Abcam, ab129002), 1:1000 rabbit anti-BRD4 (E2A7X, Cell Signaling 13440), 1:1000 rabbit anti-BRD2 (EPR7642, ab139690). Secondary antibodies: 1:1000 goat anti-mouse IgG-HRP (Cell Signaling 7076), 1:1000 goat anti-rabbit IgG-HRP (Cell Signaling 7074).<br><br>Immunofluorescence:<br>Primary antibodies: 1:10000 mouse anti-alpha-tubulin (Sigma T6199), 1:1000 rabbit anti-histone H3pS28 (Abcam, ab5169), 1:1000 mouse anti-emerin (Abcam ab204987), 1:1000 rabbit anti-Lamin B-receptor (Abcam, ab32535), 1:1000 rabbit anti-Lamin A (Abcam, ab26300), 1:1000 mouse anti-Lamin A/C (Santa Cruz sc-7292 (636)), 1:500 mouse anti-Elys (BioMatrix research, BMR00513), 1:1000 rabbit anti-Nup160 (Abcam, ab73293), 1:1000 mouse anti-Mab414 (Abcam, ab24609), 1:2000 rabbit anti-SON (Thermofisher, PA5-65107), 1:2000 mouse anti-NPM1 (Thermofisher, 60096-1), 1:1000 rabbit anti-Rad21 (Abcam, ab154769), 1:1000 rabbit anti-CTCF (Cell signaling, 2899), 1:1000 rabbit anti-RNAPolII pS2 (Abcam, ab5095). Secondary antibodies: 1:1000 goat anti-mouse IgG H+L Alexa Fluor 488 (Abcam, ab150113), 1:1000 goat anti-mouse IgG H+L Alexa Fluor 568 (Abcam ab175473), 1:1000 goat anti-rabbit IgG H&L Alexa Fluor 488 (Abcam ab15007), 1:1000 goat anti-rabbit IgG H&L Alexa Fluor 568 (Abcam, ab175471). |
| Validation | all previously published commercial antibodies |

# Eukaryotic cell lines

Policy information about cell lines and Sex and Gender in Research

| | |
|---|---|
| Cell line source(s) | Engineered DLD-1 cell lines<br>Previously published and described in Supplementary methods.<br><br>The CRISPR/Cas9 system was used to endogenously target the RanGAP1, RCC1 (Aksenova et al. 2022, 2020), NUP93 (Regmi et al. 2020) and AAVS1 (Chu et al. 2015) genes. With the exception of nuclear import assays, all experiments described here employed human colorectal adenocarcinoma DLD-1 cells (ATCC CCL-221) expressing either RanGAP1 or Nup93 homozygously tagged with NeonGreen and an Auxin-Inducible Degron, Infra-Red protein (IFP)-tagged RCC1 and Tir1, as described previously. We refer to these cell lines as RanGAP1-AID and Nup93-AID.<br><br>For cell lines used in nuclear import assays, the sequences of MBP and mScarlet, were amplified by PCR from pMAL (NEB) and pmScarlet_alphaTubulin_C1 (Addgene, #85045), respectively. The NLS sequence was synthesized, and all fragments were inserted by Gibson reaction (E2611S, NEB) into the MCS of AAVS1_Puro_PGK1 vector (Addgene, #68375) through replacement of 3xFlagTwinStep-Tag. MBP-mScarlet-NLS was inserted into the AAVS1 locus in RanGAP1-AID and NUP93-AID DLD-1 cell lines. |
| Authentication | DNA from DLD-1 and CRISPR/Cas9-targeted cells was extracted with the Wizard® Genomic DNA Purification Kit (Promega). Clones were genotyped by PCR for homozygous insertion of tags with two sets of primers as described in Aksenova et al 2020 (RanGAP1) and Regmi et al 2020 (Nup93). |
| Mycoplasma contamination | Cell lines were routinely tested for mycoplasma infection and tested negative (MycoAlertTM Mycoplasma Detection Kit, Lonza). |
| Commonly misidentified lines<br>(See ICLAC register) | No commonly misidentified cell lines were used in this study. |

# Plants

| | |
|---|---|
| Seed stocks | *Report on the source of all seed stocks or other plant material used. If applicable, state the seed stock centre and catalogue number. If plant specimens were collected from the field, describe the collection location, date and sampling procedures.* |
| Novel plant genotypes | *Describe the methods by which all novel plant genotypes were produced. This includes those generated by transgenic approaches, gene editing, chemical/radiation-based mutagenesis and hybridization. For transgenic lines, describe the transformation method, the number of independent lines analyzed and the generation upon which experiments were performed. For gene-edited lines, describe the editor used, the endogenous sequence targeted for editing, the targeting guide RNA sequence (if applicable) and how the editor was applied.* |
| Authentication | *Describe any authentication procedures for each seed stock used or novel genotype generated. Describe any experiments used to assess the effect of a mutation and, where applicable, how potential secondary effects (e.g. second site T-DNA insertions, mosiacism, off-target gene editing) were examined.* |

# Flow Cytometry

## Plots

Confirm that:

☒ The axis labels state the marker and fluorochrome used (e.g. CD4-FITC).

☒ The axis scales are clearly visible. Include numbers along axes only for bottom left plot of group (a 'group' is an analysis of identical markers).

☒ All plots are contour plots with outliers or pseudocolor plots.

☒ A numerical value for number of cells or percentage (with statistics) is provided.

## Methodology

| | |
|---|---|
| Sample preparation | See supplementary methods. EtOH or FA/DSG fixation used for downstream application:<br><br>For cell cycle dynamics determinations:<br>Cells were collected at various points of mitotic exit. Adherent cells were dissociated with accutase (ThermoFisher Scientific, A11105-01) and pooled with non-adherent collected cells in order to assess the entire population. To assess the cell-cycle profile (DNA content), cell pellets were resuspended in 200ul PBS and fixed with 800μl of cold 100% ethanol. Cells were stored at −20°C for at least 24h. Approximately 1 million fixed cells were stained with 50 ug/ml propidium iodide (PI) (Thermo, P1304MP), diluted in 1 ml PBS containing 50 ug/ml RNaseA (Roche, 10109169001) and 0.1% Saponin, for 1 hour at room temperature. After staining, cells were spun, and pellets were resuspended in 1ml of PBS and passed through a 35 um filter (Falcon 352235).<br><br>For Hi-C sorting:<br>Cells were collected at various points of mitotic exit and fixed for Hi-C 3.0 (Lafontaine et al. 2021) with a few modifications to |

facilitate cell sorting.  Adherent cells collected 5 and 10 hours after prometaphase release, were dissociated with accutase. Prometaphase (t=0) and early (t=1.25-1.5h) released cells were directly collected by shake-off. Cell suspensions were pelleted and treated with accutase for an additional 5 minutes at room temperature to prevent aggregation and washed with HBSS (Thermofisher, 14025134). Fixation proceeded first with 1% formaldehyde (Fisher, BP531-25) in HBSS for 10 minutes, which was quenched with 0.125M Glycine for 5 minutes at room temperature and 15 minutes on ice.  Next, cells were fixed with 3mM disuccinimidyl glutarate (DSG) in PBS for 40 minutes rotating at room temperature, followed by a second quenching with 0.125M Glycine. Fixed cells were washed twice with PBS + 0.1% BSA and snap-frozen in liquid nitrogen prior to staining for Fluorescence-activated cell sorting (FACS). In order to sort cells by DNA content, approximately 10 million fixed cells were stained with 50 ug/ml PI, diluted in 5 ml PBS containing 50 ug/ml RNaseA and 0.1% Saponin, for 1 hour at room temperature. Cells were then spun and washed with PBS prior to resuspension in 2 ml PBS + 0.1% BSA and passage through a 35 um filter.

| Instrument | See supplementary methods. EtOH or FA/DSG fixation used for downstream application: <br><br> For cell cycle dynamics determinations: <br> Cells were collected at various points of mitotic exit.  Adherent cells were dissociated with accutase (ThermoFisher Scientific, A11105-01) and pooled with non-adherent collected cells in order to assess the entire population. To assess the cell-cycle profile (DNA content), cell pellets were resuspended in 200ul PBS and fixed with 800µl of cold 100% ethanol. Cells were stored at −20°C for at least 24h. Approximately 1 million fixed cells were stained with 50 ug/ml propidium iodide (PI) (Thermo, P1304MP), diluted in 1 ml PBS containing 50 ug/ml RNaseA (Roche, 10109169001) and 0.1% Saponin, for 1 hour at room temperature.  After staining, cells were spun, and pellets were resuspended in 1ml of PBS and passed through a 35 um filter (Falcon 352235). <br><br> For Hi-C sorting: <br> Cells were collected at various points of mitotic exit and fixed for Hi-C 3.0 (Lafontaine et al. 2021) with a few modifications to facilitate cell sorting.  Adherent cells collected 5 and 10 hours after prometaphase release, were dissociated with accutase. Prometaphase (t=0) and early (t=1.25-1.5h) released cells were directly collected by shake-off. Cell suspensions were pelleted and treated with accutase for an additional 5 minutes at room temperature to prevent aggregation and washed with HBSS (Thermofisher, 14025134). Fixation proceeded first with 1% formaldehyde (Fisher, BP531-25) in HBSS for 10 minutes, which was quenched with 0.125M Glycine for 5 minutes at room temperature and 15 minutes on ice.  Next, cells were fixed with 3mM disuccinimidyl glutarate (DSG) in PBS for 40 minutes rotating at room temperature, followed by a second quenching with 0.125M Glycine. Fixed cells were washed twice with PBS + 0.1% BSA and snap-frozen in liquid nitrogen prior to staining for Fluorescence-activated cell sorting (FACS). In order to sort cells by DNA content, approximately 10 million fixed cells were stained with 50 ug/ml PI, diluted in 5 ml PBS containing 50 ug/ml RNaseA and 0.1% Saponin, for 1 hour at room temperature. Cells were then spun and washed with PBS prior to resuspension in 2 ml PBS + 0.1% BSA and passage through a 35 um filter. |
|---|---|
| Software | FlowJo v10 |
| Cell population abundance | 10,000 total events typically recorded. Downsampling for plotting live events as indicated |
| Gating strategy | Live cell gating based on FSC-A/SSC-A distribution (see Supplementary Fig. 1). <br><br> For cell cycle determination: <br> Flow cytometry for the mitotic release timecourse was performed on a MACSQUANT set-up for at least three biological replicates. Analysis was performed using FlowJo software (v10) and plots reflect populations from a representative experiment gated for debris but not doublets and sampled to equal live cell event numbers. <br><br> For Hi-C sorting: <br> Propidium iodide-stained cell suspensions were sorted on a BD FACS Melody using the 561 nm laser for FSC, SSC, and PI. All populations were gated based on FSC/SSC to eliminate cell debris and cells sorted for either prometaphase (4n) or G1 (2n) DNA content were also subject to doublet discrimination. To enrich for telophase or cytokinesis, cells fixed 1.25 and 1.5 hours after mitotic release, respectively, were sorted based on doubled PI signal (DNA content) area and width.  All sorted cells were collected in PBS containing 1% BSA and washed twice in PBS prior to snap-freezing. |

☒ Tick this box to confirm that a figure exemplifying the gating strategy is provided in the Supplementary Information.

