## [Peer Review File · Nature Cell Biology]

Interphase chromosome conformation is specified by distinct folding programs inherited through mitotic chromosomes or the cytoplasm

Corresponding Author: Dr Job Dekker

Version 0:

Decision Letter:

*Please delete the link to your author homepage if you wish to forward this email to co-authors.

Dear Dr Dekker,

Your manuscript, "Interphase chromosome conformation is specified by distinct folding programs inherited via mitotic chromosomes or through the cytoplasm", has now been seen by 3 referees, who are experts in 3D genome organisation (referee 1); chromatin dynamics and gene expression regulation (referee 2); and chromosome folding (referee 3). As you will see from their comments (attached below) they find this work of potential interest, but have raised some concerns, which in our view would need to be addressed with considerable revisions before we can consider publication in Nature Cell Biology.

Nature Cell Biology editors discuss the referee reports in detail within the editorial team, including the chief editor, to identify key referee points that should be addressed with priority, and requests that are overruled as being beyond the scope of the current study. To guide the scope of the revisions, I have listed these points below. We are committed to providing a fair and constructive peer-review process, so please feel free to contact me if you would like to discuss any of the referee comments further.

In particular, it would be essential to:

A- Further experimentally test the role of microcompartment domains in mitotic bookmarking (Reviewer#1 pt 1, Reviewer#2 paragraph "The MCDs coincide with the active CREs of a given cell type and...")

B- Experimentally test the role of histone modifications and the role of BRD4 in MCD formation (Reviewer#2 paragraph "The MCDs coincide with the active CREs of.." and Reviewer#3 paragraph "The authors discuss the possibility that histone modifications such as..")

C- Test whether RanGAP1 and Nup43 are involved in MCD interactions (reviewer#2 paragraph "the authors demonstrate an overlap...")

D- Further clarify the concept of resorption (Reviewer#1 pt, Reviewer#3 paragraph) and whether resorption is distinct from pruning.

- All other referee concerns pertaining to strengthening existing data, providing controls, methodological details, clarifications and textual changes, should also be addressed.

- Finally please pay close attention to our guidelines on statistical and methodological reporting (listed below) as failure to do so may delay the reconsideration of the revised manuscript. In particular please provide:

We would be happy to consider a revised manuscript that would satisfactorily address these points, unless a similar paper is

published elsewhere, or is accepted for publication in Nature Cell Biology in the meantime.

- ensure that it conforms to our format instructions and publication policies (see below and <https://www.nature.com/nature/for-authors>).

- provide a point-by-point rebuttal to the full referee reports verbatim, as provided at the end of this letter.

- provide the completed Reporting Summary (found here <https://www.nature.com/documents/nr-reporting-summary.pdf>). This is essential for reconsideration of the manuscript will be available to editors and referees in the event of peer review. For more information see <http://www.nature.com/authors/policies/availability.html> or contact me.

Nature Cell Biology is committed to improving transparency in authorship. As part of our efforts in this direction, we are now requesting that all authors identified as 'corresponding author' on published papers create and link their Open Researcher and Contributor Identifier (ORCID) with their account on the Manuscript Tracking System (MTS), prior to acceptance. ORCID helps the scientific community achieve unambiguous attribution of all scholarly contributions. You can create and link your ORCID from the home page of the MTS by clicking on 'Modify my Springer Nature account'. For more information please visit www.springernature.com/orcid.

This journal strongly supports public availability of data. Please place the data used in your paper into a public data repository, or alternatively, present the data as Supplementary Information. If data can only be shared on request, please explain why in your Data Availability Statement, and also in the correspondence with your editor. Please note that for some data types, deposition in a public repository is mandatory - more information on our data deposition policies and available repositories appears below.

Link Redacted

We would like to receive a revised submission within six months.

We hope that you will find our referees' comments, and editorial guidance helpful. Please do not hesitate to contact me if there is anything you would like to discuss.

Best wishes,

Sabrya Carim

Sabrya Carim, PhD
(she/her/hers)
Associate Editor, Nature Cell Biology
Nature Portfolio

Springer Nature
The Campus, 4 Crinan Street, London N1 9XW, UK
sabrya.carim@springernature.com
<https://orcid.org/0000-0001-9485-1938>

Reviewers' Comments:

Reviewer #1 (Remarks to the Author):

This study by Schooley et al. describes an ingenious system to characterize how cytoplasmic proteins fold the genome following mitosis. By blocking nuclear import in synchronized cells as they enter G1, the authors show that chromosomes form microcompartments—small domains of interaction between regions with enhancer and promoter histone marks. This process is independent of cytoplasmic factors and is short lived. Early in G1, microcompartments are quickly dissolved and absorbed by A/B compartments. As this doesn't happen in cells where nuclear import is blocked this shows that loop extrusion and likely the beginning of transcription are responsible for the dissolving microcompartments. I find it also important that this manuscript validates the existence of microcompartments that had only previously been seen with extremely deep-coverage Hi-C or with Region Capture Micro-C. This is a very thorough study where the authors performed a careful analysis of the data to provide strong support for their claims. I think the following would make the study much stronger.

Point 1 Considering that the main finding of the paper is that microcompartments arise through exclusively nuclear factors inherited through mitosis, the bookmarking analysis in Figure 4 could be more complete. For example, generating H3K4me3, H3K27ac, and CTCF from cells in mitosis and G1 cells would provide a much better view of the bookmarking happening in these cells. It would also be interesting to look at the binding of TFs like AP2 but that might be much more challenging. Just like the authors did throughout the paper, this section would also benefit from specific example loci showing ATAC signal at bookmarked and non-bookmarked sites, and footprint score, along with EV, and histone marks. That would really help the reader understand better the global analysis that is shown.

In addition, I thought that some plots in figure 4 were particularly hard to follow. The label 5H MCD used in figure 4 is not very intuitive. It is also not written in the figure legend or the text. That means extra effort is required each time to understand that 5H MCD represents the ATAC peaks at MCDs in 4A and the actual microcompartments in 4C. Having the consistent nomenclature in text, legend and panel labels will be helpful.

Panel 4B- I am not sure using the union of atac peaks is very informative here. Did the authors try to plot these same marks in the specific peaksets from figure 4A? Does that provide a clearer characterization of retained mitotic peaks? Furthermore, although the authors decided to highlight the enrichment enhancer and promoter marks, it is also clear that they contain lots of CTCF binding. So just looking at this plot it could be argued that CTCF is playing a role here, which I don't believe is true. Panel 4C – I found these plots very non intuitive and their description insufficient. What does color mean? Breaking it into 4 categories when in the end the analysis is described mostly as 0 or non zero enhancer/promoter also unnecessarily complicates the plots. I would also try to guide the reader through the plots in the text much better. It took me more than 30 minutes to understand how these plots show what the authors highlight in the text. Perhaps trying to refer what specific plot in this panel shows would help.

Point 2 – I think it would be interesting to expand the characterization of what sets microcompartments going by quantifying expression in control cells initiating G1 (either TTseq or PoIII chip). Are the first loci to be transcribed the most microcompartmentalized? The data in the paper suggests that transcription is not required for microcompartmentalization, but maybe microcompartmentalization contributes to initiate transcription post-mitosis? This should be a quick straightforward experiment and the result interesting either way.

Point 3 The concept described in figure 6 is also very interesting. As in figure 4, if the readers could visualize a specific locus as an example of a microcompartment being resorbed into a larger compartment would be very helpful.

Point 4 I thought figure 7 should have come straight after figure 1. In fact in some parts of the paper the authors refer to “as we show below”. Putting it up in the beginning wouldn't affect anything and would actually inform the reader about how to best interpret the experiments and which factors are excluded from the nuclei.

Smaller points

-I really appreciate how careful the authors were in being consistent with nomenclature. And figure 5i is really helpful. However, the accompanying paper uses the word microcompartments instead of microcompartment domains, which was also previously used (Goel et al 2023). Since the good old A/B compartments also don't have the word domain, shouldn't the D be dropped in MCD?

-2b please add genomic coordinates

-2c Do all K27ac and K4me3 rich regions show evidence of compartmentalization? Could you plot EV1 at all K27ac/K3me3 regions?

-what is the last column in 3e?

-5e does not have figure legend

6a-what are the two Ev1 lines below hi-c maps?

Reviewer #2 (Remarks to the Author):

Review Dekker

In this paper, Schooley et al., developed degron systems for two proteins essential for nuclear transport, RanGAP1 and Nup93. These proteins were degraded pro-metaphase cells to prevent an influx of cytoplasmic proteins during mitotic exit. With this system they uncovered a transient chromatin folding intermediate program that gives rise to microcompartmentalization of mitotically bookmarked CREs, which form during telophase. Microcompartmentalization is modified by the entry of cohesin and the basal transcription protein as well as other factors modulated by chromatin remodellers. By depleting RanGAP1 and Nup93 after prometaphase release, they were able to recover MCD interactions, demonstrating that the distinct G1 folding program and continuous active nuclear transport are important for maintenance and pruning of non-specific interactions between MCDs. The study thus provides important new insight into chromatin folding in mitotic cells and highlights a requirement for a continuous influx of cytoplasmic factors for establishing and maintaining interphase chromatin folding.

The MCDs coincide with the active CREs of a given cell type and appear to be associated with bookmarking, but this has not been established. Since the BET family, particularly BRD4, have been found to associate with mitotic chromosomes and could potentially be involved in MCD formation, the authors should perform a BRD4 ChIP-seq to see if it overlaps with MCDs and if it does they can use a BRD4 inhibitor to determine whether BRD4 inhibition impairs bookmarking upon mitotic exit and alters CTCF-mediated looping. Demonstrating a more causal role for MCDs in bookmarking would add value to the paper.

The authors demonstrate an overlap between active histone marks and MCDs in control cells, but in the absence of similar analyses in the RanGAP1 and Nup93-depleted cells, it is not clear whether these modifications change in the absence of an influx of cytoplasmic proteins. This would be important to know if they are somehow involved in the MCD interactions. These experiments should be included.

Figure 2b and 2c: Can the authors show the ATACseq profiles and the ChIP-seq for H3K27ac and H3K4me3 also for the RanGAP1-depleted cells (at the moment only the control cells are shown).

For the 4 conditions shown in Fig 3c it would be nice to have RNA Pol II ChIP-seq for both control and RanGAP1-depleted cells, to see how the profile changes over time relative to the microcompartments.

Minor points:

- Figures and figure legends don't have sufficient information making them hard to navigate
- No call out for Figure S1
- Why are controls for the samples in Fig 1g so different?

Reviewer #3 (Remarks to the Author):

Summary:

Current data indicate that chromosomes refold every cell cycle, leading Schooley et al. to investigate here determinants of interphase chromosome conformation after cell division. They establish an experimental system to physically segregate the G1 cytoplasm from the nucleus which enables the authors to study intrinsic versus extrinsic factors that affect chromosome conformation. To accomplish this, they induce degradation of RanGAP1 and Nup93, which are essential for nuclear-cytoplasmic transport, in pro-metaphase, follow the cells into G1, which appears to work as expected based upon cell biological assays, and then primarily perform Hi-C analysis to investigate chromosome conformation, ATAC-seq to study cis regulatory elements, and label-based mass spectrometry proteomics to identify the proteins localized to G1 nuclei independent of cytoplasmic to nuclear transport. This approach is very creative and original, setting this study apart from others in the field. Because of the authors' thoughtfulness they identify an interphase chromosome conformation determined solely by chromatin state that reflects compartmentalization of cis-regulatory elements that were bookmarked during the prior mitosis. A particular strength of the study is that induced degradation of RanGAP1 and Nup93 both show the formation of this chromosome conformation. During normal cell physiology, factors that drive biochemical processes including loop extrusion, transcription, RNA processing, etc. are inherited from the cytoplasm and impact chromosome conformation but this is distinct from chromosome intrinsic factors. This work is important for the 3D genome structure field as it provides compelling evidence that cell-type-specific compartmentalization is inherited from chromatin through the formation of an intermediate composed of self-associating chromatin features, which are subsequently modified by cytoplasmic factors. Therefore, the experiments and analyses performed are rigorous and informative on a significant question in chromosome

biology that is relevant to readers of *Nature Cell Biology* and beyond. This reviewer would like to emphasize their enthusiasm for the work as the comments below are to improve an already impressive manuscript and hopefully will not substantially delay publication.

Major comments:

The authors claim that microcompartment formation is not limited to telophase and cytokinesis and can still occur within G1 cells upon loss of nuclear import. To test this, they perform an interesting experiment by treating cells with auxin 3.5 hours after prometaphase block release and performing Hi-C during late G1. An important control to strengthen the claim that the microcompartment is re-forming would be to show that the long-range MCD interactions have fully disappeared in control cells by 3.5 hours after prometaphase, the timepoint of auxin addition, before reforming at late G1. If the authors can complete this, they may want to consider including Extended Fig 6 as a main figure. If not, they should consider the alternative interpretation that MCD-MCD interactions have not been fully pruned/resorbed by 3.5 hours after prometaphase block release.

The approach to disrupt nuclear import is rigorous and robust as two distinct proteins, RanGAP1 and Nup93, that control compartmentalization of nuclei from cytosol by different mechanisms are separately induced for degradation. The main results and interpretations outlined in the manuscript seem to generally agree upon degradation of both factors. However, quantitatively, the authors state that lower enrichment of MCD contact frequency in Nup93 depleted cells compared to RanGAP1 depleted cells may reflect clonal variation between the two cell lines as MCDs were assigned in the latter context. The authors do have Hi-C data from Nup93 depleted cells (Extended Data Fig. 2). Can the authors please identify MCDs from the Nup93 depleted Hi-C data and determine if the quantitative differences in MCD contact frequency are due to clonal differences? An alternative explanation for quantitative differences in MCD contact frequency could be related to incomplete MBP-mScarlet-NLS nuclear localization in Nup93 control cells (Figure 1g and Extended Figure 1f).

To make the manuscript as clear and impactful as possible, the authors may want to consider terminology. The chromosome conformation capture field now has three terms related to compartmentalization: compartments, subcompartments, and microcompartments. It is currently unclear the significant difference between these terms; a possibility is new terms are being added as technology evolves without an underlying meaningful difference between the terms. Furthermore, investigators outside the chromosome conformation capture field, who likely will be very interested in this exciting manuscript, are unlikely to understand the differences between compartments, subcompartments, and microcompartments. Fig 6 might be confusing for these readers where some subcompartments (the B subcompartments) are pooled, whereas others are not. And, based on terms alone, questions arise such as what is the difference between a subcompartment and microcompartment (both imply small)? Or, are they all just compartments? A suggestion from this reviewer would be for the authors to clearly and explicitly define each of the three terms (compartments, subcompartments, and microcompartments) or to pick one term while acknowledging that there are other terms and ambiguity, at present, in the field.

When thinking about physiological normal G1 cells, the present interpretation is that MCD-MCD interactions are pruned in G1. I would appreciate the authors' thoughts on the following questions: Is cohesin-dependent loop extrusion actively pruning the interactions, or, in a population of cells (Hi-C experiments presented here are bulk experiments) are more of the Hi-C interactions in G1 representative of loop extrusion interactions compared to MCD-MCD interactions? Put another way, can the authors distinguish pruning, an active process, from a superimposition of folding states, a technical consideration when interpreting Hi-C data?

Fig. 6D shows that MCD-MCD interactions change from cytokinesis to G1 in control cells, but whether this change is resorption is unclear. This reviewer would expect the 2x2 boxes in cis for the A1, A2, and VVI IPGs (regardless of MCD or not) in control G1 cells to be uniform in contact frequency if resorption were occurring. If the cCRE microcompartment is being resorbed by active subcompartments, then why is the VVI-MCD IPG depleted for interactions with the A1 IPG in control G1 cells in cis? For interactions in trans, in control G1 cells the contact frequency for IPGs is more what one might expect if resorption were occurring, but the contact frequencies at cytokinesis don't show as clear distinction between MCDs and subcompartments as the cis contact frequencies. For example, the A1-MCD IPG interacts more strongly with the A1 IPG than with the A2-MCD IPG indicating that the A1-MCD IPG is more like the A1 subcompartment than the cCRE microcompartment in trans in control cells in cytokinesis. Either the concept of resorption needs to be clarified, or other interpretations considered, such as that the data presented in Fig 6D also reflect the cytoplasmic folding program thereby impacting interpretation of results. This leads to the following questions: Is resorption distinct from pruning? If so, how can these be distinguished?

The authors discuss the possibility that histone modifications such as H3K27ac may mediate affinity-driven compartmentalization through bridging factors such as Brd4. An omitted and relevant study to this discussion is the work of Rosencrance et al., *Mol Cell*, 2020, which revealed that hyperacetylated chromatin domains formed by the fusion oncoprotein BRD4-NUT make chromatin interactions comparable to those observed here. Combined with the observation by immunofluorescence microscopy that BRD4-NUT localizes to mitotic chromosomes (Fig. 1A from Alekseyenko et al., *Genes & Development*, 2015) these studies are consistent with and support the work presented here on a connection between mitotic bookmarking and affinity-driven compartmentalization.

Minor comments:

Fig 1B: Can the authors please perform a Western blot from parental DLD-1 cells of untagged RanGAP1 and Nup93? Does the AID tag affect stability of either protein?

The resolution at which the detection of MCDs was performed should be stated in the main text.

Fig 2D and F, Extended Data Fig 3: Please define dELS, pELS, and PLS in the figure legends.

Fig 4C: Legend: circles overlap legend text. Please add a legend for the color scale.

The figure legends for Extended Data Fig 4D and E do not match the figure.

Datapoint labels in Supplementary Fig 2C and D are not clear. How do the labels in panels C and D correspond to panel A?

Supplementary Fig 3 is very helpful and clear, but the legend starts with c rather than a.

Page 6: "Furthermore, nucleoli and nuclear speckles, two import dependent organelles with important roles in nuclear organization (Prasanth et al. 2003; Spector and Lamond 2011; Leung et al. 2004; Görlich and Kutay 1999), are not found in RanGAP1-AID or Nup93-AID-depleted nuclei and apparent interchromatin granules with strong speckle protein (SON) localization are found exclusively in the cytoplasm (Extended Data Fig. 1f)." This should refer to Extended Data Fig 1E.

REFERENCES – are limited to a total of 70 for Articles, Resources, Technical Reports; and 40 for Letters. This includes references in the main text and Methods combined. References must be numbered sequentially as they appear in the main text, tables and figure legends and Methods and must follow the precise style of Nature Cell Biology references. References only cited in the Methods should be numbered consecutively following the last reference cited in the main text. References

only associated with Supplementary Information (e.g. in supplementary legends) do not count toward the total reference limit and do not need to be cited in numerical continuity with references in the main text. Only published papers can be cited, and each publication cited should be included in the numbered reference list, which should include the manuscript titles. Footnotes are not permitted.

Methods should be written concisely, but should contain all elements necessary to allow interpretation and replication of the results. As a guideline, Methods sections typically do not exceed 3,000 words. The Methods should be divided into subsections listing reagents and techniques. When citing previous methods, accurate references should be provided and any alterations should be noted. Information must be provided about: antibody dilutions, company names, catalogue numbers and clone numbers for monoclonal antibodies; sequences of RNAi and cDNA probes/primers or company names and catalogue numbers if reagents are commercial; cell line names, sources and information on cell line identity and authentication. Animal studies and experiments involving human subjects must be reported in detail, identifying the committees approving the protocols. For studies involving human subjects/samples, a statement must be included confirming that informed consent was obtained. Statistical analyses and information on the reproducibility of experimental results should be provided in a section titled "Statistics and Reproducibility".

All Nature Cell Biology manuscripts submitted on or after March 21 2016 must include a Data availability statement as a separate section after Methods but before references, under the heading "Data Availability". For Springer Nature policies on data availability see <http://www.nature.com/authors/policies/availability.html>; for more information on this particular policy see <http://www.nature.com/authors/policies/data/data-availability-statements-data-citations.pdf>. The Data availability statement should include:

- Accession codes for primary datasets (generated during the study under consideration and designated as "primary accessions") and secondary datasets (published datasets reanalysed during the study under consideration, designated as "referenced accessions"). For primary accessions data should be made public to coincide with publication of the manuscript. A list of data types for which submission to community-endorsed public repositories is mandated (including sequence, structure, microarray, deep sequencing data) can be found here <http://www.nature.com/authors/policies/availability.html#data>.
- Unique identifiers (accession codes, DOIs or other unique persistent identifier) and hyperlinks for datasets deposited in an approved repository, but for which data deposition is not mandated (see here for details <http://www.nature.com/sdata/data-policies/repositories>).
- At a minimum, please include a statement confirming that all relevant data are available from the authors, and/or are included with the manuscript (e.g. as source data or supplementary information), listing which data are included (e.g. by figure panels and data types) and mentioning any restrictions on availability.
- If a dataset has a Digital Object Identifier (DOI) as its unique identifier, we strongly encourage including this in the Reference list and citing the dataset in the Methods.

We recommend that you upload the step-by-step protocols used in this manuscript to protocols.io. More details can be found at <https://www.protocols.io/help/publish-articles>.

All imaging data should be accompanied by scale bars, which should be defined in the legend.

Cropped images of gels/blots are acceptable, but need to be accompanied by size markers, and to retain visible background signal within the linear range (i.e. should not be saturated). The boundaries of panels with low background have to be demarked with black lines. Splicing of panels should only be considered if unavoidable, and must be clearly marked on the figure, and noted in the legend with a statement on whether the samples were obtained and processed simultaneously. Quantitative comparisons between samples on different gels/blots are discouraged; if this is unavoidable, it should only be performed for samples derived from the same experiment with gels/blots were processed in parallel, which needs to be stated in the legend.

Figures should be provided at approximately the size that they are to be printed at (single column is 86 mm, double column is 170 mm) and should not exceed an A4 page (8.5 x 11"). Reduction to the scale that will be used on the page is not necessary, but multi-panel figures should be sized so that the whole figure can be reduced by the same amount at the smallest size at which essential details in each panel are visible. In the interest of our colour-blind readers we ask that you avoid using red and green for contrast in figures. Replacing red with magenta and green with turquoise are two possible

colour-safe alternatives. Lines with widths of less than 1 point should be avoided. Sans serif typefaces, such as Helvetica (preferred) or Arial should be used. All text that forms part of a figure should be rewritable and removable.

The total number of Supplementary Figures (not including the "unprocessed scans" Supplementary Figure) should not exceed the number of main display items (figures and/or tables (see our Guide to Authors and March 2012 editorial <http://www.nature.com/ncb/authors/submit/index.html#suppinfo>; <http://www.nature.com/ncb/journal/v14/n3/index.html#ed>). No restrictions apply to Supplementary Tables or Videos, but we advise authors to be selective in including supplemental data.

GUIDELINES FOR EXPERIMENTAL AND STATISTICAL REPORTING

REPORTING REQUIREMENTS – We are trying to improve the quality of methods and statistics reporting in our papers. To that end, we are now asking authors to complete a reporting summary that collects information on experimental design and reagents. The Reporting Summary can be found here <https://www.nature.com/documents/nr-reporting-summary.pdf> If you would like to reference the guidance text as you complete the template, please access these flattened versions at <http://www.nature.com/authors/policies/availability.html>.

Version 1:

Decision Letter:

Our ref: NCB-A55514A

2nd October 2025

Dear Dr. Dekker,

Thank you for submitting your revised manuscript "Interphase chromosome conformation is specified by distinct folding programs inherited via mitotic chromosomes or through the cytoplasm" (NCB-A55514A). It has now been seen by the original referees and their comments are below. The reviewers find that the paper has improved in revision, and therefore we'll be happy in principle to publish it in Nature Cell Biology, pending minor revisions to satisfy the referees' final requests and to comply with our editorial and formatting guidelines.

Please note that our articles must have 5 to 8 main figures and they can have up to 10 Extended Data figures. If there are more figures than that, they become supplementary figures. Supplementary materials are less accessed than our main and ED figures so we try to limit the use of supplementary figures as much as we can. Please ensure that all figures fit into a single page (not multiple pages) and adhere to a maximum page size of roughly 180mm wide x 200mm high and use a font size of no smaller than 6pt throughout the figures, to ensure legibility of the figures once resized for publication. Also, please use the full page space to fill the figure and remove the figure labels from the Figures. Several figures need to be rearranged as they are in landscape format and this will not work with our layout. We would ask that you rearrange the panels of the figures which are in landscape format to adhere to a maximum page size of roughly 180mm wide x 200mm high. Please ensure that the figure labels are not too close to the page margins.

We are now performing detailed checks on your paper and will send you a checklist detailing our editorial and formatting requirements in a week. ****Please do not upload the final materials and make any revisions until you receive this additional information from us.****

Thank you again for your interest in Nature Cell Biology Please do not hesitate to contact me if you have any questions.

Sincerely,

Sabrya

Sabrya Carim, PhD
(she/her/hers)
Senior Editor, Nature Cell Biology
Nature Portfolio

Springer Nature
The Campus, 4 Crinan Street, London N1 9XW, UK
sabrya.carim@springernature.com
<https://orcid.org/0000-0001-9485-1938>

Reviewer #1 (Remarks to the Author):

The authors made vast and impressive additions to the manuscript. I congratulate them on this study and recommend publication.

Reviewer #2 (Remarks to the Author):

The authors have performed all the additional experiments requested and as a result the manuscript is much improved and I would recommend publication.

Reviewer #3 (Remarks to the Author):

The authors have done an excellent job thoroughly responding to all my comments. This reviewer remains very enthusiastic about this creative and original study. With all the new data and analyses this manuscript is above the bar for Nature Cell Biology. Congratulations on a fantastic paper!

Version 2:

Decision Letter:

Dear Dr Dekker,

I am writing on behalf of my colleague Dr Sabrya Carim, who is out of the office.

I am pleased to inform you that your manuscript, "Interphase chromosome conformation is specified by distinct folding programs inherited through mitotic chromosomes or the cytoplasm", has now been accepted for publication in Nature Cell Biology.

Please note that *Nature Cell Biology* is a Transformative Journal (TJ). Authors may publish their research with us through the traditional subscription access route or make their paper immediately open access through payment of an article-processing charge (APC). Authors will not be required to make a final decision about access to their article until it has been accepted. [Find out more about Transformative Journals](https://www.springernature.com/gp/open-research/transformative-journals)

Authors may need to take specific actions to achieve compliance with funder and institutional open access mandates. If your research is supported by a funder that requires immediate open access (e.g. according to [Plan S principles](https://www.springernature.com/gp/open-science/plan-s-compliance) or the [NIH public access policy](https://www.springernature.com/gp/open-science/us-federal-agency-compliance)) then you should select the gold OA route, and we will direct you to the compliant route where possible. Because authors warrant under our subscription licensing terms that they haven't committed to licensing any version of their article under a licence inconsistent with the terms of our agreement – including the applicable embargo period – publication under the subscription model isn't suitable for authors whose funders require no embargo.

If you have not already done so, we strongly recommend that you upload the step-by-step protocols used in this manuscript to protocols.io (<https://protocols.io>), an open online resource that allows researchers to share their detailed experimental know-how. All uploaded protocols are made freely available and are assigned DOIs for ease of citation. Protocols and Nature Portfolio journal papers in which they are used can be linked to one another, and this link is clearly and prominently visible in the online versions of both. Authors who performed the specific experiments can act as primary authors for the Protocol as they will be best placed to share the methodology details, but the Corresponding Author of the present research paper should be included as one of the authors. By uploading your Protocols onto protocols.io, you are enabling researchers to more readily reproduce or adapt the methodology you use, as well as increasing the visibility of your protocols and papers. You can also establish a dedicated workspace to collect your lab Protocols. Further information can be found at <https://www.protocols.io/help/publish-articles>.

Nature Cell Biology encourages authors presenting evidence for cell, biological, molecular, and genetic interactions to consider communicating these findings using Biofactoid (<https://biofactoid.org/>). This tool helps users share a searchable representation of interactions (e.g. binding, gene expression, post-translational modification) between genes, gene products, or chemicals. Information added to Biofactoid, with author attribution, is shared on social media and public databases, such as Pathway Commons, where it can be discovered and analyzed in the context of a large and growing corpus of knowledge.

With kind regards,

Melina Casadio, PhD
Senior Editor, Nature Cell Biology
Consulting Editor, Nature Structural & Molecular Biology
ORCID ID: <https://orcid.org/0000-0003-2389-2243>

** Visit the Springer Nature Editorial and Publishing website at http://editorial-jobs.springernature.com?utm_source=ejp_NCB_email&utm_medium=ejp_NCB_email&utm_campaign=ejp_NCB for more information about our career opportunities. If you have any questions please click [here](mailto:editorial.publishing.jobs@springernature.com).

Response to reviewers

We were very pleased to read that all three reviewers thought the work of significant interest and that the work was generally rigorous and supported our claims of mitotic inheritance of chromosome folding programs. The reviewers also asked for new experiments and analyses to further support the work. We have now addressed all comments, added extensive new experimental data and thoroughly updated most figures. All our claims continue to be fully supported. We feel that with these revisions and additions, the work is further enhanced.

Specifically, we have added the following new data and analyses:

1. We have added 74 new sequencing data sets for Cut&Run, SLAM-seq, and Hi-C experiments.
2. We have now generated new Cut&Run data to map positions of H3K27me3, H2K27ac, H3K4me3, and CTCF in mitosis and in G1 for cells with or without RanGAP1. The new data are shown in revised Figures 2 and 4. The results fully support our conclusions that MCDs are bookmarked in mitosis.
3. In response to the reviewers, we now have tested the roles of BET proteins in mediating micro-compartmentalization during mitotic exit, and in RanGAP1-depleted cells. We inhibited or depleted BET proteins in prometaphase and then released the cells into G1. Micro-compartmentalization occurred normally, and in RanGAP1-depleted cells became increasingly pronounced as cells progressed in G1. We conclude BET proteins are not essential for micro-compartmentalization.
4. We have added new analyses to show that loci enriched in H3K27Ac are generally capable of micro-compartmentalization.
5. To test roles for transcription in micro-compartmentalization, we have now performed extensive nascent transcript analysis using SLAM-seq in control and RanGAP1-depleted cells during mitotic exit and as cells enter G1. Overall, we observe progressive waves of transcriptional re-activation during G1 entry, and this is severely impeded in cells depleted for RanGAP1. Many early expressed genes overlap microcompartment domains, but we also observed that many microcompartment domains contain genes expressed much later. These results show that micro-compartmentalization is mostly independent of gene expression itself, or the timing of re-activation.
6. Figure 4 has been thoroughly revised, as recommended by all reviewers.
7. We have made textual clarifications throughout, including the use and definition of terms related to compartments, sub-compartments, micro-compartments, and domains.
8. We have edited the text to clarify the phenomena of “pruning” and “resorption”. We refrain from any mechanistic interpretations, which would be beyond the scope of the current manuscript.

Reviewer #1:

This study by Schooley et al. describes an ingenious system, to characterize how cytoplasmic proteins fold the genome following mitosis. By blocking nuclear import in synchronized cells as they enter G1, the authors show that chromosomes form microcompartments—small domains of interaction between regions with enhancer and promoter histone marks. This process is independent of cytoplasmic factors and is short lived. Early in G1, microcompartments are quickly dissolved and absorbed by A/B compartments. As this doesn't happen in cells where nuclear import is blocked this shows that loop extrusion and likely the beginning of transcription are responsible for the dissolving microcompartments. I find it also important that this manuscript validates the existence of microcompartments that had only previously been seen with extremely deep-coverage Hi-C or with Region Capture Micro-C. This is a very thorough study where the authors performed a careful analysis of the data to provide strong support for their claims.

Response

We thank the reviewer for the supportive comments. Below we address each comment in detail.

I think the following would make the study much stronger.

Point 1 Considering that the main finding of the paper is that microcompartments arise through exclusively nuclear factors inherited through mitosis, the bookmarking analysis in Figure 4 could be more complete. For example, generating H3K4me3, H3K27ac, and CTCF from cells in mitosis and G1 cells would provide a much better view of the bookmarking happening in these cells. It would also be interesting to look at the binding of TFs like AP2 but that might be much more challenging. Just like the authors did throughout the paper, this section would also benefit from specific example loci showing ATAC signal at bookmarked and non-bookmarked sites, and footprint score, along with EV, and histone marks. That would really help the reader understand better the global analysis that is shown.

Response

The reviewer brings up a really important issue. We have now generated new Cut&Run data to map positions of H3K27me3, H2K27ac, H3K4me3, and CTCF in mitosis and in G1 for cells with or without RanGAP1. The new data are shown in revised Figures 2 and 4. The results fully support our conclusion that MCDs are bookmarked in mitosis, i.e., they continue to be marked by H3K27ac and H3K4Me3 in mitosis. In cells without nuclear import during mitotic exit, these sites remain similarly marked. Thus, these loci are marked stably through mitosis and do not require cytoplasmic factors to regain or maintain their chromatin status during mitotic exit and the subsequent G1.

We now show more extensive examples of this phenomenon, as requested by the reviewer. Figure 2b now shows tracks of ATAC, H3K27ac, and H3K4me3 performed in G1 cells with and without nuclear import (RanGAP1). Stack-ups indicating the prevalence of these marks at microcompartment domains are shown in Figure 2c. In figure 4(a,c) and Extended data figure 4f, we demonstrate the presence of H3K27ac

and H3K4me3 marks in mitosis bookmarking these active regions that persist in the presence or absence of RanGAP1 as cells enter G1 and enriched at the microcompartment domains we identified.

We also show (Figure 4) new tracks of nascent transcription obtained with new SLAM-seq data for control cells (see below) to illustrate that MCDs encompass genes/promoters that are all active in this cell line (in control conditions).

We agree with the reviewer that analysis of factors like AP2 would be of interest, but we feel that is beyond the current scope of the manuscript and will be a topic of future investigation.

In addition, I thought that some plots in figure 4 were particularly hard to follow. The label 5H MCD used in figure 4 is not very intuitive. It is also not written in the figure legend or the text. That means extra effort is required each time to understand that 5H MCD represents the ATAC peaks at MCDs in 4A and the actual microcompartments in 4C. Having the consistent nomenclature in text, legend and panel labels will be helpful.

Response

All reviewers brought up related issues with Figure 4. We clearly did not explain the results in an intuitive way. We have now entirely revised Figure 4, and included important new data including new data related to mitotic marking of loci in control and RanGAP1-depleted cells, the role of BET proteins in microcompartmentalization, and new SLAM-seq data on nascent transcription. As a result Figure 4 has been extensively modified. With these changes, we hope the comments from this reviewer, and the other reviewers, pertaining to this figure are now addressed.

Panel 4B- I am not sure using the union of atac peaks is very informative here. Did the authors try to plot these same marks in the specific peaksets from figure 4A? Does that provide a clearer characterization of retained mitotic peaks? Furthermore, although the authors decided to highlight the enrichment enhancer and promoter marks, it is also clear that they contain lots of CTCF binding. So just looking at this plot it could be argued that CTCF is playing a role here, which I don't believe is true.

Response

As mentioned above, we have almost entirely replaced Figure 4 with a revised version that now shows important new datasets on bookmarking, transcription and role of BET proteins in MCD-MCD interactions. The panel that the reviewer refers to is still included but has also changed. We tried, but thought that plotting the marks for the specific peaksets from Figure 4A did not provide a clearer picture of bookmarked mitotic peaks. Instead, we have now revised this panel by adding the new mitotic H3K4Me3 and H3K27Ac Cut&Run data. We feel that this new panel now shows much more clearly that MCDs display open chromatin marked by H3K4Me3 and H3K27Ac in mitosis, and do not require RanGap1 for their establishment or maintenance. This makes the case that MCDs are bookmarked in mitosis.

Further, we now show CTCF binding at MCDs, in G1 control and RanGAP1-depleted cells. As expected, MCDs bind CTCF in G1, and this is strongly reduced in mitosis. This new data is shown in the revised Figure 4a. This is consistent with our previously published data where we reported that most CTCF dissociated from its binding sites in mitosis (Oomen et al. Genome Res. 2019, and Genome Res. 2025). In RanGAP1-depleted G1 cells we also detect CTCF binding at levels comparable to control cells. That CTCF is found to bind chromatin in RanGAP1-depleted G1 cells is expected given that CTCF is known to rapidly bind to chromatin during mitotic exit prior to full nuclear envelope closure and is supported by our immunofluorescence microscopy data (Fig 5c).

We cannot rule out roles for CTCF in mediating MCD-MCD interactions, but note that MCD-MCD interactions are not centered at CTCF sites. Also, although most MCDs do contain CTCF-bound sites, not all do.

Panel 4C – I found these plots very non intuitive and their description insufficient. What does color mean? Breaking it into 4 categories when in the end the analysis is described mostly as 0 or non zero enhancer/promoter also unnecessarily complicates the plots. I would also try to guide the reader through the plots in the text much better. It took me more than 30 minutes to understand how these plots show what the authors highlight in the text. Perhaps trying to refer what specific plot in this panel shows would help.

Response

Other reviewers also raised issues with this visualization. In the revised manuscript we have now replaced panel 4c with a panel showing stacked bar plots illustrating the overlap between MCDs and 1 or more bookmarked promoters and enhancers (generally active cCREs maintained in mitosis) as defined by our new Cut&Run datasets for H3K4Me3 and H3K27Ac obtained with G1 and mitotic cells. We feel, and hope the reviewer agrees, that this plot much better shows that MCDs tend to contain multiple bookmarked cCREs.

Point 2 – I think it would be interesting to expand the characterization of what sets microcompartments going by quantifying expression in control cells initiating G1 (either TTseq or PolIII chip). Are the first loci to be transcribed the most microcompartmentalized? The data in the paper suggests that transcription is not required for microcompartmentalization, but maybe microcompartmentalization contributes to initiate transcription post-mitosis? This should be a quick straightforward experiment and the result interesting either way.

Response

The reviewer raises an interesting point. We have now performed SLAM-seq analyses to quantify the restart of transcription during G1 entry in control cells and in cells in which RanGAP1 was depleted during prometaphase. The results show that a low level of transcription for a small set of genes (a few hundred) is observed in mitotic cells arrested in nocodazole and during the first hour of release. A large wave of post-mitotic

transcription then begins between 1 and 2 hours after mitotic release in control DLD-1 cells. In RanGAP1-depleted cells, the low level of transcription of a small set of genes in nocodazole arrested cells is still observed and a subset of early genes are transcribed in the first 2h of post-mitotic release into G1. However, nascent synthesis is globally attenuated and late gene activations programs (2-5h) are completely absent. Thus, general micro-compartmentalization is inversely correlated with global transcription: in G1 RanGAP1-depleted cells transcription is strongly inhibited, yet micro-compartmentalization is much enhanced compared to control cells. These new data are shown in the revised Figure 4e-h, Extended Figure 4d,e, and Supplementary Figure 6.

We also analyzed whether micro-compartments, and particularly those that appear the first in RanGAP1-depleted cells (i.e., during cytokinesis), involve genes that are expressed during mitosis or earliest during mitotic exit in control cells. We do find that early expressed genes are enriched at MCDs, as shown in Figure 4h, and suggest that MCD contacts could thus modulate the early wave of gene activation. However, based on the inverse relationship between MCDs and gene expression in RanGAP1-depleted cells, we conclude that microcompartment formation is correlated with H3K27Ac, but not with timing of gene re-activation in control cells. In the revised manuscript we have added this conclusion to the results section (Figure 4).

Point 3 The concept described in figure 6 is also very interesting. As in figure 4, if the readers could visualize a specific locus as an example of a microcompartment being resorbed into a larger compartment would be very helpful.

Response

We apologize for not better indicating that such examples were already visible in Figure 6. We have now explicitly indicated MCD-MCD interactions seen in RanGAP1-depleted cells, that are resorbed within larger compartments in control G1 cells (arrows added in Figure 6a).

Point 4 I thought figure 7 should have come straight after figure 1. In fact in some parts of the paper the authors refer to “as we show below”. Putting it up in the beginning wouldn’t affect anything and would actually inform the reader about how to best interpret the experiments and which factors are excluded from the nuclei.

Response

We thank the reviewer for this suggestion. We tried re-ordering the results so that what is now in Figure 7 would be presented after Figure 1. We felt such an order did not work as well as the original order because it broke up the narrative. Therefore we prefer to keep the analysis in Figure 7 at the end of the manuscript. If the editor and reviewers feel we really should change the order, we will of course follow their advice.

Smaller points

-I really appreciate how careful the authors were in being consistent with nomenclature. And figure 5i is really helpful. However, the accompanying paper uses the word microcompartments instead of microcompartment domains, which was also previously

used (Goel et al 2023). Since the good old A/B compartments also don't have the word domain, shouldn't the D be dropped in MCD?

Response

The reviewer brings up a general issue in the field of chromosome folding. We (JD) always thought of the term "compartment" to refer to the "spatial compartment" in the nucleus where multiple "domains" of a certain type cluster together, e.g., A- and B-compartments in our original Hi-C paper in 2009. Since then the term has become used in different ways including usages where the term compartment refers to a single domain of a certain type (A, or B, or a subcompartment). An example of the latter usage is Goel et al. 2023. We prefer to stick to our initial definition that a "compartment" is a cluster of more than 1 "domains". Using that definition, we refer to "micro-compartments" as spatial clusters of multiple "micro-compartment domains" (MCDs). In the revised manuscript we now rationalize and define the use of these terms more precisely. We added this statement to the results section:

"The length scale and grid-like nature of these interactions implies a chromosome-intrinsic capacity to form a type of sub-compartment independently of factors that would normally enter the nucleus after telophase. We refer to the pairwise Hi-C contacts that make up this intrinsic subcompartment as microcompartments, based on their scale and similarity to previously described structures (Goel et al. 2023; Harris et al. 2023). As shown below, microcompartment domains are assigned to multiple different conventionally defined sub-compartments in asynchronous cells but possess a similarly distinct set of properties that defines them as a specific subset of the genome".

-2b please add genomic coordinates

Response

We have added coordinates to Figure 2b.

-2c Do all K27ac and K4me3 rich regions show evidence of compartmentalization?
Could you plot EV1 at all K27ac/K3me3 regions?

Response

This is a very good suggestion. We have analyzed all K27Ac peaks and find that they interact preferentially with each other strongly in RanGAP1-depleted cells (Extended data figure 4). These sites do indeed show higher EV1 values. These new analyses that strongly support our conclusions are shown in the updated Extended Data Figure 4. We note that such enriched interactions, e.g., between active promoters defined by H3K4Me3, are also shown in Figure 2f.

-what is the last column in 3e?

Response

We apologize for not explaining this clearly. This column shows the density of loop-extrusion driven loop anchors observed around an MCD. These are derived from control cells. We have added this to the legend.

-5e does not have figure legend

Response

We apologize for the omission. In the revised manuscript we have corrected the legend.

6a-what are the two Ev1 lines below hi-c maps?

Response

Each heatmap shown in panel a contains two datasets (top and bottom triangle). The two Ev1 lines below each map correspond to these two datasets (e.g., left: the Hi-C map for cytokinesis and for the 5 hour timepoint). This is indicated in the plot.

Reviewer #2:

Review Dekker

In this paper, Schooley et al., developed degron systems for two proteins essential for nuclear transport, RanGAP1 and Nup93. These proteins were degraded pro-metaphase cells to prevent an influx of cytoplasmic proteins during mitotic exit. With this system they uncovered a transient chromatin folding intermediate program that gives rise microcompartmentalization of mitotically bookmarked CREs, which form during telophase. Microcompartmentalization is modified by the entry of cohesin and the basal transcription protein as well as other factors modulated by chromatin remodellers. By depleting RanGAP1 and Nup93 after prometaphase release, they were able to recover MCD interactions, demonstrating that the distinct G1 folding program and continuous active nuclear transport are important for maintenance pruning of non-specific interactions between MCDs. The study thus provides important new insight into chromatin folding in mitotic cells and highlights a requirement for a continuous influx of cytoplasmic factors for establishing and maintaining interphase chromatin folding.

Response

We thank the reviewer for their supportive and helpful comments. Below we address each comment in detail.

The MCDs coincide with the active CREs of a given cell type and appear to be associated with bookmarking, but this has not been established. Since the BET family, particularly BRD4, have been found to associate with mitotic chromosomes and could potentially be involved in MCD formation, the authors should perform a BRD4 ChIP-seq to see if it overlaps with MCDs and if it does they can use a BRD4 inhibitor to determine whether BRD4 inhibition impairs bookmarking upon mitotic exit and alters CTCF-mediated looping. Demonstrating a more causal role for MCDs in bookmarking would add value to the paper.

Response

The reviewer is correct that in the original manuscript we did not show directly that MCDs are bookmarked in mitosis, either in control or RanGAP1/Nup93-depleted cells. Note that we did show that MCDs maintain their open chromatin status in mitosis as detected by ATACseq.

We have now added extensive new data to show that MCDs also maintain their histone modification status in prometaphase, and that their histone modification status is not altered in cells depleted for RanGAP1. Thus, the chromatin state at MCDs is stable through mitosis and G1 entry and is independent of RanGAP1. The new data is shown in Figure 4a-d of the revised manuscript.

The reviewer also suggests that interactions between MCDs, i.e., micro-compartmentalization, may be driven by proteins of the BET family. We have now added new data to show that, perhaps surprisingly, BET proteins do not appear to drive microcompartmentalization. We treated prometaphase-arrested cells with the BET

protein inhibitors JQ1 and dBET6, concurrent with auxin-mediated degradation of RanGAP1. Western blot analysis showed that BRD2 and BRD4 are both degraded in the presence of dBET6, but not in the presence of JQ1, as expected and as reported before (Otto et al. 2019; Zheng et al. 2023; Zheng et al. 2021). When these RanGAP1-depleted and BET-inhibited/depleted cells were then released into G1, micro-compartments were formed and we could not observe any difference from cells in which only RanGAP1 was depleted. These results show that interactions between MCDs occur in the absence of BET protein activity (JQ1-treated cells), and in the absence of BRD2 and BRD4 (dBET6-treated cells). Clearly, other factors associated with these loci, or H3K27Ac itself, mediate these associations. These important new data are shown in Figure 4i and extended data Figure 4h-i.

Despite extensive efforts, we were not able to obtain high quality BRD4 Cut&Run data, but feel that the experiments described above (and now shown in the revised Figure 4), convincingly rule out a key role for BET proteins in micro-compartmentalization.

The authors demonstrate an overlap between active histone marks and MCDs in control cells, but in the absence of similar analyses in the RanGAP1 and Nup93-depleted cells, it is not clear whether these modifications change in the absence of an influx of cytoplasmic proteins. This would be important to know if they are somehow involved in the MCD interactions. These experiments should be included.

Response

The reviewer makes a very important point. As we also mention in response to the previous comment by the reviewer, we have now added important new experimental Cut&Run data showing that active histone marks occur normally at MCDs in RanGAP1-depleted cells.

This is our response to a similar comment by reviewer 1:

We have now generated new Cut&Run data to map positions of H3K27me3, H2K27ac, H3K4me3, and CTCF in mitosis and in G1 for cells with or without RanGAP1. The new data are shown in revised Figures 2 and 4. The results fully support our conclusion that MCDs are bookmarked in mitosis, i.e., they continue to be marked by H3K27ac and H3K4Me3 in mitosis. In cells without nuclear import during mitotic exit, these sites remain similarly marked. Thus, these loci are marked stably through mitosis and do not require cytoplasmic factors to regain their chromatin status during mitotic exit.

We now show more extensive examples of this phenomenon, as requested by the reviewer. Figure 2b now shows tracks of ATAC, H3K27ac, and H3K4me3 performed in G1 cells with and without nuclear import (RanGAP1). Stack-ups indicating the prevalence of these marks at microcompartment domains are shown in Figure 2c. In figure 4(a,c) and Extended data figure 4, we demonstrate the presence of H3K27ac and H3K4me3 marks in mitosis bookmarking these active regions that persist in the

presence or absence of RanGAP1 as cells enter G1 and enriched at the microcompartment domains we identified.

We also show new tracks of nascent transcription obtained with new SLAM-seq data for control cells (see below) to illustrate that MCDs encompass genes/promoters that are all active in this cell line (in control conditions).”

These important new data are shown in Figure 4 and Extended Data Figure 4..

Figure 2b and 2c: Can the authors show the ATACseq profiles and the ChIP-seq for H3K27ac and H3K4me3 also for the RanGAP1-depleted cells (at the moment only the control cells are shown).

Response

As also outlined above, and in response to reviewer 1, we have now added ATACseq profiles, and Cut&Run data for H3K27Ac and H3K4me3 for RanGAP1-depleted cells. No differences with control cells were observed, indicating that these chromatin marks do not require RanGAP1 for maintenance in G1. We also demonstrate that these marks are present in mitosis and show they are stable through mitosis even in the RanGAP1 depletion (Fig 4a,c-d)

For the 4 conditions shown in Fig 3c it would be nice to have RNA Pol II ChIP-seq for both control and RanGAP1-depleted cells, to see how the profile changes over time relative to the microcompartments.

Response

We tried but have been unable to obtain high quality Cut&Run data for RNA PolIII, either in control of RanGAP1-depleted cells. As an alternative, we have generated extensive SLAM-seq data to directly measure RNA PolIII activity (nascent transcripts) during mitotic exit. As shown in the revised Figure 4, we find that transcription re-starts genome-wide in control cells about 1-2 hours after release from mitotic arrest. In RanGAP1-depleted cells transcription is severely inhibited, though some nascent transcripts were detected in mitotic cells. Together with the proteomics and imaging data (Figure 7) that show that RNA PolIII, as well as a substantial portion of the RNA synthesis and processing machinery, is largely depleted from nuclei in RanGAP1-depleted cells, these results show there little, if any, RNA polIII active in the nuclei of RanGAP1-depleted cells. These observations further support the conclusion that transcription and RNA PolIII are unlikely to be involved in actively driving micro-compartment formation.

Minor points:

- Figures and figure legends don't have sufficient information making them hard to navigate

Response

We have revised most figures quite extensively, and have edited the legends for clarity, while adhering to the word limitations.

- No call out for Figure S1

Response

We now call out of for Figure S1 (which is now Supplemental Figure 3) when first referring to Hi-C data.

- Why are controls for the samples in Fig 1g so different?

Response

There are probably multiple reasons for this. First, the two control datasets shown in Figure 1g represent examples of individual timecourses performed with two different cell lines: the DLD-RanGAP1-AID cell line and the DLD-Nup93-AID cell line. It is typical, in our hands, that different time courses show slightly different kinetics, even when performed with the same cell line. Second, the two cell lines display different growth rates (the DLD-Nup93-AID cell line grows slower), which may affect the kinetics of import. Third, and most critically, the DLD-Nup93-AID cell line may be less import-competent even when Nup93-AID is not degraded due to the fact that the steady-state expression level of Nup93-AID is lower than the expression of untagged Nup93 in the parental line. This is shown in the new Supplemental Figure 1. In our hands, it is often the case that adding the AID tag to a protein of interest lowers its steady state expression even without induction of the degron. We note that steady-state levels of RanGAP1 are also affected by the AID tag. It appears that Nup93 levels are more limiting than RanGAP1 levels. We hope this clarifies the slight differences between the control datasets. The key finding here is that compared to their appropriate control condition (i.e., no depletion), depletion of either RanGAP1 or Nup93 blocks nuclear import.

Reviewer #3:

Summary:

Current data indicate that chromosomes re-fold every cell cycle, leading Schooley et al. to investigate here determinants of interphase chromosome conformation after cell division. They establish an experimental system to physically segregate the G1 cytoplasm from the nucleus which enables the authors to study intrinsic versus extrinsic factors that affect chromosome conformation. To accomplish this, they induce degradation of RanGAP1 and Nup93, which are essential for nuclear-cytoplasmic transport, in pro-metaphase, follow the cells into G1, which appears to work as expected based upon cell biological assays, and then primarily perform Hi-C analysis to investigate chromosome conformation, ATAC-seq to study cis regulatory elements, and label-based mass spectrometry proteomics to identify the proteins localized to G1 nuclei independent of cytoplasmic to nuclear transport. This approach is very creative and original, setting this study apart from others in the field.

Because of the authors' thoughtfulness they identify an interphase chromosome conformation determined solely by chromatin state that reflects compartmentalization of cis-regulatory elements that were bookmarked during the prior mitosis. A particular strength of the study is that induced degradation of RanGAP1 and Nup93 both show the formation of this chromosome conformation. During normal cell physiology, factors that drive biochemical processes including loop extrusion, transcription, RNA processing, etc. are inherited from the cytoplasm and impact chromosome conformation but this is distinct from chromosome intrinsic factors. This work is important for the 3D genome structure field as it provides compelling evidence that cell-type-specific compartmentalization is inherited from chromatin through the formation of an intermediate composed of self-associating chromatin features, which are subsequently modified by cytoplasmic factors. Therefore, the experiments and analyses performed are rigorous and informative on a significant question in chromosome biology that is relevant to readers of Nature Cell Biology and beyond. This reviewer would like to emphasize their enthusiasm for the work as the comments below are to improve an already impressive manuscript and hopefully will not substantially delay publication.

Response

We thank the reviewer for their supportive and also very constructive comments. Below we describe how we addressed all of them.

Major comments:

The authors claim that microcompartment formation is not limited to telophase and cytokinesis and can still occur within G1 cells upon loss of nuclear import. To test this, they perform an interesting experiment by treating cells with auxin 3.5 hours after prometaphase block release and performing Hi-C during late G1. An important control to strengthen the claim that the microcompartment is re-forming would be to show that the long-range MCD interactions have fully disappeared in control cells by 3.5 hours after prometaphase, the timepoint of auxin addition, before reforming at late G1. If the authors can complete this, they may want to consider including Extended Fig 6 as a

main figure. If not, they should consider the alternative interpretation that MCD-MCD interactions have not been fully pruned/resorbed by 3.5 hours after prometaphase block release.

Response

We thank the reviewer for this excellent suggestion. We have now performed the proposed experiment, shown in Extended Data Figure 6 in the revised manuscript. The results provide two important insights; First, at $t = 3.5$ hours MCD-MCD interactions are reduced (compared to cytokinesis) but have not yet been fully pruned/resolved in control cells. Second, when RanGAP1 is then depleted from $t=3.5$ hours onwards, MCD-MCD interactions are slightly increased, while in control cells they are substantially reduced. Extended Figure 6 also shows that cohesin is less abundant in the nucleus at later time points even if RanGAP1 depletion is initiated at $t = 3.5$ hours after mitotic release, indicating that nuclear accumulation of cohesin is at least partly RanGAP1-dependent, even in later G1. These important new data support our conclusion that microcompartmentalization can occur within G1 cells after a normal conformation is already formed during the early hours of G1.

However, we acknowledge that the magnitude of the effect is quite small, probably because bi-directional nucleo-cytoplasmic transport of cohesin is very low in RanGAP1-depleted cells during later time points. These results do support the inverse relationship between loop extrusion and microcompartment formation described in Figure 5 and our statement that microcompartmentalization can occur at later time points, but this is not a major claim of our manuscript. Therefore we prefer to keep this figure as an Extended Data Figure and not a main figure.

The approach to disrupt nuclear import is rigorous and robust as two distinct proteins, RanGAP1 and Nup93, that control compartmentalization of nuclei from cytosol by different mechanisms are separately induced for degradation. The main results and interpretations outlined in the manuscript seem to generally agree upon degradation of both factors. However, quantitatively, the authors state that lower enrichment of MCD contact frequency in Nup93 depleted cells compared to RanGAP1 depleted cells may reflect clonal variation between the two cell lines as MCDs were assigned in the latter context. The authors do have Hi-C data from Nup93 depleted cells (Extended Data Fig. 2). Can the authors please identify MCDs from the Nup93 depleted Hi-C data and determine if the quantitative differences in MCD contact frequency are due to clonal differences? An alternative explanation for quantitative differences in MCD contact frequency could be related to incomplete MBP-mScarlet-NLS nuclear localization in Nup93 control cells (Figure 1g and Extended Figure 1f).

Response

We thank the reviewer for this important comment. We have now attempted to call MCDs in the Nup93-depleted cells. We discovered that our MCD-calling approach identified many CTCF-CTCF loops in Nup93-depleted cells (but not in RanGAP1-depleted cells). This is illustrated in the figure below. Initially we were surprised by this, but then we realized that several lines of evidence show that in Nup93-depleted cells

there is still a considerable level of cohesin-mediated loop extrusion: First, the $P(s)$ curve shows evidence for loops as seen by the characteristic bump in interaction frequency for loci separated by about 100 kb. Second, CTCF-CTCF loops are reduced compared to control cells, but not to the same extent as observed in RanGAP1-depleted cells (compare Extended Data Figure 5a, derivative of $P(s)$ plot with Main Figure 5a, derivative plots). Finally, the presence of significant levels of remaining loop extrusion features, such as TADs, visible in the Hi-C interaction heatmaps of Nup93-depleted cells (see Extended Data Figure 2a, arrows) explains why we detect weaker micro-compartmentalization compared to RanGAP1-depleted cells.

We conclude that the Nup93-depleted cells show weaker micro-compartmentalization in part because more cohesin was able to enter the nucleus in that cell line.

~1300 MCD anchors detected in a combination of $\Delta N93-5hr$ and $\Delta N93-10hr$ samples.

Figure legend

The figure shows a series of stack-ups where each row is a MCD locus called in cells in which Nup93 was depleted. We split the set of MCDs in two groups: The top group was also called in cells in which RanGAP1 was depleted (Delta-RG MCDs). These MCD display all features expected for MCDs: high levels of H3K27Ac, high levels of H3K4Me3, low levels of H3K27Me3. Bottom group of MCDs were not called in RanGAP1-depleted cells. This latter set have features inconsistent with MCDs, and instead appear to be CTCF-bound sites and their pairwise interactions are most likely the result of loop extrusion. We conclude that in Nup93-depleted cells there is sufficient loop extrusion still occurring that leads to detectable CTCF-CTCF loops. These loops complicate our MCD calling approach. (Blue lines on top of the stack-ups represent the average profile for the set of MCDs that were also called in RanGAP1-depleted cells, orange lines represent the average profile for the set of MCDs that were only called on Nup93-depleted cells.

To make the manuscript as clear and impactful as possible, the authors may want to consider terminology. The chromosome conformation capture field now has three terms related to compartmentalization: compartments, subcompartments, and microcompartments. It is currently unclear the significant difference between these terms; a possibility is new terms are being added as technology evolves without an underlying meaningful difference between the terms. Furthermore, investigators outside the chromosome conformation capture field, who likely will be very interested in this exciting manuscript, are unlikely to understand the differences between compartments, subcompartments, and microcompartments. Fig 6 might be confusing for these readers where some subcompartments (the B subcompartments) are pooled, whereas others are not. And, based on terms alone, questions arise such as what is the difference between a subcompartment and microcompartment (both imply small)? Or, are they all just compartments? A suggestion from this reviewer would be for the authors to clearly and explicitly define each of the three terms (compartments, subcompartments, and microcompartments) or to pick one term while acknowledging that there are other terms and ambiguity, at present, in the field.

Response

The reviewer is correct that the compartment terminology used in the literature is quite confusing. We propose that micro-compartments are different from conventionally defined subcompartments (e.g., defined by Hi-C interaction profiles combined with histone modification data, as in Rao et al. Cell 2014, and Spracklin et al. NSMB 2023). Such compartments are referred to as sub-compartments or IGPs (They overlap to a significant amount but differ in the computational methods by which they are called). Micro-compartments detected in early G1 represent a subset of loci from several conventionally defined sub-compartments (as we show in our manuscript in Figure 6d, they are embedded within A1, A2, and VVI IPGs. Thus, MCDs are somehow different, and they surely behave distinctly from subcompartments during G1 entry, as we show here.

To clarify our usage of the terms we added this statement to the results section:

“The length scale and grid-like nature of these interactions implies a chromosome-intrinsic capacity to form a type of sub-compartment independently of factors that would normally enter the nucleus after telophase. We refer to the pairwise Hi-C contacts that make up this intrinsic subcompartment as microcompartments, based on their scale and similarity to previously described structures (Goel et al. 2023; Harris et al. 2023). As shown below, these microcompartment domains form a subset of loci assigned to several conventionally defined sub-compartments, suggesting their distinct properties”.

When thinking about physiological normal G1 cells, the present interpretation is that MCD-MCD interactions are pruned in G1. I would appreciate the authors' thoughts on

the following questions: Is cohesin-dependent loop extrusion actively pruning the interactions, or, in a population of cells (Hi-C experiments presented here are bulk experiments) are more of the Hi-C interactions in G1 representative of loop extrusion interactions compared to MCD-MCD interactions? Put another way, can the authors distinguish pruning, an active process, from a superimposition of folding states, a technical consideration when interpreting Hi-C data?

Response

This is a really interesting question. As we describe in our manuscript, we indeed think pruning, i.e., the loss of long-range MCD-MCD interactions while short-range MCD-MCD interactions are maintained, is due to cohesin-mediated loop extrusion. We propose that loop extrusion, and blocking of that process at CTCF-bound sites within MCDs, reinforces MCD-MCD interactions that occur within the loop extrusion range (up to a few hundred Kb). By reinforcing shorter-range MCD-MCD interactions, longer-range MCD-MCD interactions become less favored and in many cases they are lost entirely (see comments on resorption where MCD-MCD interactions are replaced with compartment/subcompartment-level associations). We do not think the observed changes in long vs. short-range MCD-MCD interactions are simply due to superimposition of folding states, because many long-range MCD-MCD interactions entirely disappear (see Figure 6a, arrows). Also, it is well established that loop extrusion can disrupt compartmental interaction in general (e.g., Schwarzer et al. Nature 2017). Exploring how addition of loop extrusion alters patterns of MCD-MCD interactions through experiments and polymer simulations are ongoing in our lab but we feel these are beyond the scope of the current manuscript.

Fig. 6D shows that MCD-MCD interactions change from cytokinesis to G1 in control cells, but whether this change is resorption is unclear. This reviewer would expect the 2x2 boxes in cis for the A1, A2, and VVI IPGs (regardless of MCD or not) in control G1 cells to be uniform in contact frequency if resorption were occurring. If the cCRE microcompartment is being resorbed by active subcompartments, then why is the VVI-MCD IPG depleted for interactions with the A1 IPG in control G1 cells in cis?

Response

The reviewer is correct that when resorption were complete the 2x2 boxes in Figure 6D for interactions in cis for the A1, A2, and VVI IPGs would be uniform. This is not what we see: the MCDs for each IPG still interact the most with other MCDs of the same type (on diagonal signals in 6D). This is likely the result of the fact that MCDs tend to still interact somewhat in G1 (see also Figure 1e), but in G1 these interactions are now restricted to MCDs of the *same* IPG. We observe that frequent interactions between MCDs across *different* IPGs during cytokinesis are substantially reduced in G1 cells. This is what we propose is the result of resorption.

The reason that the VVI-MCD IPG is not interacting with the A1 IPG in control G1 cells is simply because these IPGs are defined in control cells as sets of loci that self-interact but not with other IPGs. Thus MCDs are resorbed into their corresponding IPG: VVI-MCD loci are resorbed within VVI, and not in A1. We hope this clarifies this issue. In the

revised manuscript we added a statement that MCDs get resorbed within their corresponding IPG subcompartments.

For interactions in trans, in control G1 cells the contact frequency for IPGs is more what one might expect if resorption were occurring, but the contact frequencies at cytokinesis don't show as clear distinction between MCDs and subcompartments as the cis contact frequencies. For example, the A1-MCD IPG interacts more strongly with the A1 IPG than with the A2-MCD IPG indicating that the A1-MCD IPG is more like the A1 subcompartment than the cCRE microcompartment in trans in control cells in cytokinesis. Either the concept of resorption needs to be clarified, or other interpretations considered, such as that the data presented in Fig 6D also reflect the cytoplasmic folding program thereby impacting interpretation of results.

Response

We agree that in trans there is one case that seems different from all others. In general the A1-MCDs interact frequently with MCDs from other IPGs in cytokinesis and these strong interactions become much weaker in G1, while they remain or even get stronger in RanGAP1-depleted cells. The exception is frequent trans-interaction between A1-MCDs and other loci in the A1 subcompartment. We do not know why this is the case. We consider two possibilities: first, the A1 subcompartment generally shows very strong trans interactions, as also shown in other datasets (e.g., Spracklin et al. NSMB 2023, Scelfo et al. 2024), and this IPG may behave uniquely in trans as compared to other IPGs. Second, the A1-IPG is strongly enriched in MCDs. Given that our MCD calling method is intentionally very conservative, MCD calling likely missed many weaker MCDs and the A1-IPG may very likely contain many unannotated MCDs, and these may lead to the high interaction frequency detected. Nonetheless, interactions between A1-MCDs and other A1 loci do increase over time in both cis and trans, while A1-MCD interactions with A2- and VV1-MCDs are relatively reduced in control G1.

This leads to the following questions: Is resorption distinct from pruning? If so, how can these be distinguished?

Response

This is a very interesting question. We do not know the answer to this question, and answering it would require experiments where one attempts to disrupt one or the other. Such experiments would be beyond the scope of the current manuscript. In the discussion we have edited the section in pruning and resorption and added some outstanding questions.

The authors discuss the possibility that histone modifications such as H3K27ac may mediate affinity-driven compartmentalization through bridging factors such as Brd4. An omitted and relevant study to this discussion is the work of Rosencrance et al., Mol Cell, 2020, which revealed that hyperacetylated chromatin domains formed by the fusion oncoprotein BRD4-NUT make chromatin interactions comparable to those observed

here. Combined with the observation by immunofluorescence microscopy that BRD4-NUT localizes to mitotic chromosomes (Fig. 1A from Alekseyenko et al., Genes & Development, 2015) these studies are consistent with and support the work presented here on a connection between mitotic bookmarking and affinity-driven compartmentalization.

Response

This is an excellent comment. In the revised manuscript we have now added new data to address the role of BET proteins. In short we find that, perhaps surprisingly, BET proteins, and especially BRD2 and BRD4 are most likely not involved in micro-compartment formation. We responded to a similar comment by reviewer 2 in the following way:

“The reviewer is correct that in the original manuscript we did not show directly that MCDs are bookmarked in mitosis, either in control or RanGAP1/Nup93-depleted cells. We have now added extensive new data to show that MCDs maintain their histone modification status in prometaphase, and that their histone modification status is not altered in cells depleted for RanGAP1. Thus, the chromatin state at MCDs is stable through mitosis and G1 entry and independent of RanGAP1. The new data is shown in Figure 4a-d of the revised manuscript.

The reviewer also suggests that interactions between MCDs, i.e., micro-compartmentalization may be driven by proteins of the BET family. We have now added new data to show that that appears to be not the case. We treated prometaphase-arrested cells with the BET protein inhibitor JQ1 or with dBET6, while we also depleted RanGAP1 through addition of auxin. Western blot analysis showed that BRD2 and BRD4 are both degraded in the presence of dBET6, but not in the presence of JQ1, as expected and as reported before (Otto et al. 2019; Zheng et al. 2023; Zheng et al. 2021) . When these RanGAP1-depleted and BET-inhibited/depleted cells were then released into G1, micro-compartments were formed and we could not observe any difference from cells in which only RanGAP1 was depleted. These results show that interactions between MCDs occur in the absence of BET protein activity (JQ1-treated cells), or at least in the absence of BRD2 and BRD4 (dBET-treated cells). Clearly, other factors associated with these loci, or H3K27Ac itself, mediate these associations. These important new data are shown in Figure 4i, Extended Data Figure 4h,i. “

Our data, together with the data presented by Rosencrance et al. Mol Cell 2020 and Alekseyenko et al., Genes & Development, 2015, suggests that high levels of H3K27Ac are required for microcompartment formation but not BRD-protein binding *per se*.

Minor comments:

Fig 1B: Can the authors please perform a Western blot from parental DLD-1 cells of untagged RanGAP1 and Nup93? Does the AID tag affect stability of either protein?

Response

We thank the reviewer for pointing out this omission. We have added this Western blot to the new Supplemental Figure 1. This plot shows that addition of the AID tag to Nup93 does indeed affect protein levels. In our hands, it is often the case that adding the AID tag to a protein of interest lowers its steady state expression even without induction of the degran. The reduced levels of these proteins in uninduced cells are sufficient to support growth, although Nup93-AID cells grow somewhat slower than RanGAP1-AID cells and import seems to be somewhat reduced (See Figure 1, and response to a related comment from reviewer 2 above).

The resolution at which the detection of MCDs was performed should be stated in the main text.

Response

We have now added that MCDs were detected at 10 kb resolution.

Fig 2D and F, Extended Data Fig 3: Please define dELS, pELS, and PLS in the figure legends.

Response

We have now explained these abbreviations in the legend.

Fig 4C: Legend: circles overlap legend text. Please add a legend for the color scale.

Response

This panel is no longer included in the revised manuscript.

The figure legends for Extended Data Fig 4D and E do not match the figure.

Response

This panel is no longer included in the revised manuscript.

Datapoint labels in Supplementary Fig 2C and D are not clear. How do the labels in panels C and D correspond to panel A?

Response

We have updated and clarified this Supplemental Figure (now Supplemental Figure 3).

Supplementary Fig 3 is very helpful and clear, but the legend starts with c rather than a.

Response

We have updated and corrected all legends.

Page 6: "Furthermore, nucleoli and nuclear speckles, two import dependent organelles with important roles in nuclear organization (Prasanth et al. 2003; Spector and Lamond 2011; Leung et al. 2004; Görlich and Kutay 1999), are not found in RanGAP1-AID or Nup93-AID-depleted nuclei and apparent interchromatin granules with strong speckle

protein (SON) localization are found exclusively in the cytoplasm (Extended Data Fig. 1f).” This should refer to Extended Data Fig 1E.

Response

We apologize for the error. In the revised Extended Data Figure 1 we changed the labeling of the panels, and now this statement refers to panel F (which used to be panel E in the original submission).